# Uniform-in-Time Wasserstein Stability Bounds for (Noisy) Stochastic Gradient Descent

**Lingjiong Zhu[1], Mert Gürbüzbalaban[2,3], Anant Raj[4,5], Umut Şimşekli[5]**
1: Dept. of Mathematics, Florida State University
2: Dept. of Management Science and Information Systems, Rutgers Business School
3: Center for Statistics and Machine Learning, Princeton University
4: Coordinated Science Laboratory, University of Illinois Urbana-Champaign
5: Inria Paris, CNRS, Ecole Normale Supérieure, PSL Research University

## Abstract

Algorithmic stability is an important notion that has proven powerful for deriving generalization bounds for practical algorithms. The last decade has witnessed an increasing number of stability bounds for different algorithms applied on different classes of loss functions. While these bounds have illuminated various properties of optimization algorithms, the analysis of each case typically required a different proof technique with significantly different mathematical tools. In this study, we make a novel connection between learning theory and applied probability and introduce a unified guideline for proving Wasserstein stability bounds for stochastic optimization algorithms. We illustrate our approach on stochastic gradient descent (SGD) and we obtain time-uniform stability bounds (i.e., the bound does not increase with the number of iterations) for strongly convex losses and non-convex losses with additive noise, where we recover similar results to the prior art or extend them to more general cases by using a single proof technique. Our approach is flexible and can be generalizable to other popular optimizers, as it mainly requires developing Lyapunov functions, which are often readily available in the literature. It also illustrates that ergodicity is an important component for obtaining time-uniform bounds – which might not be achieved for convex or non-convex losses unless additional noise is injected to the iterates. Finally, we slightly stretch our analysis technique and prove time-uniform bounds for SGD under convex and non-convex losses (without additional additive noise), which, to our knowledge, is novel.

## 1 Introduction

With the development of modern machine learning applications, understanding the generalization properties of stochastic gradient descent (SGD) has become a major challenge in statistical learning theory. In this context, the main goal is to obtain computable upper-bounds on the *population risk* associated with the output of the SGD algorithm that is given as follows: $F(\theta) := \mathbb{E}_{x \sim \mathcal{D}}[f(\theta, x)]$, where $x \in \mathcal{X}$ denotes a random data point, $\mathcal{D}$ is the (unknown) data distribution defined on the data space $\mathcal{X}$, $\theta$ denotes the parameter vector, and $f : \mathbb{R}^d \times \mathcal{X} \to \mathbb{R}$ is an instantaneous loss function.

In a practical setting, directly minimizing $F(\theta)$ is not typically possible as $\mathcal{D}$ is unknown; yet one typically has access to a finite data set $X_n = \{x_1, \dots, x_n\} \in \mathcal{X}^n$, where we assume each $x_i$ is independent and identically distributed (i.i.d.) with the common distribution $\mathcal{D}$. Hence, given $X_n$, one can then attempt to minimize the *empirical risk* $\hat{F}(\theta, X_n) := \frac{1}{n} \sum_{i=1}^{n} f(\theta, x_i)$ as a proxy for $F(\theta)$. In this setting, SGD has been one of the most popular optimization algorithms for minimizing

37th Conference on Neural Information Processing Systems (NeurIPS 2023).

$\hat{F}(\theta)$ and is based on the following recursion:

$$\theta_k = \theta_{k-1} - \eta\tilde{\nabla}\hat{F}_k(\theta_{k-1}, X_n), \qquad \tilde{\nabla}\hat{F}_k(\theta_{k-1}, X_n) := \frac{1}{b}\sum_{i\in\Omega_k}\nabla f(\theta_{k-1}, x_i), \qquad (1.1)$$

where $\eta$ is the step-size, $b$ is the batch-size, $\Omega_k$ is the minibatch that is chosen randomly from the set $\{1, 2, \ldots, n\}$, and its cardinality satisfies $|\Omega_k| = b$.

One fruitful approach for estimating the population risk attained by SGD, i.e., $F(\theta_k)$, is based on the following simple decomposition:

$$F(\theta_k) \leq \hat{F}(\theta_k) + |\hat{F}(\theta_k) - F(\theta_k)|, \qquad (1.2)$$

where the last term is called the *generalization error*. Once a computable upper-bound for the generalization error can be obtained, this decomposition directly leads to a computable upper bound for the population risk $F(\theta_k)$, since $\hat{F}(\theta_k)$ can be computed thanks to the availability of $X_n$. Hence, the challenge here is reduced to derive upper-bounds on $|\hat{F}(\theta_k) - F(\theta_k)|$, typically referred to as generalization bounds.

Among many approaches for deriving generalization bounds, *algorithmic stability* [BE02] has been one of the most fruitful notions that have paved the way to numerous generalization bounds for stochastic optimization algorithms [HRS16, CJY18, MWZZ18, FV19, LY20, ZZB$^+$22]. In a nutshell, algorithmic stability measures how much the algorithm output differs if we replace one data point in $X_n$ with a new sample. More precisely, in the context of SGD, given another data set $\hat{X}_n = \{\hat{x}_1, \ldots, \hat{x}_n\} = \{x_1, \ldots, x_{i-1}, \hat{x}_i, x_{i+1}, \ldots x_n\} \in \mathcal{X}^n$ that differs from $X_n$ by at most one element, we (theoretically) consider running SGD on $\hat{X}_n$, i.e.,

$$\hat{\theta}_k = \hat{\theta}_{k-1} - \eta\tilde{\nabla}\hat{F}_k(\hat{\theta}_{k-1}, \hat{X}_n), \qquad \tilde{\nabla}\hat{F}_k(\hat{\theta}_{k-1}, \hat{X}_n) := \frac{1}{b}\sum_{i\in\Omega_k}\nabla f(\hat{\theta}_{k-1}, \hat{x}_i), \qquad (1.3)$$

and we are interested in the discrepancy between $\theta_k$ and $\hat{\theta}_k$ in some precise sense (to be formally defined in the next section). The wisdom of algorithmic stability indicates that a smaller discrepancy between $\theta_k$ and $\hat{\theta}_k$ implies a smaller generalization error.

The last decade has witnessed an increasing number of stability bounds for different algorithms applied on different classes of loss functions. In a pioneering study, [HRS16] proved a variety of stability bounds for SGD, for strongly convex, convex, and non-convex problems. Their analysis showed that, under strong convexity and bounded gradient assumptions, the generalization error of SGD with constant step-size is of order $n^{-1}$; whereas for general convex and non-convex problems, their bounds diverged with the number of iterations (even with a projection step), unless a decreasing step-size is used. In subsequent studies [LY20, KWS23] extended the results of [HRS16], by either relaxing the assumptions or generalizing the setting to more general algorithms. However, their bounds still diverged for constant step-sizes, unless strong convexity is assumed. In a recent study, [BFGT20] proved stability lower-bounds for projected SGD when the loss is convex and non-smooth. Their results showed for general non-smooth loss functions we cannot expect to prove time-uniform (i.e., non-divergent with the number of iterations) stability bounds for SGD, even when a projection step is appended.

In a related line of research, several studies investigated the algorithmic stability of the stochastic gradient Langevin dynamics (SGLD) algorithm [WT11], which is essentially a 'noisy' version of SGD that uses the following recursion: $\theta_k = \theta_{k-1} - \eta\tilde{\nabla}\hat{F}_k(\theta_{k-1}, X_n) + \xi_k$, where $(\xi_k)_{k\geq 0}$ is a sequence of i.i.d. Gaussian vectors, independent of $\theta_{k-1}$ and $\Omega_k$. The authors of [RRT17, MWZZ18] proved stability bounds for SGLD for non-convex losses, which were then extended to more general (non-Gaussian) noise settings in [LLQ19]. While these bounds hinted at the benefits of additional noise in terms of stability, they still increased with the number of iterations, which limited the impact of their results. More recently, [FR21] proved the first time-uniform stability bounds for SGLD under non-convexity, indicating that, with the presence of additive Gaussian noise, better stability bounds can be achieved. Their time-uniform results were then extended to non-Gaussian, heavy-tailed perturbations in [RBG$^+$23, RZGŞ23] for quadratic and a class of non-convex problems.

While these bounds have illuminated various properties of optimization algorithms, the analysis of each case typically required a different proof technique with significantly different mathematical

tools. Hence, it is not straightforward to extend the existing techniques to different algorithms with different classes of loss functions. Moreover, currently, it is not clear how the noisy perturbations affect algorithmic stability so that time-uniform bounds can be achieved, and more generally, it is not clear in which circumstances one might hope for time-uniform stability bounds.

In this study, we contribute to this line of research and prove novel time-uniform algorithmic stability bounds for SGD and its noisy versions. Our main contributions are as follows:

- We make a novel connection between learning theory and applied probability, and introduce a unified guideline for proving Wasserstein stability bounds for stochastic optimization algorithms with a constant step-size. Our approach is based on Markov chain perturbation theory [RS18], which offers a three-step proof technique for deriving stability bounds: (i) showing the optimizer is geometrically ergodic, (ii) obtaining a Lyapunov function for the optimizer and the loss, and (iii) bounding the discrepancy between the Markov transition kernels associated with the chains $(\theta_k)_{k\geq 0}$ and $(\hat{\theta}_k)_{k\geq 0}$. We illustrate this approach on SGD and show that time-uniform stability bounds can be obtained under a pseudo-Lipschitz-like condition for smooth strongly-convex losses (we recover similar results to the ones of [HRS16]) and a class of non-convex losses (that satisfy a dissipativity condition) when a noisy perturbation with finite variance (not necessarily Gaussian, hence more general than [FR21]) is introduced. Our results shed more light on the role of the additional noise in terms of obtaining time-uniform bounds: in the non-convex case the optimizer might not be geometrically ergodic unless additional noise is introduced, hence the bound cannot be obtained. Moreover, our approach is flexible and can be generalizable to other popular optimizers, as it mainly requires developing Lyapunov functions, which are often readily available in the literature [AFGO20, LRP16, FGO+22, GRZ22, LGY20, AFGO19].

- We then investigate the case where no additional noise is introduced to the SGD recursion and the geometric ergodicity condition does not hold. First, for non-convex losses, we prove a time-uniform stability bound, where the bound converges to a positive number (instead of zero) as $n \to \infty$, and this limit depends on the 'level of non-convexity'. Then, we consider a class of (non-strongly) convex functions and prove stability bounds for the stationary distribution of $(\theta_k)_{k\geq 0}$, which vanish as $n$ increases. To the best of our knowledge, these results are novel, and indicate that the stability bounds do not need to increase with time even under non-convexity and without additional perturbations; yet, they might have a different nature depending on the problem class.

One limitation of our analysis is that it requires Lipschitz surrogate loss functions and does not directly handle the original loss function, due to the use of the Wasserstein distance [RRT+16]. Yet, surrogate losses have been readily utilized in the recent stability literature (e.g., [RBG+23, RZGŞ23]) and we believe that our analysis might illuminate uncovered aspects of SGD even with this requirement. All the proofs are provided in the Appendix.

## 2 Technical Background

### 2.1 The Wasserstein distance and Wasserstein algorithmic stability

**Wasserstein distance.** For $p \geq 1$, the $p$-Wasserstein distance between two probability measures $\mu$ and $\nu$ on $\mathbb{R}^d$ is defined as [Vil09]:

$$\mathcal{W}_p(\mu, \nu) = \{\inf \mathbb{E}\|X - Y\|^p\}^{1/p}, \tag{2.1}$$

where the infimum is taken over all couplings of $X \sim \mu$ and $Y \sim \nu$. In particular, the dual representation for the 1-Wasserstein distance is given as [Vil09]:

$$\mathcal{W}_1(\mu, \nu) = \sup_{h \in \mathrm{Lip}(1)} \left| \int_{\mathbb{R}^d} h(x)\mu(dx) - \int_{\mathbb{R}^d} h(x)\nu(dx) \right|, \tag{2.2}$$

where $\mathrm{Lip}(1)$ consists of the functions $h : \mathbb{R}^d \to \mathbb{R}$ that are 1-Lipschitz.

**Wasserstein algorithmic stability.** Algorithmic stability is a crucial concept in learning theory that has led to numerous significant theoretical breakthroughs [BE02, HRS16]. To begin, we will present the definition of algorithmic stability as stated in [HRS16]:

**Definition 2.1** ([HRS16], Definition 2.1). *Let $\mathcal{RV}(\mathbb{R}^d)$ denote the set of $\mathbb{R}^d$-valued random vectors. For a (surrogate) loss function $\ell : \mathbb{R}^d \times \mathcal{X} \to \mathbb{R}$, an algorithm $\mathcal{A} : \bigcup_{n=1}^{\infty} \mathcal{X}^n \to \mathcal{RV}(\mathbb{R}^d)$ is $\varepsilon$-uniformly stable if*

$$\sup_{X \cong \hat{X}} \sup_{z \in \mathcal{X}} \mathbb{E}\left[\ell(\mathcal{A}(X), z) - \ell(\mathcal{A}(\hat{X}), z)\right] \leq \varepsilon, \tag{2.3}$$

*where the first supremum is taken over data $X, \hat{X} \in \mathcal{X}^n$ that differ by one element, denoted by $X \cong \hat{X}$.*

In this context, we purposefully employ a distinct notation for the loss function $\ell$ (in contrast to $f$) since our theoretical framework necessitates measuring algorithmic stability through a surrogate loss function, which may differ from the original loss $f$. More precisely, our bounds will be based on the 1-Wasserstein distance, hence, we will need the surrogate loss $\ell$ to be a Lipschitz continuous function, as we will detail in (2.2). On the other hand, for the original loss $f$ we will need some form of convexity (e.g., strongly convex, convex, or dissipative) and we will need the gradient of $f$ to be Lipschitz continuous, in order to derive Wasserstein bounds. Unfortunately, under these assumptions, we cannot further impose $f$ itself to be Lipschitz, hence the need for surrogate losses. Nevertheless, the usage of surrogate losses is common in learning theory, see e.g, [FR21, RZGŞ23], and we present concrete practical examples in the Appendix.

Now, we present a result from [HRS16] that establishes a connection between algorithmic stability and the generalization performance of a randomized algorithm. Prior to presenting the result, we define the empirical and population risks with respect to the loss function $\ell$ as follows:

$$\hat{R}(\theta, X_n) := \frac{1}{n} \sum_{i=1}^{n} \ell(\theta, x_i), \quad R(\theta) := \mathbb{E}_{x \sim \mathcal{D}}[\ell(\theta, x)].$$

**Theorem 2.1** ([HRS16], Theorem 2.2). *Suppose that $\mathcal{A}$ is an $\varepsilon$-uniformly stable algorithm, then the expected generalization error is bounded by*

$$\left|\mathbb{E}_{\mathcal{A}, X_n}\left[\hat{R}(\mathcal{A}(X_n), X_n) - R(\mathcal{A}(X_n))\right]\right| \leq \varepsilon. \tag{2.4}$$

For a randomized algorithm, if $\nu$ and $\hat{\nu}$ denotes the law of $\mathcal{A}(X)$ and $\mathcal{A}(\hat{X})$ then for a $\mathcal{L}$-Lipschitz surrogate loss function $\ell$, we have the following generalization error guarantee,

$$\left|\mathbb{E}_{\mathcal{A}, X_n}\left[\hat{R}(\mathcal{A}(X_n), X_n) - R(\mathcal{A}(X_n))\right]\right| \leq \mathcal{L} \sup_{X \cong \hat{X}} \mathcal{W}_1(\nu, \hat{\nu}). \tag{2.5}$$

The above result can be directly obtained from the combination of the results given in (2.2), Definition 2.1, and Theorem 2.1 (see also [RRT$^+$16]).

## 2.2 Perturbation theory for Markov chains

Next, we recall the Wasserstein perturbation bound for Markov chains from [RS18]. Let $(\theta_n)_{n=0}^{\infty}$ be a Markov chain with transition kernel $P$ and initial distribution $p_0$, i.e., we have almost surely

$$\mathbb{P}(\theta_n \in A | \theta_0, \cdots, \theta_{n-1}) = \mathbb{P}(\theta_n \in A | \theta_{n-1}) = P(\theta_{n-1}, A), \tag{2.6}$$

and $p_0(A) = \mathbb{P}(\theta_0 \in A)$ for any measurable set $A \subseteq \mathbb{R}^d$ and $n \in \mathbb{N}$. We assume that $(\hat{\theta}_n)_{n=0}^{\infty}$ is another Markov chain with transition kernel $\hat{P}$ and initial distribution $\hat{p}_0$. We denote by $p_n$ the distribution of $\theta_n$ and by $\hat{p}_n$ the distribution of $\hat{\theta}_n$. By $\delta_\theta$, we denote the Dirac delta distribution at $\theta$, i.e. the probability measure concentrated at $\theta$. For a measurable set $A \subseteq \mathbb{R}^d$, we also use the notation $\delta_\theta P(A) := P(\theta, A)$.

**Lemma 2.1** ([RS18], Theorem 3.1). *Assume that there exist some $\rho \in [0, 1)$ and $C \in (0, \infty)$ such that*

$$\sup_{\theta, \tilde{\theta} \in \mathbb{R}^d : \theta \neq \tilde{\theta}} \frac{\mathcal{W}_1(P^n(\theta, \cdot), P^n(\tilde{\theta}, \cdot))}{\|\theta - \tilde{\theta}\|} \leq C\rho^n, \tag{2.7}$$

*for any $n \in \mathbb{N}$. Further assume that there exist some $\delta \in (0, 1)$ and $L \in (0, \infty)$ and a measurable Lyapunov function $\hat{V} : \mathbb{R}^d \to [1, \infty)$ of $\hat{P}$ such that for any $\theta \in \mathbb{R}^d$:*

$$(\hat{P}\hat{V})(\theta) \leq \delta \hat{V}(\theta) + L, \tag{2.8}$$

*where* $(\hat{P}\hat{V})(\theta) := \int_{\mathbb{R}^d} \hat{V}(\hat{\theta})\hat{P}(\theta, d\hat{\theta})$. *Then, we have*

$$\mathcal{W}_1(p_n, \hat{p}_n) \leq C\left(\rho^n \mathcal{W}_1(p_0, \hat{p}_0) + (1 - \rho^n)\frac{\gamma\kappa}{1 - \rho}\right), \qquad (2.9)$$

*where* $\gamma := \sup_{\theta \in \mathbb{R}^d} \frac{\mathcal{W}_1(\delta_\theta P, \delta_\theta \hat{P})}{\hat{V}(\theta)}$, $\quad \kappa := \max\left\{\int_{\mathbb{R}^d} \hat{V}(\theta)d\hat{p}_0(\theta), \frac{L}{1-\delta}\right\}$.

Lemma 2.1 provides a sufficient condition for the distributions $p_n$ and $\hat{p}_n$ after $n$ iterations to stay close to each other given the initial distributions $p_0$. Lemma 2.1 will provide a key role in helping us derive the main results in Section 3.1 and Section 3.2. Later, in the Appendix, we will state and prove a modification of Lemma 2.1 (see Lemma E.5 in the Appendix) that will be crucial to obtaining the main result in Section 3.3.

# 3 Wasserstein Stability of SGD via Markov Chain Perturbations

In this section, we will derive *time-uniform* Wasserstein stability bounds for SGD by using the perturbation theory presented in [RS18]. Before considering general losses that can be non-convex, we first consider the simpler case of quadratic losses to illustrate our key ideas.

## 3.1 Warm up: quadratic case

To illustrate the proof technique, we start by considering a quadratic loss of the form: $f(\theta, x_i) := (a_i^\top \theta - y_i)^2/2$ where, $x_i := (a_i, y_i)$ and $\nabla f(\theta, x_i) = a_i(a_i^\top \theta - y_i)$. In this setting, the SGD recursion takes the following form:

$$\theta_k = \left(I - \frac{\eta}{b}H_k\right)\theta_{k-1} + \frac{\eta}{b}q_k, \quad \text{where,} \quad H_k := \sum_{i \in \Omega_k} a_i a_i^\top, \quad q_k := \sum_{i \in \Omega_k} a_i y_i. \qquad (3.1)$$

The sequence $(H_k, q_k)$ are i.i.d. and for every $k$, $(H_k, q_k)$ is independent of $\theta_{k-1}$.

Similarly, we can write down the iterates of SGD with a different data set $\hat{X}_n := \{\hat{x}_1, \ldots, \hat{x}_n\}$ with $\hat{x}_i = (\hat{a}_i, \hat{y}_i)$, where $\hat{X}_n$ differs from $X_n$ with at most one element:

$$\hat{\theta}_k = \left(I - \frac{\eta}{b}\hat{H}_k\right)\hat{\theta}_{k-1} + \frac{\eta}{b}\hat{q}_k, \quad \text{where} \quad \hat{H}_k := \sum_{i \in \Omega_k} \hat{a}_i \hat{a}_i^\top, \quad \hat{q}_k := \sum_{i \in \Omega_k} \hat{a}_i \hat{y}_i. \qquad (3.2)$$

Our goal is to obtain an algorithmic stability bound, through estimating the 1-Wasserstein distance between the distribution of $\theta_k$ and $\hat{\theta}_k$ and we will now illustrate the three-step proof technique that we described in Section 1. To be able to apply the perturbation theory [RS18], we start by establishing the geometric ergodicity of the Markov process $(\theta_k)_{k \geq 0}$ with transition kernel $P(\theta, \cdot)$, given in the following lemma.

**Lemma 3.1.** *Assume that* $\rho := \mathbb{E}\left\|I - \frac{\eta}{b}H_1\right\| < 1$. *Then, for any* $k \in \mathbb{N}$, *we have the following inequality:* $\mathcal{W}_1\left(P^k(\theta, \cdot), P^k\left(\tilde{\theta}, \cdot\right)\right) \leq \rho^k \|\theta - \tilde{\theta}\|$.

We note that since $H_1 \succeq 0$, the assumption in Lemma 3.1 can be satisfied under mild assumptions, for example when $H_1 \succ 0$ with a positive probability, which is satisfied for $\eta$ small enough.

In the second step, we construct a Lyapunov function $\hat{V}$ that satisfies the conditions of Lemma 2.1.

**Lemma 3.2.** *Let* $\hat{V}(\theta) := 1 + \|\theta\|$. *Assume that* $\hat{\rho} := \mathbb{E}\left\|I - \frac{\eta}{b}\hat{H}_1\right\| < 1$. *Then, we have*

$$(\hat{P}\hat{V})(\theta) \leq \hat{\rho}\hat{V}(\theta) + 1 - \hat{\rho} + \frac{\eta}{b}\mathbb{E}\|\hat{q}_1\|. \qquad (3.3)$$

In our third and last step, we estimate the perturbation gap based on the Lyapunov function $\hat{V}$ in the form of (2.7), assuming that the data is bounded. Such bounded data assumptions have been commonly made in the literature [Bac14, BM13].

**Lemma 3.3.** *If* $\sup_{x \in \mathcal{X}} \|x\| \leq D$ *for some* $D < \infty$, *then, we have* $\sup_{\theta \in \mathbb{R}^d} \frac{\mathcal{W}_1(\delta_\theta P, \delta_\theta \hat{P})}{\hat{V}(\theta)} \leq \frac{2\eta D^2}{n}$.

Note that Lemma 2.1 relies on three conditions: the Wasserstein contraction in (2.7), which is obtained through Lemma 3.1, the drift condition for the Lyapunov function in (2.8), which is obtained in Lemma 3.2 and finally the estimate on $\gamma$ in (2.9) which is about the one-step 1-Wasserstein distance between two semi-groups that in our context are associated with two datasets that differ by at most one element, which is obtained in Lemma 3.3. The only place the neighborhood assumption ($\sup_{x \in \mathcal{X}} \|x\| \leq D$) is used is in the expression of $\gamma$ in equation (2.9). Now, having all the ingredients, we can invoke Lemma 2.1 and we obtain the following result which provides a 1-Wasserstein bound between the distribution of iterates when applied to datasets that differ by one point.

For $Y \in \bigcup_{n=1}^{\infty} \mathcal{X}^n$ and $k \geq 0$, let $\nu(Y, k)$ denote the law of the $k$-th the SGD iterate when $Y$ is used as the dataset, i.e., $\nu(X, k)$ and $\nu(\hat{X}, k)$ denote the distributions of $\theta_k$ and $\hat{\theta}_k$ obtained by the recursions (3.1) and (3.2) respectively. As shorthand notation, set $\nu_k := \nu(X, k)$ and $\hat{\nu}_k := \nu(\hat{X}, k)$.

**Theorem 3.1.** *Assume $\theta_0 = \hat{\theta}_0 = \theta$. We also assume that $\rho := \mathbb{E}\left\|I - \frac{\eta}{b}H_1\right\| < 1$ and $\hat{\rho} := \mathbb{E}\left\|I - \frac{\eta}{b}\hat{H}_1\right\| < 1$ and $\sup_{x \in \mathcal{X}} \|x\| \leq D$ for some $D < \infty$. Then, we have*

$$\mathcal{W}_1(\nu_k, \hat{\nu}_k) \leq \frac{1 - \rho^k}{1 - \rho} \frac{2\eta D^2}{n} \max\left\{1 + \|\theta\|, \frac{1 - \hat{\rho} + \frac{\eta}{b}\mathbb{E}\|\hat{q}_1\|}{1 - \hat{\rho}}\right\}. \tag{3.4}$$

*Proof.* The result directly follows from Lemma 3.1, Lemma 3.2, Lemma 3.3 and Lemma 2.1. $\square$

By a direct application of (2.5), we can obtain a generalization bound for an $\mathcal{L}$-Lipschitz surrogate loss function, as follows:

$$\left|\mathbb{E}_{\mathcal{A}, X_n}\left[\hat{R}(\mathcal{A}(X_n), X_n) - R(\mathcal{A}(X_n))\right]\right| \leq \frac{\mathcal{L}}{1 - \rho_0} \frac{2\eta D^2}{n} \max\left\{1 + \|\theta\|, \frac{1 - \rho_0 + \frac{\eta}{b}\mathbb{E}\|\hat{q}_1\|}{1 - \rho_0}\right\},$$

where $\rho_0 = \sup_X \|1 - \frac{\eta}{b}H_X\|$, $H_X = \sum_{i \in \Omega_k, a_j \in X} a_j a_j^\top$ and $X$ is a random set of $n$-data points from the data generating distribution. The generalization bound obtained above does not include the mean square error in the unbounded case but covers a larger class of surrogate loss functions. Because of this incompatibility, a direct comparison is not possible; however, the rate obtained in the equation above has the same dependence on the number of samples that were obtained in the previous works [LY20]. For least squares, there are other works using integral operators that develop generalization bounds for SGD under a capacity condition [LR17, PVRB18]. However, these bounds only hold for the least square loss.

## 3.2 Strongly convex case

Next, we consider strongly convex losses. In the remainder of the paper, we will always assume that for every $x \in \mathcal{X}$, $f(\cdot, x)$ is differentiable.

Before proceeding to the stability bound, we first introduce the following assumptions.

**Assumption 3.1.** *There exist constants $K_1, K_2 > 0$ such that for any $\theta, \hat{\theta} \in \mathbb{R}^d$ and every $x \in \mathcal{X}$,*

$$\|\nabla f(\theta, x) - \nabla f(\hat{\theta}, \hat{x})\| \leq K_1 \|\theta - \hat{\theta}\| + K_2 \|x - \hat{x}\|(\|\theta\| + \|\hat{\theta}\| + 1). \tag{3.5}$$

This assumption is a pseudo-Lipschitz-like condition on $\nabla f$ and is satisfied for various problems such as GLMs [Bac14]. Next, we assume that the loss function $f$ is strongly convex.

**Assumption 3.2.** *There exists a universal constant $\mu > 0$ such that for any $\theta_1, \theta_2 \in \mathbb{R}^d$ and $x \in \mathcal{X}$,*

$$\langle \nabla f(\theta_1, x) - \nabla f(\theta_2, x), \theta_1 - \theta_2 \rangle \geq \mu \|\theta_1 - \theta_2\|^2.$$

By using the same recipe as we used for quadratic losses, we obtain the following stability result.

**Theorem 3.2.** *Let $\theta_0 = \hat{\theta}_0 = \theta$. Assume that Assumption 3.1 and Assumption 3.2 hold. We also assume that $\eta < \min\left\{\frac{1}{\mu}, \frac{\mu}{K_1^2 + 64D^2 K_2^2}\right\}$, $\sup_{x \in \mathcal{X}} \|x\| \leq D$ for some $D < \infty$, and $\sup_{x \in \mathcal{X}} \|\nabla f(0, x)\| \leq E$ for some $E < \infty$. Let $\nu_k$ and $\hat{\nu}_k$ denote the distributions of $\theta_k$ and*

$\hat\theta_k$ respectively. Then, we have

$$\mathcal{W}_1(\nu_k,\hat\nu_k) \leq \frac{8DK_2(1-(1-\frac{\eta\mu}{2})^k)}{n\mu}\left(\frac{2E}{\mu}+1\right)$$
$$\cdot \max\left\{1+2\|\theta\|^2+\frac{2E^2}{\mu^2}, 2-\frac{\eta}{\mu}K_1^2-\frac{56\eta}{\mu}D^2K_2^2+\frac{64\eta}{\mu^3}D^2K_2^2E^2\right\}. \quad (3.6)$$

Similarly to the quadratic case, we can now directly obtain a bound on expected generalization error using (2.5). More precisely, for an $\mathcal{L}$-Lipschitz surrogate loss function $\ell$, we have

$$\left|\mathbb{E}_{\mathcal{A},X_n}\left[\hat R(\mathcal{A}(X_n),X_n)-R(\mathcal{A}(X_n))\right]\right| \leq \mathcal{L}\cdot\frac{8DK_2(1-(1-\frac{\eta\mu}{2})^k)}{n\mu}\left(\frac{2E}{\mu}+1\right)$$
$$\cdot \max\left\{1+2\|\theta\|^2+\frac{2E^2}{\mu^2}, 2-\frac{\eta}{\mu}K_1^2-\frac{56\eta}{\mu}D^2K_2^2+\frac{64\eta}{\mu^3}D^2K_2^2E^2\right\}.$$

The bound above has the same dependence on the number of samples as the ones of the previous stability analysis of (projected) SGD for strongly convex functions [HRS16, LLNT17, LY20]. However, we have a worse dependence on the strong convexity parameter $\mu$.

## 3.3 Non-convex case with additive noise

Finally, we consider a class of non-convex loss functions. We assume that the loss function satisfies the following dissipativity condition.

**Assumption 3.3.** *There exist constants $m>0$ and $K>0$ such that for any $\theta_1,\theta_2\in\mathbb{R}^d$ and $x\in\mathcal{X}$,*
$$\langle\nabla f(\theta_1,x)-\nabla f(\theta_2,x),\theta_1-\theta_2\rangle \geq m\|\theta_1-\theta_2\|^2-K.$$

The class of dissipative functions satisfying this assumption are the ones that admit some gradient growth in radial directions outside a compact set. Inside the compact set though, they can have quite general non-convexity patterns. As concrete examples, they include certain one-hidden-layer neural networks [AS23]; they arise in non-convex formulations of classification problems (e.g. in logistic regression with a sigmoid/non-convex link function); they can also arise in robust regression problems, see e.g. [GGZ22]. Also, any function that is strongly convex outside of a ball of radius for some will satisfy this assumption. Consequently, regularized regression problems where the loss is a strongly convex quadratic plus a smooth penalty that grows slower than a quadratic will belong to this class; a concrete example would be smoothed Lasso regression; many other examples are also given in [EHZ22]. Dissipative functions also arise frequently in the sampling and Bayesian learning and global convergence in non-convex optimization literature [RRT17, GGZ22].

Unlike the strongly-convex case, we can no longer obtain a Wasserstein contraction bound using the synchronous coupling technique as we did in the proof of Theorem 3.2. To circumvent this problem, in this setting, we consider a noisy version of SGD, with the following recursion:

$$\theta_k = \theta_{k-1}-\eta\tilde\nabla\hat F_k(\theta_{k-1},X_n)+\eta\xi_k, \qquad \tilde\nabla\hat F_k(\theta_{k-1},X_n):=\frac{1}{b}\sum_{i\in\Omega_k}\nabla f(\theta_{k-1},x_i), \quad (3.7)$$

where $\xi_k$ are additional i.i.d. random vectors in $\mathbb{R}^d$, independent of $\theta_{k-1}$ and $\Omega_k$, satisfying the following assumption.

**Assumption 3.4.** *$\xi_1$ is random vector on $\mathbb{R}^d$ with a continuous density $p(x)$ that is positive everywhere, i.e. $p(x)>0$ for any $x\in\mathbb{R}^d$ and $\mathbb{E}[\xi_1]=0$, $\sigma^2:=\mathbb{E}\left[\|\xi_1\|^2\right]<\infty$.*

Note that the SGLD algorithm [WT11] is a special case of this recursion, whilst our noise model can accommodate non-Gaussian distributions with finite second-order moment.

Analogously, let us define the (noisy) SGD recursion with the data set $\hat X_n$ as

$$\hat\theta_k = \hat\theta_{k-1}-\eta\tilde\nabla\hat F_k(\hat\theta_{k-1},\hat X_n)+\eta\xi_k,$$

and let $p(\theta,\theta_1)$ denote the probability density function of $\theta_1=\theta-\frac{\eta}{b}\sum_{i\in\Omega_1}\nabla f(\theta,x_i)+\eta\xi_1$. Further let $\theta_*$ be a minimizer of $\hat F(\cdot,X_n)$. Then, by following the same three-step recipe, we obtain the following stability bound. Here, we do not provide all the constants explicitly for the sake of clarity; the complete theorem statement is given in Theorem E.1 (Appendix E.1).

**Theorem 3.3.** *Let $\theta_0 = \hat\theta_0 = \theta$. Assume that Assumption 3.1, Assumption 3.3 and Assumption 3.4 hold. We also assume that $\eta < \min\left\{\frac{1}{m}, \frac{m}{K_1^2 + 64D^2 K_2^2}\right\}$ and $\sup_{x \in \mathcal{X}} \|x\| \leq D$ for some $D < \infty$ and $\sup_{x \in \mathcal{X}} \|\nabla f(0, x)\| \leq E$ for some $E < \infty$. For any $\hat\eta \in (0, 1)$, define $M > 0$ so that $\int_{\|\theta_1 - \theta_*\| \leq M} p(\theta_*, \theta_1) d\theta_1 \geq \sqrt{\hat\eta}$ and for any $R > \frac{2K_0}{m}$ where $K_0$ is defined in (E.2) so that*

$$\inf_{\theta, \theta_1 \in \mathbb{R}^d: V(\theta) \leq R, \|\theta_1 - \theta_*\| \leq M} \frac{p(\theta, \theta_1)}{p(\theta_*, \theta_1)} \geq \sqrt{\hat\eta}. \tag{3.8}$$

*Let $\nu_k$ and $\hat\nu_k$ denote the distributions of $\theta_k$ and $\hat\theta_k$ respectively. Then, we have*

$$\mathcal{W}_1(\nu_k, \hat\nu_k) \leq \frac{C_1(1 - \bar\eta^k)}{2\sqrt{\psi(1 + \psi)(1 - \bar\eta)}} \cdot \frac{2b}{n}, \tag{3.9}$$

*where for any $\eta_0 \in (0, \hat\eta)$ and $\gamma_0 \in \left(1 - m\eta + \frac{2\eta K_0}{R}, 1\right)$, $\psi = \frac{\eta_0}{\eta K_0}$ and $\bar\eta = (1 - (\hat\eta - \eta_0)) \vee \frac{2 + R\psi\gamma_0}{2 + R\psi}$. The constant $C_1 \equiv C_1(\psi, \eta, \hat\eta, R, \gamma_0, K_1, K_2, \sigma^2, D, E)$ is explicitly stated in the proof.*

Contrary to our previous results, the proof technique for showing Wasserstein contraction (as in Lemma 3.1) for this theorem relies on verifying the drift condition (Assumption B.1) and the minorization condition (Assumption B.2) as given in [HM11]. Once these conditions are satisfied, we invoke the explicitly computable bounds on the convergence of Markov chains developed in [HM11].

From equation (2.5), we directly obtain the following generalization error bound for $\mathcal{L}$-Lipschitz surrogate loss function,

$$\left| \mathbb{E}_{\mathcal{A}, X_n} \left[ \hat R(\mathcal{A}(X_n), X_n) - R(\mathcal{A}(X_n)) \right] \right| \leq \mathcal{L} \cdot \frac{C_1(1 - \bar\eta^k)}{2\sqrt{\psi(1 + \psi)(1 - \bar\eta)}} \cdot \frac{2b}{n},$$

where the constants are defined in Theorem 3.3[1]. The above result can be directly compared with the result in [FR21, Theorem 4.1] that has the same dependence on $n$ and $b$. However, our result is more general in the sense that we do not assume our noise to be Gaussian noise. Note that [LLQ19] can also accommodate non-Gaussian noise; however, their bounds increase with the number of iterations.

**Remark 3.4.** *In Theorem 3.3, we can take $R = \frac{2K_0}{m}(1 + \epsilon)$ for some fixed $\epsilon \in (0, 1)$ so we can take*

$$\hat\eta = \left( \max_{M > 0} \left\{ \min \left\{ \int_{\|\theta_1 - \theta_*\| \leq M} p(\theta_*, \theta_1) d\theta_1, \inf_{\substack{\theta, \theta_1 \in \mathbb{R}^d: \|\theta_1 - \theta_*\| \leq M \\ \|\theta - \theta_*\|^2 \leq \frac{2K_0}{m}(1 + \epsilon) - 1}} \frac{p(\theta, \theta_1)}{p(\theta_*, \theta_1)} \right\} \right\} \right)^2. \tag{3.10}$$

*Moreover, we can take $\eta_0 = \frac{\hat\eta}{2}$, $\gamma_0 = 1 - \frac{m\eta\epsilon}{2}$, and $\psi = \frac{\hat\eta}{2\eta K_0}$, so that*

$$\bar\eta = \max\left\{ 1 - \frac{\hat\eta}{2}, \frac{2 + \frac{(1+\epsilon)\hat\eta}{m}(1 - \frac{m\eta\epsilon}{2})}{2 + \frac{(1+\epsilon)\hat\eta}{m}} \right\} = 1 - \frac{m\eta\epsilon(1 + \epsilon)\hat\eta}{4m + 2(1 + \epsilon)\hat\eta}, \tag{3.11}$$

*provided that $\eta \leq 1$. Note that the parameter $\hat\eta$ in (3.10) appears in the upper bound in equation (3.9) that controls the 1-Wasserstein algorithmic stability of the SGD. It is easy to see from equation (3.9) that the smaller $\bar\eta$, the smaller the 1-Wasserstein bound. By the defintion of $\bar\eta$, the larger $\hat\eta$, the smaller the 1-Wasserstein bound. As a result, we would like to choose $\hat\eta$ to be as large as possible, and the equation (3.10) provides an explicit value that $\hat\eta$ can take, which is already the largest as possible.*

Next, let us provide some explicitly computable lower bounds for $\hat\eta$ in (3.10). This is achievable if we specify further the noise assumption. Under the assumption that $\xi_k$ are i.i.d. Gaussian distributed, we have the following corollary.

---

[1] By using the decomposition (1.2), we can obtain excess risk bounds for SGLD by combining our results with [XR17]: it was shown that gradient Langevin dynamics has the following optimization error is $O(\varepsilon + d^{3/2}b^{-1/4}\lambda^{-1}\log 1/\varepsilon)$ after $K = O(d\varepsilon^{-1}\lambda^{-1}\log 1/\varepsilon)$ iterations, where $b$ is the mini-batch size and $\lambda$ is the uniform spectral gap of the continuous-time Langevin dynamics. Similar results are given for SGLD in [XR17, Theorem 3.6].

**Corollary 3.5.** *Under the assumptions in Theorem 3.3, we further assume the noise $\xi_k$ are i.i.d. Gaussian $\mathcal{N}(0, \Sigma)$ so that $\mathbb{E}[\|\xi_1\|^2] = tr(\Sigma) = \sigma^2$. We also assume that $\Sigma \prec I_d$. Then, we have*

$$
\hat{\eta} \geq \max_{M \geq \eta \sup_{x \in \mathcal{X}} \|\nabla f(\theta_*, x)\|} \left\{ \min \left\{ \left( 1 - \frac{\exp(-\frac{1}{2}(\frac{M}{\eta} - \sup_{x \in \mathcal{X}} \|\nabla f(\theta_*, x)\|)^2)}{\sqrt{\det (I_d - \Sigma)}} \right)^2, \right. \right.
$$

$$
\exp \left\{ -\frac{(1 + K_1 \eta) \left( \frac{2K_0}{m}(1 + \epsilon) - 1 \right)^{1/2}}{\eta^2} \|\Sigma^{-1}\| \right.
$$

$$
\left. \left. \left. \cdot \left( (1 + K_1 \eta) \left( \frac{2K_0}{m}(1 + \epsilon) - 1 \right)^{1/2} + 2 \left( M + \eta \sup_{x \in \mathcal{X}} \|\nabla f(\theta_*, x)\| \right) \right) \right\} \right\} \right\}. \quad (3.12)
$$

The above corollary provides an explicit lower bound for $\hat{\eta}$ (instead of the less transparent inequality constraints in Theorem 3.3), and by combining with Remark 3.4 (see equation (3.11)) leads to an explicit formula for $\bar{\eta}$ which is essential to characterize the Wasserstein upper bound in (3.9) in Theorem 3.3.

# 4 Wasserstein Stability of SGD without Geometric Ergodicity

While the Markov chain perturbation theory enabled us to develop stability bounds for the case where we can ensure geometric ergodicity in the Wasserstein sense (i.e., proving contraction bounds), we have observed that such a strong ergodicity notion might not hold for non-strongly convex losses. In this section, we will prove two more stability bounds for SGD, without relying on [RS18], hence without requiring geometric ergodicity. To the best of our knowledge, these are the first uniform-time stability bounds for the considered classes of convex and non-convex problems.

## 4.1 Non-convex case without additive noise

The stability result we obtained in Theorem 3.3 required us to introduce an additional noise (Assumption 3.4) to be able to invoke Lemma 2.1. We will now show that it is possible to use a more direct approach to obtain 2-Wasserstein algorithmic stability in the non-convex case under Assumption 3.3 without relying on [RS18]. However, we will observe that without geometric ergodicity will have a non-vanishing bias term in the bound. Note that, since $\mathcal{W}_1(\nu_k, \hat{\nu}_k) \leq \mathcal{W}_p(\nu_k, \hat{\nu}_k)$ for all $p \geq 1$, the following bound still yields a generalization bound by (2.5).

**Theorem 4.1.** *Assume $\theta_0 = \hat{\theta}_0 = \theta$. We also assume that Assumption 3.1 and Assumption 3.3 hold and $\eta < \min \left\{ \frac{1}{m}, \frac{m}{K_1^2 + 64D^2 K_2^2} \right\}$ and $\sup_{x \in \mathcal{X}} \|x\| \leq D$ for some $D < \infty$ and $\sup_{x \in \mathcal{X}} \|\nabla f(0, x)\| \leq E$ for some $E < \infty$. Let $\nu_k$ and $\hat{\nu}_k$ denote the distributions of $\theta_k$ and $\hat{\theta}_k$ respectively. Then, we have*

$$
\mathcal{W}_2^2(\nu_k, \hat{\nu}_k) \leq \left( 1 - (1 - \eta m)^k \right) \cdot \left( \frac{4D^2 K_2^2 \eta (8B + 2)}{bnm} + \frac{4K_2 D(1 + K_1 \eta)}{nm}(1 + 5B) + \frac{2K}{m} \right), \quad (4.1)
$$

*where the constant $B$ is explicitly defined in the proof.*

While the bound (4.1) does not increase with the number of iterations, it is easy to see that it does not vanish as $n \to \infty$, and it is small only when $K$ from the dissipativity condition (Assumption 3.3) is small. In other words, if we consider $K$ to be the level of non-convexity (e.g., $K = 0$ corresponds to strong convexity), as the function becomes 'more non-convex' the persistent term in the bound will get larger. While this persistent term might make the bound vacuous when $n \to \infty$, for moderate $n$ the bound can be still informative as the persistent term might be dominated by the first two terms.

Moreover, discarding the persistent bias term, this bound leads to a generalization bound with rate $n^{-1/2}$, rather than $n^{-1}$ as before. This indicates that it is beneficial to add additional noise $\xi_k$ in SGD as in Theorem 3.3 in order for the dynamics to be geometrically ergodic that can lead to a sharp bound as $n \to \infty$. Finally, we note that as Theorem 4.1 involves 2-Wasserstein distance, it can pave the way for generalization bounds without requiring a surrogate loss. Yet, this is not immediate and would require deriving uniform $L^2$ bounds for the iterates, e.g., [RRT17].

## 4.2 Convex case with additional geometric structure

We now present our final stability bound, where we consider relaxing the strong convexity assumption (Assumption 3.2) to the following milder assumption.

**Assumption 4.1.** *There exists universal constants $\mu > 0$ and $p \in (1, 2)$ such that for any $\theta_1, \theta_2 \in \mathbb{R}^d$ and $x \in \mathcal{X}$, $\langle \nabla f(\theta_1, x) - \nabla f(\theta_2, x), \theta_1 - \theta_2 \rangle \geq \mu \|\theta_1 - \theta_2\|^p$.*

Note that as $p < 2$, the function class can be seen as an intermediate class between convex and strongly convex functions, and such a class of functions has been studied in the optimization literature [Dun81, Ber15].

We analogously modify Assumption 3.1 and consider the following assumption.

**Assumption 4.2.** *There exist constants $K_1, K_2 > 0$ and $p \in (1, 2)$ such that for any $\theta, \hat{\theta} \in \mathbb{R}^d$ and every $x \in \mathcal{X}$, $\|\nabla f(\theta, x) - \nabla f(\hat{\theta}, \hat{x})\| \leq K_1 \|\theta - \hat{\theta}\|^{\frac{p}{2}} + K_2 \|x - \hat{x}\|(\|\theta\|^{p-1} + \|\hat{\theta}\|^{p-1} + 1)$.*

The next theorem establishes a stability bound for the considered class of convex losses in the stationary regime of SGD.

**Theorem 4.2.** *Let $\theta_0 = \hat{\theta}_0 = \theta$. Suppose Assumption 4.1 and Assumption 4.2 hold (with $p \in (1, 2)$) and $\eta \leq \frac{\mu}{K_1^2 + 2^{p+4} D^2 K_2^2}$ and $\sup_{x \in \mathcal{X}} \|x\| \leq D$ for some $D < \infty$ and $\sup_{x \in \mathcal{X}} \|\nabla f(0, x)\| \leq E$ for some $E < \infty$. Then $\nu_k$ and $\hat{\nu}_k$ converge to the unique stationary distributions $\nu_\infty$ and $\hat{\nu}_\infty$ respectively and moreover, we have*

$$\mathcal{W}_p^p(\nu_\infty, \hat{\nu}_\infty) \leq \frac{C_2}{bn} + \frac{C_3}{n}, \tag{4.2}$$

*where the constants $C_2 \equiv C_2(\eta, \mu, K_2, D, E)$ and $C_3 \equiv C_3(\eta, \mu, K_1, K_2, D, E)$ are explicitly stated in the proof.*

While we have relaxed the geometric ergodicity condition for this case, in the proof of Theorem 4.2, we show that the Markov chain $(\theta_k)_{k \geq 0}$ is simply ergodic, i.e., $\lim_{k \to \infty} \mathcal{W}_p(\nu_k, \nu_\infty) = 0$. Hence, even though we still obtain a time-uniform bound, our bound holds asymptotically in $k$, due to the lack of an explicit convergence rate for $\mathcal{W}_p(\nu_k, \nu_\infty)$. On the other hand, the lack of strong convexity here results in a generalization bound with rate $n^{-1/p}$, whereas for the strongly convex case, i.e., $p = 2$, we previously obtained a rate of $n^{-1}$. This might be an indicator that there might be still room for improvement in terms of the rate, at least for this class of loss functions.

## 5 Conclusion

We proved time-uniform Wasserstein-stability bounds for SGD and its noisy versions under different strongly convex, convex, and non-convex classes of functions. By making a connection to Markov chain perturbation results [RS18], we introduced a three-step guideline for proving stability bounds for stochastic optimizers. As this approach required geometric ergodicity, we finally relaxed this condition and proved two other stability bounds for a large class of loss functions.

The main limitation of our approach is that it requires Lipschitz surrogate loss functions, as it is based on the Wasserstein distance. Hence, our natural next step will be to extend our analysis without such a requirement. Finally, due to the theoretical nature of this study, it does not contain any direct potential societal impacts.

## Acknowledgments

Lingjiong Zhu is partially supported by the grants NSF DMS-2053454, NSF DMS-2208303, and a Simons Foundation Collaboration Grant. Mert Gürbüzbalaban's research are supported in part by the grants Office of Naval Research Award Number N00014-21-1-2244, National Science Foundation (NSF) CCF-1814888, NSF DMS-2053485. Anant Raj is supported by the a Marie Sklodowska-Curie Fellowship (project NN-OVEROPT 101030817). Umut Şimşekli's research is supported by the French government under management of Agence Nationale de la Recherche as part of the "Investissements d'avenir" program, reference ANR-19-P3IA-0001 (PRAIRIE 3IA Institute) and the European Research Council Starting Grant DYNASTY – 101039676.

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

# Uniform-in-Time Wasserstein Stability Bounds for (Noisy) Stochastic Gradient Descent

## APPENDIX

The Appendix is organized as follows:

- In Section A, we provide further details and examples about the usage of surrogate losses.
- In Section B, we provide technical background for the computable bounds for the convergence of Markov chains which will be used to prove the results in Section 3.3 in the main paper.
- In Section C, we provide technical proofs for 1-Wasserstein perturbation results for the quadratic loss in Section 3.1 in the main paper.
- In Section D, we provide technical proofs for 1-Wasserstein perturbation results for the strongly-convex loss in Section 3.2 in the main paper.
- In Section E, we provide technical proofs for 1-Wasserstein perturbation results for the non-convex loss (with additive noise) in Section 3.3 in the main paper.
- In Section F, we provide technical proofs for 2-Wasserstein stability bounds for the non-convex loss without additive noise in Section 4.1 in the main paper.
- In Section G, we provide technical proofs for $p$-Wasserstein stability bounds for the convex loss with additional geometric structure in Section 4.2 in the main paper.

## A  On the Usage of Surrogate Losses

While the requirement of surrogate losses is a drawback of our framework, nevertheless our setup can cover several practical settings. In this section, we will provide two such examples.

**Example 1.**   We can choose the surrogate loss as the *truncated loss*, such that:

$$\ell(\theta, x) = \min(f(\theta, x), C),$$

where $C > 0$ is a chosen constant. This can be seen as a "robust" version of the original loss, which has been widely used in robust optimization and is conceptually similar to adding a projection step to the optimizer.

**Example 2.**   Another natural setup for our framework is the $\ell_2$-regularized Lipschitz loss that was also used in [FR21]. As opposed to the previous case, for the sake of this example, let us consider $\ell$ as the true loss and $f$ as the surrogate loss. Then, we can choose the pair $f$ and $\ell$ as follows:

$$f(\theta, x) = \ell(\theta, x) + \frac{\mu}{2}\|\theta\|_2^2,$$

where $\mu > 0$. Intuitively, this setting means that, we have a true loss $\ell$ which can be Lipschitz, but in the optimization framework we consider a regularized version of the loss. In other words, we have a loss $\ell$; however, we run the algorithm on the regularized loss $f$ to have better convergence properties, and finally, we would like to understand if the algorithm generalizes on $\ell$ or not, and we are typically not interested if the algorithm generalizes well on the regularized loss $f$.

Next, we illustrate how a generalization bound for the loss $f$, i.e., $\left|\mathbb{E}[\hat{F}(\theta) - F(\theta)]\right|$. For this example, a bound on the quantity can be obtained by building on our analysis. To obtain such a bound, in addition to the bounds that we developed on $\left|\mathbb{E}[\hat{R}(\theta) - R(\theta)]\right|$, we would need to estimate the following quantity:

$$\left|\mathbb{E}_{\theta, X_n}\left[\frac{1}{n}\sum_{i=1}^{n}(f(\theta, x_i) - \ell(\theta, x_i))\right]\right|.$$

For illustration purposes, assume that $\ell$ is convex and Lipschitz in the first parameter. Then, $f$ is $\mu$-strongly convex. Further consider that we initialize SGD from 0, i.e., $\theta_0 = 0$ and set the batch

size $b$ to 1. Denote $\theta = \theta_k$ as the $k$-th iterate of SGD when applied on $\hat{F}(\theta, X_n)$, i.e., (1.1). Further define the minimum:

$$\theta^\star_{X_n} = \arg\min_\theta \hat{F}(\theta, X_n).$$

We can now analyze the error induced by the surrogate loss as follows:

$$
\begin{aligned}
\left| \mathbb{E}_{\theta, X_n} \left[ \frac{1}{n} \sum_{i=1}^n \left( f(\theta, x_i) - \ell(\theta, x_i) \right) \right] \right| &= \frac{\mu}{2} \mathbb{E}_{\theta, X_n} \|\theta\|^2 = \frac{\mu}{2} \mathbb{E}_{\theta, X_n} \left\| \theta - \theta^\star_{X_n} + \theta^\star_{X_n} \right\|^2 \\
&\leq \mu \mathbb{E}_{\theta, X_n} \left\| \theta - \theta^\star_{X_n} \right\|^2 + \mu \mathbb{E}_{X_n} \left\| \theta^\star_{X_n} \right\|^2 \\
&\leq \mu \mathbb{E}_{X_n} \left[ (1 - \eta\mu)^k \left\| \theta^\star_{X_n} \right\|^2 + \frac{2\eta}{\mu} \sigma_{X_n} \right] + \mu \mathbb{E}_{X_n} \left\| \theta^\star_{X_n} \right\|^2 \\
&= \mu \left( (1 - \eta\mu)^k + 1 \right) \mathbb{E}_{X_n} \left\| \theta^\star_{X_n} \right\|^2 + 2\eta \mathbb{E}_{X_n} \left[ \sigma_{X_n} \right].
\end{aligned}
$$

Here, the second inequality follows from standard convergence analysis for SGD [GG23, Theorem 5.7] and we define $\sigma_{X_n}$ as the stochastic gradient noise variance:

$$\sigma_{X_n} := \mathrm{Var}\left[ \nabla f\left(\theta^\star_{X_n}, x_i\right) \right],$$

where for a random vector $V$ we define $\mathrm{Var}[V] := \mathbb{E}\|V - \mathbb{E}[V]\|^2$. Hence, we can see that the error induced by the surrogate loss depends on the following factors:

- The regularization parameter $\mu$,
- The expected norm of the minimizers,
- The step-size $\eta$,
- The expected stochastic gradient noise variance.

These terms can be controlled by adjusting $\mu$ and $\eta$.

# B  Technical Background

## B.1  Computable bounds for the convergence of Markov chains

Geometric ergodicity and convergence rate of Markov chains has been well studied in the literature [MT93, MT94, HM11]. In this section, we state a result from [HM11] that provides an explicitly computable bound on the Wasserstein contraction for the Markov chains that satisfies a drift condition that relies on the construction of an appropriate Lyapunov function and a minorization condition.

Let $\mathcal{P}(\theta, \cdot)$ be a Markov transition kernel for a Markov chain $(\theta_k)$ on $\mathbb{R}^d$. For any measurable function $\varphi : \mathbb{R}^d \to [0, +\infty]$, we define:

$$(\mathcal{P}\varphi)(\theta) = \int_{\mathbb{R}^d} \varphi(\tilde{\theta}) \mathcal{P}(\theta, d\tilde{\theta}).$$

**Assumption B.1** (Drift Condition). *There exists a function $V : \mathbb{R}^d \to [0, \infty)$ and some constants $K \geq 0$ and $\gamma \in (0, 1)$ so that*

$$(\mathcal{P}V)(\theta) \leq \gamma V(\theta) + K,$$

*for all $\theta \in \mathbb{R}^d$.*

**Assumption B.2** (Minorization Condition). *There exists some constant $\hat{\eta} \in (0, 1)$ and a probability measure $\nu$ so that*

$$\inf_{\theta \in \mathbb{R}^d : V(\theta) \leq R} \mathcal{P}(\theta, \cdot) \geq \hat{\eta}\nu(\cdot),$$

*for some $R > 2K/(1 - \gamma)$.*

We define the weighted total variation distance:

$$d_\psi(\mu_1, \mu_2) = \int_{\mathbb{R}^d} (1 + \psi V(\theta)) |\mu_1 - \mu_2|(d\theta),$$

where $\psi > 0$ and $V(\theta)$ is the Lyapunov function that satisfies the drift condition (Assumption B.1). It is known that $d_\psi$ has the following alternative expression [HM11]:

$$d_\psi(\mu_1, \mu_2) = \sup_{\varphi: \|\varphi\|_\psi \leq 1} \int_{\mathbb{R}^d} \varphi(\theta)(\mu_1 - \mu_2)(d\theta),$$

where $\| \cdot \|_\psi$ is the weighted supremum norm such that for any $\psi > 0$:

$$\|\varphi\|_\psi := \sup_{\theta \in \mathbb{R}^d} \frac{|\varphi(\theta)|}{1 + \psi V(\theta)}.$$

It is also noted in [HM11] that $d_\psi$ has yet another equivalent expression:

$$d_\psi(\mu_1, \mu_2) = \sup_{\varphi: \|\|\varphi\|\|_\psi \leq 1} \int_{\mathbb{R}^d} \varphi(\theta)(\mu_1 - \mu_2)(d\theta),$$

where

$$\|\|\varphi\|\|_\psi := \sup_{\theta \neq \tilde{\theta}} \frac{|\varphi(\theta) - \varphi(\tilde{\theta})|}{2 + \psi V(\theta) + \psi V(\tilde{\theta})}.$$

**Lemma B.1** (Theorem 1.3. [HM11]). *If the drift condition (Assumption B.1) and minorization condition (Assumption B.2) hold, then there exists $\bar{\eta} \in (0,1)$ and $\psi > 0$ so that*

$$d_\psi(\mathcal{P}\mu_1, \mathcal{P}\mu_2) \leq \bar{\eta} d_\psi(\mu_1, \mu_2)$$

*for any probability measures $\mu_1, \mu_2$ on $\mathbb{R}^d$. In particular, for any $\eta_0 \in (0, \hat{\eta})$ and $\gamma_0 \in (\gamma + 2K/R, 1)$ one can choose $\psi = \eta_0/K$ and $\bar{\eta} = (1 - (\hat{\eta} - \eta_0)) \vee (2 + R\psi\gamma_0)/(2 + R\psi)$.*

## C  Proofs of Wasserstein Perturbation Results: Quadratic Case

### C.1  Proof of Lemma 3.1

*Proof.* Let $P^k(\theta, \cdot)$ denote the law of $\theta_k$ starting with $\theta_0 = \theta$ and $P^k(\tilde{\theta}, \cdot)$ the law of $\tilde{\theta}_k$:

$$\tilde{\theta}_k = \left(I - \frac{\eta}{b} H_k\right)\tilde{\theta}_{k-1} + \frac{\eta}{b} q_k, \tag{C.1}$$

with $\tilde{\theta}_0 = \tilde{\theta}$. Note that

$$\theta_k = \left(I - \frac{\eta}{b} H_k\right)\theta_{k-1} + \frac{\eta}{b} q_k, \tag{C.2}$$

$$\tilde{\theta}_k = \left(I - \frac{\eta}{b} H_k\right)\tilde{\theta}_{k-1} + \frac{\eta}{b} q_k, \tag{C.3}$$

which implies that

$$\mathbb{E}\left\|\theta_k - \tilde{\theta}_k\right\| = \mathbb{E}\left\|\left(I - \frac{\eta}{b} H_k\right)\left(\theta_{k-1} - \tilde{\theta}_{k-1}\right)\right\|$$
$$\leq \mathbb{E}\left[\left\|I - \frac{\eta}{b} H_k\right\| \left\|\theta_{k-1} - \tilde{\theta}_{k-1}\right\|\right] = \rho \mathbb{E}\left\|\theta_{k-1} - \tilde{\theta}_{k-1}\right\|. \tag{C.4}$$

By iterating over $j = k, k-1, \ldots 1$, we conclude that

$$\mathcal{W}_1\left(P^k(\theta, \cdot), P^k(\tilde{\theta}, \cdot)\right) \leq \mathbb{E}\|\theta_k - \tilde{\theta}_k\| \leq \rho^n \|\theta_0 - \tilde{\theta}_0\| = \rho^k \|\theta - \tilde{\theta}\|. \tag{C.5}$$

This completes the proof. $\qquad\square$

### C.2  Proof of Lemma 3.2

*Proof.* First, we recall that

$$\hat{\theta}_k = \left(I - \frac{\eta}{b} \hat{H}_k\right)\hat{\theta}_{k-1} + \frac{\eta}{b} \hat{q}_k, \tag{C.6}$$

where $\hat{H}_k := \sum_{i \in \Omega_k} \hat{a}_i \hat{a}_i^\top$ and $\hat{q}_k := \sum_{i \in \Omega_k} \hat{a}_i \hat{y}_i$. Therefore, starting with $\hat{\theta}_0 = \theta$, we have

$$\hat{\theta}_1 = \left(I - \frac{\eta}{b} \hat{H}_1\right)\theta + \frac{\eta}{b} \hat{q}_1, \tag{C.7}$$

which implies that

$$(\hat{P}\hat{V})(\theta) = \mathbb{E}\hat{V}(\hat{\theta}_1) = 1 + \mathbb{E}\|\hat{\theta}_1\| \leq 1 + \hat{\rho}\|\theta\| + \frac{\eta}{b}\mathbb{E}\|\hat{q}_1\| = \hat{\rho}\hat{V}(\theta) + 1 - \hat{\rho} + \frac{\eta}{b}\mathbb{E}\|\hat{q}_1\|. \tag{C.8}$$

This completes the proof. $\qquad\square$

## C.3 Proof of Lemma 3.3

*Proof.* Let us recall that

$$\theta_1 = \left(I - \frac{\eta}{b}H_1\right)\theta + \frac{\eta}{b}q_1, \tag{C.9}$$

$$\hat{\theta}_1 = \left(I - \frac{\eta}{b}\hat{H}_1\right)\theta + \frac{\eta}{b}\hat{q}_1, \tag{C.10}$$

which implies that

$$\mathcal{W}_1\left(\delta_\theta P, \delta_\theta \hat{P}\right) \leq \mathbb{E}\left\|H_1 - \hat{H}_1\right\| \frac{\eta}{b}\|\theta\| + \frac{\eta}{b}\mathbb{E}\|q_1 - \hat{q}_1\|. \tag{C.11}$$

Since $X_n$ and $\hat{X}_n$ differ by at most one element and $\sup_{x \in \mathcal{X}}\|x\| \leq D$ for some $D < \infty$, we have $(H_1, q_1) = (\hat{H}_1, q_1)$ with probability $\frac{n-b}{n}$ and $(H_1, q_1) \neq (\hat{H}_1, q_1)$ with probability $\frac{b}{n}$ and moreover

$$\mathbb{E}\left\|H_1 - \hat{H}_1\right\| \leq \frac{b}{n}\max_{1\leq i\leq n}\left\|a_i a_i^\top - \hat{a}_i \hat{a}_i^\top\right\| \leq \frac{b}{n}\max_{1\leq i\leq n}\left(\|a_i\|^2 + \|\hat{a}_i\|^2\right) \leq \frac{2bD^2}{n}, \tag{C.12}$$

and

$$\mathbb{E}\|q_1 - \hat{q}_1\| \leq \frac{b}{n}\max_{1\leq i\leq n}\|a_i y_i - \hat{a}_i \hat{y}_i\| \leq \frac{b}{n}\max_{1\leq i\leq n}\left(\|a_i\|\|q_i\| + \|\hat{a}_i\|\|\hat{q}_i\|\right) \leq \frac{2bD^2}{n}. \tag{C.13}$$

Hence, we conclude that

$$\sup_{\theta \in \mathbb{R}^d} \frac{\mathcal{W}_1(\delta_\theta P, \delta_\theta \hat{P})}{\hat{V}(\theta)} \leq \sup_{\theta \in \mathbb{R}^d} \frac{\frac{\eta}{b}(\|\theta\| + 1)\frac{2bD^2}{n}}{1 + \|\theta\|} = \frac{2\eta D^2}{n}. \tag{C.14}$$

This completes the proof. $\qquad\square$

## D  Proofs of Wasserstein Perturbation Results: Strongly Convex Case

In order to obtain the algorithmic stability bound, that is a 1-Wasserstein distance between the distribution of $\theta_k$ and $\hat{\theta}_k$, we need to establish a sequence of technical lemmas. First, we show a 1-Wasserstein contraction rate in the following lemma.

**Lemma D.1.** *Assume that Assumption 3.1 and Assumption 3.2 hold, and further assume that* $\eta < \min\left\{\frac{1}{\mu}, \frac{\mu}{K_1^2}\right\}$. *Then, for any* $n \in \mathbb{N}$,

$$\mathcal{W}_1\left(P^n(\theta, \cdot), P^n\left(\tilde{\theta}, \cdot\right)\right) \leq \left(1 - \frac{\eta\mu}{2}\right)^n \|\theta - \tilde{\theta}\|. \tag{D.1}$$

*Proof.* Let $P^k(\theta, \cdot)$ denote the law of $\theta_k$ starting with $\theta_0 = \theta$:

$$\theta_k = \theta_{k-1} - \eta\tilde{\nabla}\hat{F}_k(\theta_{k-1}, X_n), \tag{D.2}$$

and $P^k(\tilde{\theta}, \cdot)$ the law of $\tilde{\theta}_k$:

$$\tilde{\theta}_k = \tilde{\theta}_{k-1} - \eta\tilde{\nabla}\hat{F}_k\left(\tilde{\theta}_{k-1}, X_n\right), \tag{D.3}$$

with $\tilde{\theta}_0 = \tilde{\theta}$. Note that

$$\theta_k = \theta_{k-1} - \frac{\eta}{b}\sum_{i\in\Omega_k}\nabla f(\theta_{k-1}, x_i), \tag{D.4}$$

$$\tilde{\theta}_k = \tilde{\theta}_{k-1} - \frac{\eta}{b}\sum_{i\in\Omega_k}\nabla f\left(\tilde{\theta}_{k-1}, x_i\right). \tag{D.5}$$

Therefore, we have

$$\mathbb{E}\left\|\theta_k - \tilde{\theta}_k\right\|^2 = \mathbb{E}\left\|\theta_{k-1} - \tilde{\theta}_{k-1} - \frac{\eta}{b}\sum_{i\in\Omega_k}\left(\nabla f(\theta_{k-1}, x_i) - \nabla f\left(\tilde{\theta}_{k-1}, x_i\right)\right)\right\|^2$$

$$= \mathbb{E}\left\|\theta_{k-1} - \tilde{\theta}_{k-1}\right\|^2 + \frac{\eta^2}{b^2}\mathbb{E}\left\|\sum_{i\in\Omega_k}\left(\nabla f(\theta_{k-1}, x_i) - \nabla f\left(\tilde{\theta}_{k-1}, x_i\right)\right)\right\|^2$$

$$- \frac{2\eta}{b}\mathbb{E}\left\langle\theta_{k-1} - \tilde{\theta}_{k-1}, \sum_{i\in\Omega_k}\left(\nabla f(\theta_{k-1}, x_i) - \nabla f\left(\tilde{\theta}_{k-1}, x_i\right)\right)\right\rangle. \tag{D.6}$$

By applying Assumption 3.1 and Assumption 3.2, we get

$$
\mathbb{E}\left\|\theta_k - \tilde{\theta}_k\right\|^2
$$

$$
\leq (1 - 2\eta\mu)\mathbb{E}\left\|\theta_{k-1} - \tilde{\theta}_{k-1}\right\|^2 + \frac{\eta^2}{b^2}\mathbb{E}\left[\left(\sum_{i \in \Omega_k}\left\|\nabla f(\theta_{k-1}, x_i) - \nabla f\left(\tilde{\theta}_{k-1}, x_i\right)\right\|\right)^2\right]
$$

$$
\leq (1 - 2\eta\mu)\mathbb{E}\left\|\theta_{k-1} - \tilde{\theta}_{k-1}\right\|^2 + \eta^2 K_1^2 \mathbb{E}\left\|\theta_{k-1} - \tilde{\theta}_{k-1}\right\|^2
$$

$$
\leq (1 - \eta\mu)\mathbb{E}\left\|\theta_{k-1} - \tilde{\theta}_{k-1}\right\|^2, \tag{D.7}
$$

provided that $\eta \leq \frac{\mu}{K_1^2}$. By iterating over $k = n, n-1, \dots 1$, we conclude that

$$
\left(\mathcal{W}_1\left(P^n(\theta, \cdot), P^n(\tilde{\theta}, \cdot)\right)\right)^2 \leq \left(\mathcal{W}_2\left(P^n(\theta, \cdot), P^n(\tilde{\theta}, \cdot)\right)\right)^2
$$

$$
\leq \mathbb{E}\left\|\theta_n - \tilde{\theta}_n\right\|^2 \leq (1 - \eta\mu)^n\left\|\theta_0 - \tilde{\theta}_0\right\|^2 \leq \left(1 - \frac{\eta\mu}{2}\right)^{2n}\left\|\theta - \tilde{\theta}\right\|^2. \tag{D.8}
$$

This completes the proof. $\qquad\square$

Next, we construct a Lyapunov function $\hat{V}$ and obtain a drift condition for the SGD $(\hat{\theta}_k)_{k=0}^\infty$.

**Lemma D.2.** *Assume that Assumption 3.1 and Assumption 3.2 hold. Let $\hat{V}(\theta) := 1 + \|\theta - \hat{\theta}_*\|^2$, where $\hat{\theta}_*$ is the minimizer of $\hat{F}(\theta, \hat{X}_n) := \frac{1}{n}\sum_{i=1}^n \nabla f(\theta, \hat{x}_i)$. Assume that $\eta < \min\left\{\frac{1}{\mu}, \frac{\mu}{K_1^2 + 64D^2 K_2^2}\right\}$ and $\sup_{x \in \mathcal{X}}\|x\| \leq D$ for some $D < \infty$. Then, we have*

$$
(\hat{P}\hat{V})(\theta) \leq (1 - \eta\mu)\hat{V}(\theta) + 2\eta\mu - \eta^2 K_1^2 - 56\eta^2 D^2 K_2^2 + 64\eta^2 D^2 K_2^2\|\hat{\theta}_*\|^2. \tag{D.9}
$$

*Proof.* First, we recall that

$$
\hat{\theta}_k = \hat{\theta}_{k-1} - \frac{\eta}{b}\sum_{i \in \Omega_k}\nabla f\left(\hat{\theta}_{k-1}, \hat{x}_i\right). \tag{D.10}
$$

Therefore, starting with $\hat{\theta}_0 = \theta$, we have

$$
\hat{\theta}_1 = \theta - \frac{\eta}{b}\sum_{i \in \Omega_1}\nabla f(\theta, \hat{x}_i)
$$

$$
= \theta - \frac{\eta}{n}\sum_{i=1}^n \nabla f(\theta, \hat{x}_i) + \eta\left(\frac{1}{n}\sum_{i=1}^n \nabla f(\theta, \hat{x}_i) - \frac{1}{b}\sum_{i \in \Omega_1}\nabla f(\theta, \hat{x}_i)\right). \tag{D.11}
$$

Moreover, we have

$$
\mathbb{E}\left[\frac{1}{b}\sum_{i \in \Omega_1}\nabla f(\theta, \hat{x}_i)\Big|\theta\right] = \frac{1}{n}\sum_{i=1}^n \nabla f(\theta, \hat{x}_i). \tag{D.12}
$$

This implies that

$$
(\hat{P}\hat{V})(\theta)
$$

$$
= \mathbb{E}\hat{V}(\hat{\theta}_1) = 1 + \mathbb{E}\left\|\hat{\theta}_1 - \hat{\theta}_*\right\|^2
$$

$$
= 1 + \left\|\theta - \hat{\theta}_* + \frac{\eta}{n}\sum_{i=1}^n \nabla f(\theta, \hat{x}_i)\right\|^2 + \eta^2\mathbb{E}\left\|\frac{1}{n}\sum_{i=1}^n \nabla f(\theta, \hat{x}_i) - \frac{1}{b}\sum_{i \in \Omega_1}\nabla f(\theta, \hat{x}_i)\right\|^2. \tag{D.13}
$$

We can compute that

$$
\left\| \theta - \hat{\theta}_* + \frac{\eta}{n} \sum_{i=1}^{n} \nabla f(\theta, \hat{x}_i) \right\|^2
$$

$$
= \left\| \theta - \hat{\theta}_* + \frac{\eta}{n} \sum_{i=1}^{n} \left( \nabla f(\theta, \hat{x}_i) - \nabla f\left(\hat{\theta}_*, \hat{x}_i\right) \right) \right\|^2
$$

$$
\leq (1 - 2\eta\mu)\|\theta - \hat{\theta}_*\|^2 + \frac{\eta^2}{n^2} \left\| \sum_{i=1}^{n} \left( \nabla f(\theta, \hat{x}_i) - \nabla f\left(\hat{\theta}_*, \hat{x}_i\right) \right) \right\|^2
$$

$$
\leq (1 - 2\eta\mu + \eta^2 K_1^2)\|\theta - \hat{\theta}_*\|^2. \tag{D.14}
$$

Moreover, we can compute that

$$
\mathbb{E} \left\| \frac{1}{n} \sum_{i=1}^{n} \nabla f(\theta, \hat{x}_i) - \frac{1}{b} \sum_{i \in \Omega_1} \nabla f(\theta, \hat{x}_i) \right\|^2 = \mathbb{E} \left\| \frac{1}{b} \sum_{i \in \Omega_1} \left( \frac{1}{n} \sum_{j=1}^{n} \nabla f(\theta, \hat{x}_j) - \nabla f(\theta, \hat{x}_i) \right) \right\|^2
$$

$$
= \mathbb{E} \left( \frac{1}{b} \sum_{i \in \Omega_1} K_2 \frac{1}{n} \sum_{j=1}^{n} \|\hat{x}_i - \hat{x}_j\|(2\|\theta\| + 1) \right)^2
$$

$$
\leq 4D^2 K_2^2 (2\|\theta\| + 1)^2
$$

$$
\leq 8D^2 K_2^2 (4\|\theta\|^2 + 1)
$$

$$
\leq 8D^2 K_2^2 \left( 8\|\theta - \hat{\theta}_*\|^2 + 8\|\hat{\theta}_*\|^2 + 1 \right). \tag{D.15}
$$

Hence, we conclude that

$$
(\hat{P}\hat{V})(\theta) \leq \left( 1 - 2\eta\mu + \eta^2 K_1^2 + 64\eta^2 D^2 K_2^2 \right) \left\| \theta - \hat{\theta}_* \right\|^2 + 1 + 8\eta^2 D^2 K_2^2 \left( 8\|\hat{\theta}_*\|^2 + 1 \right)
$$

$$
= \left( 1 - 2\eta\mu + \eta^2 K_1^2 + 64\eta^2 D^2 K_2^2 \right) \hat{V}(\theta)
$$

$$
+ 2\eta\mu - \eta^2 K_1^2 - 56\eta^2 D^2 K_2^2 + 64\eta^2 D^2 K_2^2 \|\hat{\theta}_*\|^2
$$

$$
\leq (1 - \eta\mu)\hat{V}(\theta) + 2\eta\mu - \eta^2 K_1^2 - 56\eta^2 D^2 K_2^2 + 64\eta^2 D^2 K_2^2 \|\hat{\theta}_*\|^2, \tag{D.16}
$$

provided that $\eta \leq \frac{\mu}{K_1^2 + 64D^2 K_2^2}$. This completes the proof. $\qquad \square$

Next, we estimate the perturbation gap based on the Lyapunov function $\hat{V}$.

**Lemma D.3.** *Assume that Assumption 3.1 holds. Assume that $\sup_{x \in \mathcal{X}} \|x\| \leq D$ for some $D < \infty$. Then, we have*

$$
\sup_{\theta \in \mathbb{R}^d} \frac{\mathcal{W}_1(\delta_\theta P, \delta_\theta \hat{P})}{\hat{V}(\theta)} \leq \frac{4DK_2\eta}{n}(2\|\hat{\theta}_*\| + 1), \tag{D.17}
$$

*where $\hat{\theta}_*$ is the minimizer of $\hat{F}(\theta, \hat{X}_n) := \frac{1}{n} \sum_{i=1}^{n} \nabla f(\theta, \hat{x}_i)$.*

*Proof.* Let us recall that

$$
\theta_1 = \theta - \frac{\eta}{b} \sum_{i \in \Omega_1} \nabla f(\theta, x_i), \tag{D.18}
$$

$$
\hat{\theta}_1 = \theta - \frac{\eta}{b} \sum_{i \in \Omega_1} \nabla f(\theta, \hat{x}_i), \tag{D.19}
$$

which implies that

$$\mathcal{W}_1\left(\delta_\theta P, \delta_\theta \hat{P}\right) \leq \frac{\eta}{b}\mathbb{E}\left\|\sum_{i\in\Omega_1}\nabla f(\theta, x_i) - \nabla f(\theta, \hat{x}_i)\right\|$$

$$\leq \frac{\eta}{b}\mathbb{E}\left[\sum_{i\in\Omega_1}\|\nabla f(\theta, x_i) - \nabla f(\theta, \hat{x}_i)\|\right]$$

$$\leq \frac{\eta}{b}\mathbb{E}\left[\sum_{i\in\Omega_1}K_2\|x_i - \hat{x}_i\|\,(2\|\theta\|+1)\right] \qquad (D.20)$$

Since $X_n$ and $\hat{X}_n$ differ by at most one element and $\sup_{x\in\mathcal{X}}\|x\| \leq D$ for some $D < \infty$, we have $x_i = \hat{x}_i$ for any $i \in \Omega_1$ with probability $\frac{n-b}{n}$ and $x_i \neq \hat{x}_i$ for exactly one $i \in \Omega_1$ with probability $\frac{b}{n}$ and therefore

$$\mathcal{W}_1\left(\delta_\theta P, \delta_\theta \hat{P}\right) \leq \frac{\eta}{b}\frac{b}{n}2K_2 D(2\|\theta\|+1) = \frac{2DK_2\eta}{n}(2\|\theta\|+1). \qquad (D.21)$$

Hence, we conclude that

$$\sup_{\theta\in\mathbb{R}^d}\frac{\mathcal{W}_1(\delta_\theta P, \delta_\theta \hat{P})}{\hat{V}(\theta)} \leq \sup_{\theta\in\mathbb{R}^d}\frac{4DK_2\eta}{n}\frac{\|\theta\|+\frac{1}{2}}{1+\|\theta-\hat{\theta}_*\|^2}$$

$$\leq \sup_{\theta\in\mathbb{R}^d}\frac{4DK_2\eta}{n}\frac{2\|\theta-\hat{\theta}_*\|+2\|\hat{\theta}_*\|+1}{1+\|\theta-\hat{\theta}_*\|^2}$$

$$\leq \sup_{\theta\in\mathbb{R}^d}\frac{4DK_2\eta}{n}\frac{2\|\theta-\hat{\theta}_*\|(2\|\hat{\theta}_*\|+1)}{1+\|\theta-\hat{\theta}_*\|^2}$$

$$\leq \frac{4DK_2\eta}{n}\left(2\|\hat{\theta}_*\|+1\right). \qquad (D.22)$$

This completes the proof. $\qquad\square$

Next, let us provide a technical lemma that upper bounds the norm of $\theta_*$ and $\hat{\theta}_*$, which are the minimizers of $\hat{F}(\theta, X_n) := \frac{1}{n}\sum_{i=1}^n \nabla f(\theta, x_i)$ and $\hat{F}(\theta, \hat{X}_n) := \frac{1}{n}\sum_{i=1}^n \nabla f(\theta, \hat{x}_i)$ respectively.

**Lemma D.4.** *Under Assumption 3.2, we have* $\|\theta_*\| \leq \frac{1}{\mu}\sup_{x\in\mathcal{X}}\|\nabla f(0, x)\|$ *and* $\|\hat{\theta}_*\| \leq \frac{1}{\mu}\sup_{x\in\mathcal{X}}\|\nabla f(0, x)\|$.

*Proof.* Since $f(\theta, x)$ is $\mu$-strongly convex in $\theta$ for every $x \in \mathcal{X}$, we have

$$\left\langle \nabla \hat{F}(0, X_n) - \nabla \hat{F}(\theta_*, X_n), 0 - \theta_* \right\rangle = -\frac{1}{n}\sum_{i=1}^n \langle \nabla f(0, x_i), \theta_* \rangle$$

$$= \frac{1}{n}\sum_{i=1}^n \langle \nabla f(0, x_i) - \nabla f(\theta_*, x_i), 0 - \theta_* \rangle \geq \mu\|\theta_*\|^2, \qquad (D.23)$$

which implies that

$$\mu\|\theta_*\|^2 \leq \sup_{x\in\mathcal{X}}\|\nabla f(0, x)\| \cdot \|\theta_*\|, \qquad (D.24)$$

which yields that $\|\theta_*\| \leq \frac{1}{\mu}\sup_{x\in\mathcal{X}}\|\nabla f(0, x)\|$. Similarly, one can show that $\|\hat{\theta}_*\| \leq \frac{1}{\mu}\sup_{x\in\mathcal{X}}\|\nabla f(0, x)\|$. This completes the proof. $\qquad\square$

### D.1 Proof of Theorem 3.2

*Proof.* By applying Lemma D.1, Lemma D.2, Lemma D.3 and Lemma 2.1, we obtain

$$\mathcal{W}_1(\nu_k, \hat{\nu}_k) \leq \frac{8DK_2(1 - (1 - \frac{\eta\mu}{2})^k)}{n\mu}\left(2\|\hat{\theta}_*\|+1\right)$$

$$\cdot \max\left\{1 + \left\|\theta - \hat{\theta}_*\right\|^2, 2 - \frac{\eta}{\mu}K_1^2 - \frac{56\eta}{\mu}D^2K_2^2 + \frac{64\eta}{\mu}D^2K_2^2\|\hat{\theta}_*\|^2\right\}, \qquad (D.25)$$

where $\hat{\theta}_*$ is the minimizer of $\hat{F}(\theta, \hat{X}_n) := \frac{1}{n} \sum_{i=1}^{n} \nabla f(\theta, \hat{x}_i)$. Finally, $\|\theta - \hat{\theta}_*\|^2 \leq 2\|\theta\|^2 + 2\|\hat{\theta}_*\|^2$ and by applying Lemma D.4, we complete the proof. $\qquad\square$

# E    Proofs of Wasserstein Perturbation Bounds: Non-Convex Case

**Lemma E.1.** *Assume that Assumption 3.1, Assumption 3.3 and Assumption 3.4 hold. For any* $\hat{\eta} \in (0, 1)$*. Define* $M > 0$ *so that* $\int_{\|\theta_1\| \leq M} p(\theta_*, \theta_1) d\theta_1 \geq \sqrt{\hat{\eta}}$ *and any* $R > 0$ *so that* $R > \frac{2K_0}{m}$ *and*

$$\inf_{\theta, \theta_1 \in \mathbb{R}^d : V(\theta) \leq R, \|\theta_1\| \leq M} \frac{p(\theta, \theta_1)}{p(\theta_*, \theta_1)} \geq \sqrt{\hat{\eta}}, \tag{E.1}$$

*where*

$$K_0 := 2m - \eta K_1^2 - 56\eta D^2 K_2^2 + 64\eta D^2 K_2^2 \|\theta_*\|^2 + 2K + \eta \sigma^2. \tag{E.2}$$

*Then, for any* $n \in \mathbb{N}$,

$$\mathcal{W}_1\left(P^n(\theta, \cdot), P^n(\tilde{\theta}, \cdot)\right) \leq \frac{1}{2\sqrt{\psi(1+\psi)}} \bar{\eta}^n d_\psi(\delta_\theta, \delta_{\tilde{\theta}}), \tag{E.3}$$

*for any* $\theta, \tilde{\theta}$ *in* $\mathbb{R}^d$, *where for any* $\eta_0 \in (0, \hat{\eta})$ *and* $\gamma_0 \in \left(1 - m\eta + \frac{2\eta K0}{R}, 1\right)$ *one can choose* $\psi = \frac{\eta_0}{\eta K_0}$ *and* $\bar{\eta} = (1 - (\hat{\eta} - \eta_0)) \vee \frac{2 + R\psi\gamma_0}{2 + R\psi}$ *and* $d_\psi$ *is the weighted total variation distance defined in Section B.1.*

*Proof.* Our proof relies on a computable bound on the Wasserstein contraction for the Markov chains by [HM11] that satisfies a drift condition (Assumption B.1) that relies on the construction of an appropriate Lyapunov function and a minorization condition (Assumption B.2).

By applying Lemma E.3, we can immediately show that the following drift condition holds. Let $V(\theta) := 1 + \|\theta - \theta_*\|^2$, where $\theta_*$ is the minimizer of $\frac{1}{n} \sum_{i=1}^{n} \nabla f(\theta, x_i)$. Assume that $\eta < \min\left\{\frac{1}{m}, \frac{m}{K_1 + 64D^2 K_2^2}\right\}$ and $\sup_{x \in \mathcal{X}} \|x\| \leq D$ for some $D < \infty$. Then, we have

$$(PV)(\theta) \leq (1 - m\eta)V(\theta) + \eta K_0, \tag{E.4}$$

where

$$K_0 := 2m - \eta K_1^2 - 56\eta D^2 K_2^2 + 64\eta D^2 K_2^2 \|\theta_*\|^2 + 2K + \eta \sigma^2. \tag{E.5}$$

Thus, the drift condition (Assumption B.1) holds.

Next, let us show that the minorization condition (Assumption B.2) also holds. Let us recall that

$$\theta_1 = \theta - \frac{\eta}{b} \sum_{i \in \Omega_1} \nabla f(\theta, x_i) + \eta \xi_1. \tag{E.6}$$

We denote $p(\theta, \theta_1)$ the probability density function of $\theta_1$ with the emphasis on the dependence on the initial point $\theta$. Then, to check that the minorization condition (Assumption B.2) holds, it suffices to show that there exists some constant $\hat{\eta} \in (0, 1)$

$$\inf_{\theta \in \mathbb{R}^d : V(\theta) \leq R} p(\theta, \theta_1) \geq \hat{\eta} q(\theta_1), \qquad \text{for any } \theta_1 \in \mathbb{R}^d, \tag{E.7}$$

for some $R > \frac{2\eta K_0}{1 - (1 - m\eta)} = \frac{2K_0}{m}$, where $q(\theta_1)$ is the density function of a probability distribution function on $\mathbb{R}^d$, and (E.7) follows from Lemma E.2. Hence, by Lemma B.1, we have

$$d_\psi\left(P^n(\theta, \cdot), P^n(\tilde{\theta}, \cdot)\right) \leq \bar{\eta}^n d_\psi(\delta_\theta, \delta_{\tilde{\theta}}),$$

for any $\theta, \tilde{\theta}$ in $\mathbb{R}^d$, where for any $\eta_0 \in (0, \hat{\eta})$ and $\gamma_0 \in \left(1 - m\eta + \frac{2\eta K0}{R}, 1\right)$ one can choose $\psi = \frac{\eta_0}{\eta K_0}$ and $\bar{\eta} = (1 - (\hat{\eta} - \eta_0)) \vee \frac{2 + R\psi\gamma_0}{2 + R\psi}$.

Finally, by the Kantorovich-Rubinstein duality for the Wasserstein metric, we get for any two probability measures $\mu_1, \mu_2$ on $\mathbb{R}^d$:

$$
\begin{aligned}
\mathcal{W}_1(\mu_1, \mu_2) &= \sup\left\{ \int_{\mathbb{R}^d} \phi(\theta)(\mu_1 - \mu_2)(d\theta) : \phi \text{ is 1-Lipschitz} \right\} \\
&= \sup\left\{ \int_{\mathbb{R}^d} (\phi(\theta) - \phi(\theta_*))(\mu_1 - \mu_2)(d\theta) : \phi \text{ is 1-Lipschitz} \right\} \\
&\leq \int_{\mathbb{R}^d} \|\theta - \theta_*\| |\mu_1 - \mu_2|(d\theta) \\
&\leq \frac{1}{2\sqrt{\psi(1+\psi)}} \int_{\mathbb{R}^d} (1 + \psi V(\theta)) |\mu_1 - \mu_2|(d\theta) \\
&= \frac{1}{2\sqrt{\psi(1+\psi)}} d_\psi(\mu_1, \mu_2).
\end{aligned} \tag{E.8}
$$

Hence, we conclude that

$$
\mathcal{W}_1\left( P^n(\theta, \cdot), P^n(\tilde{\theta}, \cdot) \right) \leq \frac{1}{2\sqrt{\psi(1+\psi)}} \bar{\eta}^n d_\psi(\delta_\theta, \delta_{\tilde{\theta}}).
$$

This completes the proof. $\qquad\square$

The proof of Lemma E.1 relies on the following technical lemma, which is a reformulation of Lemma 35 in [CGZ19] that helps establish the minorization condition (Assumption B.2).

**Lemma E.2.** *For any $\hat{\eta} \in (0,1)$ and $M > 0$ so that $\int_{\|\theta_1 - \theta_*\| \leq M} p(\theta_*, \theta_1) d\theta_1 \geq \sqrt{\hat{\eta}}$ and any $R > 0$ so that*

$$
\inf_{\theta, \theta_1 \in \mathbb{R}^d : V(\theta) \leq R, \|\theta_1 - \theta_*\| \leq M} \frac{p(\theta, \theta_1)}{p(\theta_*, \theta_1)} \geq \sqrt{\hat{\eta}}. \tag{E.9}
$$

*Then, we have*

$$
\inf_{\theta \in \mathbb{R}^d : V(\theta) \leq R} p(\theta, \theta_1) \geq \hat{\eta} q(\theta_1), \qquad \text{for any } \theta_1 \in \mathbb{R}^d, \tag{E.10}
$$

*where*

$$
q(\theta_1) = p(\theta_*, \theta_1) \cdot \frac{1_{\|\theta_1 - \theta_*\| \leq M}}{\int_{\|\theta_1 - \theta_*\| \leq M} p(\theta_*, \theta_1) d\theta_1}. \tag{E.11}
$$

*Proof.* The proof is an adaptation of the proof of Lemma 35 in [CGZ19]. Let us take:

$$
q(\theta_1) = p(\theta_*, \theta_1) \cdot \frac{1_{\|\theta_1 - \theta_*\| \leq M}}{\int_{\|\theta_1 - \theta_*\| \leq M} p(\theta_*, \theta_1) d\theta_1}. \tag{E.12}
$$

Then, it is clear that $q(\theta_1)$ is a probability density function on $\mathbb{R}^d$. It follows that (E.7) automatically holds for $\|\theta_1 - \theta_*\| > M$. Thus, we only need to show that (E.7) holds for $\|\theta_1 - \theta_*\| \leq M$. Since $\xi_1$ has a continuous density, $p(\theta, \theta_1)$ is continuous in both $\theta$ and $\theta_1$. Fix $M$, by continuity of $p(\theta, \theta_1)$ in both $\theta$ and $\theta_1$, there exists some $\eta' \in (0,1)$ such that uniformly in $\|\theta_1 - \theta_*\| \leq M$,

$$
\inf_{\theta \in \mathbb{R}^d : V(\theta) \leq R} p(\theta, \theta_1) \geq \eta' p(\theta_*, \theta_1) = \hat{\eta} q(\theta_1), \tag{E.13}
$$

where we can take

$$
\hat{\eta} := \eta' \int_{\|\theta_1 - \theta_*\| \leq M} p(\theta_*, \theta_1) d\theta_1. \tag{E.14}
$$

In particular, for any fixed $\hat{\eta}$, we can take $M > 0$ such that

$$
\int_{\|\theta_1 - \theta_*\| \leq M} p(\theta_*, \theta_1) d\theta_1 \geq \sqrt{\hat{\eta}}, \tag{E.15}
$$

and with fixed $\eta$ and $M$, we take $R > 0$ such that uniformly in $\|\theta_1 - \theta_*\| \leq M$,

$$
\inf_{\theta \in \mathbb{R}^d : V(\theta) \leq R} p(\theta, \theta_1) \geq \sqrt{\hat{\eta}} p(\theta_*, \theta_1). \tag{E.16}
$$

This completes the proof. $\qquad\square$

Next, we construct a Lyapunov function $\hat{V}$ and obtain a drift condition for the SGD $(\hat{\theta}_k)_{k=0}^\infty$.

**Lemma E.3.** *Assume that Assumption 3.1, Assumption 3.3 and Assumption 3.4 hold. Let $\hat{V}(\theta) := 1 + \|\theta - \hat{\theta}_*\|^2$, where $\hat{\theta}_*$ is the minimizer of $\hat{F}(\theta, \hat{X}_n) := \frac{1}{n} \sum_{i=1}^n \nabla f(\theta, \hat{x}_i)$. Assume that $\eta < \min\left\{ \frac{1}{m}, \frac{m}{K_1^2 + 64 D^2 K_2^2} \right\}$ and $\sup_{x \in \mathcal{X}} \|x\| \leq D$ for some $D < \infty$. Then, we have*

$$(\hat{P}\hat{V})(\theta) \leq (1 - m\eta)\hat{V}(\theta) + 2m\eta - \eta^2 K_1^2 - 56\eta^2 D^2 K_2^2 + 64\eta^2 D^2 K_2^2 \|\hat{\theta}_*\|^2 + 2\eta K + \eta^2 \sigma^2. \quad \text{(E.17)}$$

*Proof.* First, we recall that

$$\hat{\theta}_k = \hat{\theta}_{k-1} - \frac{\eta}{b} \sum_{i \in \Omega_k} \nabla f\left(\hat{\theta}_{k-1}, \hat{x}_i\right) + \eta \xi_k. \quad \text{(E.18)}$$

Therefore, starting with $\hat{\theta}_0 = \theta$, we have

$$\hat{\theta}_1 = \theta - \frac{\eta}{b} \sum_{i \in \Omega_1} \nabla f(\theta, \hat{x}_i) + \eta \xi_1$$

$$= \theta - \frac{\eta}{n} \sum_{i=1}^n \nabla f(\theta, \hat{x}_i) + \eta \left( \frac{1}{n} \sum_{i=1}^n \nabla f(\theta, \hat{x}_i) - \frac{1}{b} \sum_{i \in \Omega_1} \nabla f(\theta, \hat{x}_i) \right) + \eta \xi_1. \quad \text{(E.19)}$$

Moreover, we have

$$\mathbb{E}\left[ \frac{1}{b} \sum_{i \in \Omega_1} \nabla f(\theta, \hat{x}_i) \Big| \theta \right] = \frac{1}{n} \sum_{i=1}^n \nabla f(\theta, \hat{x}_i). \quad \text{(E.20)}$$

This implies that

$$(\hat{P}\hat{V})(\theta)$$

$$= \mathbb{E}\hat{V}(\hat{\theta}_1) = 1 + \mathbb{E}\left\| \hat{\theta}_1 - \hat{\theta}_* \right\|^2$$

$$= 1 + \left\| \theta - \hat{\theta}_* - \frac{\eta}{n} \sum_{i=1}^n \nabla f(\theta, \hat{x}_i) \right\|^2 + \eta^2 \mathbb{E}\left\| \frac{1}{n} \sum_{i=1}^n \nabla f(\theta, \hat{x}_i) - \frac{1}{b} \sum_{i \in \Omega_1} \nabla f(\theta, \hat{x}_i) \right\|^2 + \eta^2 \sigma^2. \quad \text{(E.21)}$$

We can compute that

$$\left\| \theta - \hat{\theta}_* - \frac{\eta}{n} \sum_{i=1}^n \nabla f(\theta, \hat{x}_i) \right\|^2$$

$$= \left\| \theta - \hat{\theta}_* - \frac{\eta}{n} \sum_{i=1}^n \left( \nabla f(\theta, \hat{x}_i) - \nabla f(\hat{\theta}_*, \hat{x}_i) \right) \right\|^2$$

$$\leq (1 - 2m\eta) \left\| \theta - \hat{\theta}_* \right\|^2 + 2\eta K + \frac{\eta^2}{n^2} \left\| \sum_{i=1}^n \left( \nabla f(\theta, \hat{x}_i) - \nabla f(\hat{\theta}_*, \hat{x}_i) \right) \right\|^2$$

$$\leq \left( 1 - 2m\eta + \eta^2 K_1^2 \right) \left\| \theta - \hat{\theta}_* \right\|^2 + 2\eta K. \quad \text{(E.22)}$$

Moreover, by following the same arguments as in the proof of Lemma D.2, we have

$$\mathbb{E}\left\| \frac{1}{n} \sum_{i=1}^n \nabla f(\theta, \hat{x}_i) - \frac{1}{b} \sum_{i \in \Omega_1} \nabla f(\theta, \hat{x}_i) \right\|^2 \leq 8 D^2 K_2^2 \left( 8 \left\| \theta - \hat{\theta}_* \right\|^2 + 8 \|\hat{\theta}_*\|^2 + 1 \right). \quad \text{(E.23)}$$

Hence, we conclude that

$$
\begin{aligned}
(\hat{P}\hat{V})(\theta) &\leq \left(1 - 2m\eta + \eta^2 K_1^2 + 64\eta^2 D^2 K_2^2\right) \left\|\theta - \hat{\theta}_*\right\|^2 \\
&\quad + 1 + 8\eta^2 D^2 K_2^2 \left(8\|\hat{\theta}_*\|^2 + 1\right) + 2\eta K + \eta^2 \sigma^2 \\
&= \left(1 - 2m\eta + \eta^2 K_1^2 + 64\eta^2 D^2 K_2^2\right) \hat{V}(\theta) \\
&\quad + 2m\eta - \eta^2 K_1^2 - 56\eta^2 D^2 K_2^2 + 64\eta^2 D^2 K_2^2 \|\hat{\theta}_*\|^2 + 2\eta K + \eta^2 \sigma^2 \\
&\leq (1 - m\eta)\hat{V}(\theta) + 2m\eta - \eta^2 K_1^2 - 56\eta^2 D^2 K_2^2 + 64\eta^2 D^2 K_2^2 \|\hat{\theta}_*\|^2 + 2\eta K + \eta^2 \sigma^2,
\end{aligned}
$$
(E.24)

provided that $\eta \leq \frac{m}{K_1^2 + 64 D^2 K_2^2}$. This completes the proof. $\qquad\square$

Next, we estimate the perturbation gap based on the Lyapunov function $\hat{V}$.

**Lemma E.4.** *Assume that Assumption 3.1 and Assumption 3.4 hold. Assume that $\sup_{x \in \mathcal{X}} \|x\| \leq D$ for some $D < \infty$. Then, we have*

$$
\begin{aligned}
&\sup_{\theta \in \mathbb{R}^d} \frac{d_\psi(\delta_\theta P, \delta_\theta \hat{P})}{\hat{V}(\theta)} \\
&\leq \frac{2b}{n} \max \left\{ \psi \cdot (4 + 8\eta^2 K_1^2), \right. \\
&\qquad\qquad \left. 1 + \psi \cdot \left(1 + \eta^2 \sigma^2 + (4 + 8\eta^2 K_1^2)\|\theta_* - \hat{\theta}_*\|^2 + 4\eta^2 \sup_{x \in \mathcal{X}} \|\nabla f(\theta_*, x)\|^2\right) \right\},
\end{aligned}
$$
(E.25)

*where $\hat{\theta}_*$ is the minimizer of $\frac{1}{n} \sum_{i=1}^n \nabla f(\theta, \hat{x}_i)$ and $\theta_*$ is the minimizer of $\frac{1}{n} \sum_{i=1}^n \nabla f(\theta, x_i)$ and $d_\psi$ is the weighted total variation distance defined in Section B.1.*

*Proof.* Let us recall that

$$
\theta_1 = \theta - \frac{\eta}{b} \sum_{i \in \Omega_1} \nabla f(\theta, x_i) + \eta \xi_1,
$$
(E.26)

$$
\hat{\theta}_1 = \theta - \frac{\eta}{b} \sum_{i \in \Omega_1} \nabla f(\theta, \hat{x}_i) + \eta \xi_1,
$$
(E.27)

which implies that

$$
\mathbb{E}_{\xi_1}[V(\theta_1)] = 1 + \mathbb{E}_{\xi_1}\left[\|\theta_1 - \theta_*\|^2\right] = 1 + \eta^2 \sigma^2 + \left\|\theta - \theta_* - \frac{\eta}{b} \sum_{i \in \Omega_1} \nabla f(\theta, x_i)\right\|^2,
$$
(E.28)

where $\mathbb{E}_{\xi_1}$ denotes expectations w.r.t. $\xi_1$ only, and we can further compute that

$$
\begin{aligned}
&\left\|\theta - \theta_* - \frac{\eta}{b} \sum_{i \in \Omega_1} \nabla f(\theta, x_i)\right\|^2 \\
&\leq 2\|\theta - \theta_*\|^2 + \frac{2\eta^2}{b^2} \left\|\sum_{i \in \Omega_1} \nabla f(\theta, x_i)\right\|^2 \\
&\leq 2\|\theta - \theta_*\|^2 + \frac{2\eta^2}{b^2} \left(\sum_{i \in \Omega_1} \|\nabla f(\theta, x_i)\|\right)^2 \\
&\leq 2\|\theta - \theta_*\|^2 + \frac{2\eta^2}{b^2} \left(\sum_{i \in \Omega_1} \|\nabla f(\theta, x_i) - \nabla f(\theta_*, x_i)\| + \|\nabla f(\theta_*, x_i)\|\right)^2 \\
&\leq 2\|\theta - \theta_*\|^2 + \frac{2\eta^2}{b^2} \left(b K_1 \|\theta - \theta_*\| + b \sup_{x \in \mathcal{X}} \|\nabla f(\theta_*, x)\|\right)^2 \\
&\leq 2\|\theta - \theta_*\|^2 + 4\eta^2 K_1^2 \|\theta - \theta_*\|^2 + 4\eta^2 \sup_{x \in \mathcal{X}} \|\nabla f(\theta_*, x)\|^2.
\end{aligned}
$$
(E.29)

Therefore, we have

$$\mathbb{E}_{\xi_1}[V(\theta_1)] \le 1 + \eta^2\sigma^2 + 2\|\theta - \theta_*\|^2 + 4\eta^2 K_1^2\|\theta - \theta_*\|^2 + 4\eta^2 \sup_{x \in \mathcal{X}} \|\nabla f(\theta_*, x)\|^2. \quad \text{(E.30)}$$

Similarly, we have

$$\mathbb{E}_{\xi_1}[V(\hat{\theta}_1)] \le 1 + \eta^2\sigma^2 + 2\|\theta - \theta_*\|^2 + 4\eta^2 K_1^2\|\theta - \theta_*\|^2 + 4\eta^2 \sup_{x \in \mathcal{X}} \|\nabla f(\theta_*, x)\|^2. \quad \text{(E.31)}$$

Since $X_n$ and $\hat{X}_n$ differ by at most one element and $\sup_{x \in \mathcal{X}} \|x\| \le D$ for some $D < \infty$, we have $x_i = \hat{x}_i$ for any $i \in \Omega_1$ with probability $\frac{n-b}{n}$ and $x_i \ne \hat{x}_i$ for exactly one $i \in \Omega_1$ with probability $\frac{b}{n}$. Therefore, we have

$$d_\psi\left(\delta_\theta P, \delta_\theta \hat{P}\right) \le \frac{2b}{n}\left(1 + \psi\left(1 + \eta^2\sigma^2 + (2 + 4\eta^2 K_1^2)\|\theta - \theta_*\|^2 + 4\eta^2 \sup_{x \in \mathcal{X}} \|\nabla f(\theta_*, x)\|^2\right)\right). \quad \text{(E.32)}$$

Hence, we conclude that

$$\sup_{\theta \in \mathbb{R}^d} \frac{d_\psi(\delta_\theta P, \delta_\theta \hat{P})}{\hat{V}(\theta)}$$

$$\le \frac{2b}{n} \sup_{\theta \in \mathbb{R}^d} \frac{1 + \psi\left(1 + \eta^2\sigma^2 + (2 + 4\eta^2 K_1^2)\|\theta - \theta_*\|^2 + 4\eta^2 \sup_{x \in \mathcal{X}} \|\nabla f(\theta_*, x)\|^2\right)}{1 + \|\theta - \hat{\theta}_*\|^2}$$

$$\le \frac{2b}{n} \sup_{\theta \in \mathbb{R}^d} \left\{ \frac{1 + \psi\left(1 + \eta^2\sigma^2 + (4 + 8\eta^2 K_1^2)\|\theta - \hat{\theta}_*\|^2\right)}{1 + \|\theta - \hat{\theta}_*\|^2} \right.$$

$$\left. + \frac{\psi\left((4 + 8\eta^2 K_1^2)\|\theta_* - \hat{\theta}_*\|^2 + 4\eta^2 \sup_{x \in \mathcal{X}} \|\nabla f(\theta_*, x)\|^2\right)}{1 + \|\theta - \hat{\theta}_*\|^2} \right\}$$

$$\le \frac{2b}{n} \max\left\{ \psi(4 + 8\eta^2 K_1^2), \right.$$

$$\left. 1 + \psi\left(1 + \eta^2\sigma^2 + (4 + 8\eta^2 K_1^2)\|\theta_* - \hat{\theta}_*\|^2 + 4\eta^2 \sup_{x \in \mathcal{X}} \|\nabla f(\theta_*, x)\|^2\right) \right\}. \quad \text{(E.33)}$$

This completes the proof. $\square$

It is worth noting that the Wasserstein contraction bound we obtained in Lemma E.1 in the non-convex case differs from the one we obtained in the strongly-convex case (Lemma D.1) in the sense that the right hand side of (E.3) is no longer $\|\theta - \tilde{\theta}\|$ so that Lemma 2.1 is not directly applicable. Instead, in the following, we will provide a modification of Lemma 2.1, which will be used in proving Theorem 3.3 in this paper. The definitions of the notations used in the following lemma can be found in Section 2.2.

**Lemma E.5.** *Assume that there exist some $\rho \in [0, 1)$ and $C \in (0, \infty)$ such that*

$$\sup_{\theta, \tilde{\theta} \in \mathbb{R}^d: \theta \ne \tilde{\theta}} \frac{\mathcal{W}_1(P^n(\theta, \cdot), P^n(\tilde{\theta}, \cdot))}{d_\psi(\delta_\theta, \delta_{\tilde{\theta}})} \le C\rho^n, \quad \text{(E.34)}$$

*for any $n \in \mathbb{N}$. Further assume that there exist some $\delta \in (0, 1)$ and $L \in (0, \infty)$ and a measurable Lyapunov function $\hat{V} : \mathbb{R}^d \to [1, \infty)$ of $\hat{P}$ such that for any $\theta \in \mathbb{R}^d$:*

$$(\hat{P}\hat{V})(\theta) \le \delta\hat{V}(\theta) + L. \quad \text{(E.35)}$$

*Then, we have*

$$\mathcal{W}_1(p_n, \hat{p}_n) \le C\left(\rho^n d_\psi(p_0, \hat{p}_0) + (1 - \rho^n)\frac{\gamma\kappa}{1 - \rho}\right), \quad \text{(E.36)}$$

*where*

$$\gamma := \sup_{\theta \in \mathbb{R}^d} \frac{d_\psi(\delta_\theta P, \delta_\theta \hat{P})}{\hat{V}(\theta)}, \qquad \kappa := \max\left\{\int_{\mathbb{R}^d} \hat{V}(\theta) d\hat{p}_0(\theta), \frac{L}{1 - \delta}\right\}. \quad \text{(E.37)}$$

*Proof.* The proof is based on the modification of the proof of Lemma 2.1 (Theorem 3.1 in [RS18]). By induction we have

$$\tilde{p}_n - p_n = (\tilde{p}_0 - p_0) P^n + \sum_{i=0}^{n-1} \tilde{p}_i \left( \tilde{P} - P \right) P^{n-i-1}, \qquad n \in \mathbb{N}. \tag{E.38}$$

We have

$$d_\psi \left( \tilde{p}_i P, \tilde{p}_i \tilde{P} \right) \leq \int_{\mathbb{R}^d} d_\psi \left( \delta_\theta P, \delta_\theta \tilde{P} \right) d\tilde{p}_i(\theta) \leq \gamma \int_{\mathbb{R}^d} \tilde{V}(\theta) d\tilde{p}_i(\theta). \tag{E.39}$$

Moreover, for any $i = 0, 1, 2, \ldots,$

$$\int_{\mathbb{R}^d} \tilde{V}(\theta) d\tilde{p}_i(\theta) = \int_{\mathbb{R}^d} \tilde{P}^i \tilde{V}(\theta) d\tilde{p}_0(\theta) \leq \delta^i \tilde{p}_0(\tilde{V}) + \frac{L(1-\delta^i)}{1-\delta} \leq \max \left\{ \tilde{p}_0(\tilde{V}), \frac{L}{1-\delta} \right\}, \tag{E.40}$$

so that we obtain $d_\psi(\tilde{p}_i P, \tilde{p}_i \tilde{P}) \leq \gamma \kappa$. Therefore, we have

$$\mathcal{W}_1 \left( \tilde{p}_i \tilde{P} P^{n-i-1}, \tilde{p}_i P P^{n-i-1} \right) \leq C \rho^{n-i-1} d_\psi \left( \tilde{p}_i P, \tilde{p}_i \tilde{P} \right) \leq C \rho^{n-i-1} \gamma \kappa. \tag{E.41}$$

By the triangle inequality of the Wasserstein distance, we have

$$\mathcal{W}_1(p_n, \tilde{p}_n) \leq \mathcal{W}_1 \left( p_0 P^n, \tilde{p}_0 P^n \right) + \sum_{i=0}^{n-1} \mathcal{W}_1 \left( \tilde{p}_i \tilde{P} P^{n-i-1}, \tilde{p}_i P P^{n-i-1} \right)$$

$$\leq C \rho^n d_\psi(p_0, \tilde{p}_0) + C \sum_{i=0}^{n-1} \rho^{n-i-1} \gamma \kappa. \tag{E.42}$$

This completes the proof. $\qquad \square$

Next, let us provide a technical lemma that upper bounds the norm of $\theta_*$ and $\hat{\theta}_*$, which are the minimizers of $\hat{F}(\theta, X_n) := \frac{1}{n} \sum_{i=1}^n \nabla f(\theta, x_i)$ and $\hat{F}(\theta, \hat{X}_n) := \frac{1}{n} \sum_{i=1}^n \nabla f(\theta, \hat{x}_i)$ respectively.

**Lemma E.6.** *Under Assumption 3.3, we have*

$$\|\theta_*\| \leq \frac{\sup_{x \in \mathcal{X}} \|\nabla f(0,x)\| + \sqrt{\sup_{x \in \mathcal{X}} \|\nabla f(0,x)\|^2 + 4mK}}{2m}, \tag{E.43}$$

$$\|\hat{\theta}_*\| \leq \frac{\sup_{x \in \mathcal{X}} \|\nabla f(0,x)\| + \sqrt{\sup_{x \in \mathcal{X}} \|\nabla f(0,x)\|^2 + 4mK}}{2m}. \tag{E.44}$$

*Proof.* By Assumption 3.3, we have

$$\left\langle \nabla \hat{F}(0, X_n) - \nabla \hat{F}(\theta_*, X_n), 0 - \theta_* \right\rangle = -\frac{1}{n} \sum_{i=1}^n \langle \nabla f(0, x_i), \theta_* \rangle$$

$$= \frac{1}{n} \sum_{i=1}^n \langle \nabla f(0, x_i) - \nabla f(\theta_*, x_i), 0 - \theta_* \rangle$$

$$\geq m\|\theta_*\|^2 - K, \tag{E.45}$$

which implies that

$$m\|\theta_*\|^2 - K \leq \sup_{x \in \mathcal{X}} \|\nabla f(0,x)\| \cdot \|\theta_*\|, \tag{E.46}$$

which yields that

$$\|\theta_*\| \leq \frac{\sup_{x \in \mathcal{X}} \|\nabla f(0,x)\| + \sqrt{\sup_{x \in \mathcal{X}} \|\nabla f(0,x)\|^2 + 4mK}}{2m}. \tag{E.47}$$

Similarly, one can show that

$$\|\hat{\theta}_*\| \leq \frac{\sup_{x \in \mathcal{X}} \|\nabla f(0,x)\| + \sqrt{\sup_{x \in \mathcal{X}} \|\nabla f(0,x)\|^2 + 4mK}}{2m}. \tag{E.48}$$

This completes the proof. $\qquad \square$

## E.1 Proof of Theorem 3.3

Before going to the proof, let us restate the full version of Theorem 3.3 that we provide below.

**Theorem E.1** (Complete **Theorem 3.3**). *Let $\theta_0 = \hat{\theta}_0 = \theta$. Assume that Assumption 3.1, Assumption 3.3 and Assumption 3.4 hold. We also assume that $\eta < \min\left\{\frac{1}{m}, \frac{m}{K_1^2 + 64D^2 K_2^2}\right\}$ and $\sup_{x \in \mathcal{X}} \|x\| \leq D$ for some $D < \infty$ and $\sup_{x \in \mathcal{X}} \|\nabla f(0, x)\| \leq E$ for some $E < \infty$. For any $\hat{\eta} \in (0, 1)$. Define $M > 0$ so that $\int_{\|\theta_1 - \theta_*\| \leq M} p(\theta_*, \theta_1) d\theta_1 \geq \sqrt{\hat{\eta}}$ and any $R > \frac{2K_0}{m}$ where $K_0$ is defined in* (E.2) *so that*

$$\inf_{\theta, \theta_1 \in \mathbb{R}^d : V(\theta) \leq R, \|\theta_1 - \theta_*\| \leq M} \frac{p(\theta, \theta_1)}{p(\theta_*, \theta_1)} \geq \sqrt{\hat{\eta}}. \tag{E.49}$$

*Let $\nu_k$ and $\hat{\nu}_k$ denote the distributions of $\theta_k$ and $\hat{\theta}_k$ respectively. Then, we have*

$$\mathcal{W}_1(\nu_k, \hat{\nu}_k)$$

$$\leq \frac{1 - \bar{\eta}^k}{2\sqrt{\psi(1 + \psi)(1 - \bar{\eta})}}$$

$$\cdot \frac{2b}{n} \max\left\{ \psi(4 + 8\eta^2 K_1^2), \right.$$

$$1 + \psi\left(1 + \eta^2 \sigma^2 + 16(1 + 2\eta^2 K_1^2)\left(\frac{E + \sqrt{E^2 + 4mK}}{2m}\right)^2\right.$$

$$\left.\left.+ 4\eta^2\left(2E^2 + 2K_1^2\left(\frac{E + \sqrt{E^2 + 4mK}}{2m}\right)^2\right)\right)\right\}$$

$$\cdot \max\left\{1 + 2\theta^2 + 2\left(\frac{E + \sqrt{E^2 + 4mK}}{2m}\right)^2,\right.$$

$$\left. 2 - \frac{\eta}{m}K_1^2 - \frac{56\eta}{m}D^2 K_2^2 + \frac{64\eta}{m}D^2 K_2^2\left(\frac{E + \sqrt{E^2 + 4mK}}{2m}\right)^2 + \frac{2K}{m} + \frac{\eta}{m}\sigma^2\right\}, \tag{E.50}$$

*where for any $\eta_0 \in (0, \hat{\eta})$ and $\gamma_0 \in \left(1 - m\eta + \frac{2\eta K_0}{R}, 1\right)$ one can choose $\psi = \frac{\eta_0}{\eta K_0}$ and $\bar{\eta} = (1 - (\hat{\eta} - \eta_0)) \vee \frac{2 + R\psi\gamma_0}{2 + R\psi}$ and $d_\psi$ is the weighted total variation distance defined in Section B.1.*

*Proof.* By applying Lemma E.1, Lemma E.3, Lemma E.4 and Lemma E.5, which is a modification of Lemma 2.1, we obtain

$$\mathcal{W}_1(\nu_k, \hat{\nu}_k)$$

$$\leq \frac{1 - \bar{\eta}^k}{2\sqrt{\psi(1 + \psi)(1 - \bar{\eta})}}$$

$$\cdot \frac{2b}{n} \max\left\{ \psi(4 + 8\eta^2 K_1^2), \right.$$

$$\left. 1 + \psi\left(1 + \eta^2\sigma^2 + (4 + 8\eta^2 K_1^2)\|\theta_* - \hat{\theta}_*\|^2 + 4\eta^2 \sup_{x \in \mathcal{X}}\|\nabla f(\theta_*, x)\|^2\right)\right\}$$

$$\cdot \max\left\{1 + \|\theta - \hat{\theta}_*\|^2, 2 - \frac{\eta}{m}K_1^2 - \frac{56\eta}{m}D^2 K_2^2 + \frac{64\eta}{m}D^2 K_2^2\|\hat{\theta}_*\|^2 + \frac{2K}{m} + \frac{\eta}{m}\sigma^2\right\}, \tag{E.51}$$

where for any $\eta_0 \in (0, \hat{\eta})$ and $\gamma_0 \in \left(1 - m\eta + \frac{2\eta K_0}{R}, 1\right)$ one can choose $\psi = \frac{\eta_0}{\eta K_0}$ and $\bar{\eta} = (1 - (\hat{\eta} - \eta_0)) \vee \frac{2 + R\psi\gamma_0}{2 + R\psi}$ and $d_\psi$ is the weighted total variation distance defined in Section B.1.

Finally, let us notice that $\|\theta_* - \hat{\theta}_*\|^2 \leq 2\|\theta_*\|^2 + 2\|\hat{\theta}_*\|^2$ and $\|\theta - \hat{\theta}_*\|^2 \leq 2\|\theta\|^2 + 2\|\hat{\theta}_*\|^2$ and for every $x \in \mathcal{X}$,

$$\|\nabla f(\theta_*, x)\|^2 \leq 2\|\nabla f(0, x)\|^2 + 2\|\nabla f(\theta_*, x) - \nabla f(0, x)\|^2 \leq 2\|\nabla f(0, x)\|^2 + 2K_1^2\|\theta_*\|^2. \quad (E.52)$$

By applying Lemma E.6, we complete the proof. $\qquad\square$

## E.2 Proof of Corollary 3.5

*Proof.* Under our assumptions, the noise $\xi_k$ are i.i.d. Gaussian $\mathcal{N}(0, \Sigma)$ so that $\mathbb{E}[\|\xi_1\|^2] = \text{tr}(\Sigma) = \sigma^2$. Moreoever, we have $\Sigma \prec I_d$. Then $p(\theta_*, \theta_1)$ is the probability density function of

$$\theta_* - \frac{\eta}{b} \sum_{i \in \Omega_1} \nabla f(\theta_*, x_i) + \eta \xi_1. \quad (E.53)$$

Therefore,

$$\int_{\|\theta_1 - \theta_*\| \leq M} p(\theta_*, \theta_1) d\theta_1 = \mathbb{P}\left( \left\| -\frac{\eta}{b} \sum_{i \in \Omega_1} \nabla f(\theta_*, x_i) + \eta \xi_1 \right\| \leq M \right). \quad (E.54)$$

Notice that for any $\Omega_1$,

$$\left\| \frac{\eta}{b} \sum_{i \in \Omega_1} \nabla f(\theta_*, x_i) \right\| \leq \eta \sup_{x \in \mathcal{X}} \|\nabla f(\theta_*, x)\|. \quad (E.55)$$

Thus, we have

$$\int_{\|\theta_1 - \theta_*\| \leq M} p(\theta_*, \theta_1) d\theta_1 = 1 - \mathbb{P}\left( \left\| -\frac{\eta}{b} \sum_{i \in \Omega_1} \nabla f(\theta_*, x_i) + \eta \xi_1 \right\| > M \right)$$

$$\geq 1 - \mathbb{P}\left( \|\eta \xi_1\| > M - \eta \sup_{x \in \mathcal{X}} \|\nabla f(\theta_*, x)\| \right)$$

$$= 1 - \mathbb{P}\left( \|\xi_1\| > \frac{M}{\eta} - \sup_{x \in \mathcal{X}} \|\nabla f(\theta_*, x)\| \right). \quad (E.56)$$

Since $\xi \sim \mathcal{N}(0, \Sigma)$ and $\Sigma \prec I_d$, for any $\gamma \leq \frac{1}{2}$, we have

$$\mathbb{E}\left[ e^{\gamma \|\xi_1\|^2} \right] = \frac{1}{\sqrt{\det(I_d - 2\gamma\Sigma)}}.$$

By Chebychev's inequality, letting $\gamma = \frac{1}{2}$, for any $M \geq \eta \sup_{x \in \mathcal{X}} \|\nabla f(\theta_*, x)\|$, we get

$$\int_{\|\theta_1 - \theta_*\| \leq M} p(\theta_*, \theta_1) d\theta_1 \geq 1 - \frac{1}{\sqrt{\det(I_d - \Sigma)}} \exp\left( -\frac{1}{2} \left( \frac{M}{\eta} - \sup_{x \in \mathcal{X}} \|\nabla f(\theta_*, x)\| \right)^2 \right).$$

Next, for any $\theta, \theta_1 \in \mathbb{R}^d$ such that $\|\theta_1 - \theta_*\| \leq M$ and $\|\theta - \theta_*\|^2 \leq \frac{2K_0}{m}(1 + \epsilon) - 1$, we have

$$\frac{p(\theta, \theta_1)}{p(\theta_*, \theta_1)} = \frac{\mathbb{E}_{\Omega_1}[p_{\Omega_1}(\theta, \theta_1)]}{\mathbb{E}_{\Omega_1}[p_{\Omega_1}(\theta_*, \theta_1)]}, \quad (E.57)$$

where $\mathbb{E}_{\Omega_1}$ denotes the expectation w.r.t. $\Omega_1$, and $p_{\Omega_1}$ denotes the probability density function conditional on $\Omega_1$. For any given $\Omega_1$, we can compute that

$$\frac{p_{\Omega_1}(\theta, \theta_1)}{p_{\Omega_1}(\theta_*, \theta_1)}$$

$$= \exp\left\{ -\frac{1}{2\eta^2}(\theta_1 - \mu(\theta))^\top \Sigma^{-1}(\theta_1 - \mu(\theta)) + \frac{1}{2\eta^2}(\theta_1 - \mu(\theta_*))^\top \Sigma^{-1}(\theta_1 - \mu(\theta_*)) \right\}, \quad (E.58)$$

where

$$\mu(\theta) := \theta - \frac{\eta}{b} \sum_{i \in \Omega_1} \nabla f(\theta, x_i), \qquad \mu(\theta_*) := \theta_* - \frac{\eta}{b} \sum_{i \in \Omega_1} \nabla f(\theta_*, x_i). \quad (E.59)$$

Therefore, for any $\theta, \theta_1 \in \mathbb{R}^d$ such that $\|\theta_1 - \theta_*\| \leq M$ and $\|\theta - \theta_*\|^2 \leq \frac{2K_0}{m}(1+\epsilon) - 1$, we have

$$
\begin{aligned}
\frac{p_{\Omega_1}(\theta, \theta_1)}{p_{\Omega_1}(\theta_*, \theta_1)} &\geq \exp\left\{-\frac{1}{2\eta^2}\|\mu(\theta) - \mu(\theta_*)\| \cdot \|\Sigma^{-1}\| \left(\|\theta_1 - \mu(\theta)\| + \|\theta_1 - \mu(\theta_*)\|\right)\right\} \\
&\geq \exp\left\{-\frac{1}{2\eta^2}\|\mu(\theta) - \mu(\theta_*)\| \cdot \|\Sigma^{-1}\| \left(\|\mu(\theta) - \mu(\theta_*)\| + 2\|\theta_1 - \mu(\theta_*)\|\right)\right\}.
\end{aligned}
\tag{E.60}
$$

We can further compute that

$$
\|\theta_1 - \mu(\theta_*)\| \leq \|\theta_1 - \theta_*\| + \left\|\frac{\eta}{b}\sum_{i \in \Omega_1} \nabla f(\theta_*, x_i)\right\| \leq M + \eta \sup_{x \in \mathcal{X}} \|\nabla f(\theta_*, x)\|,
\tag{E.61}
$$

and

$$
\begin{aligned}
\|\mu(\theta) - \mu(\theta_*)\| &\leq \|\theta - \theta_*\| + \frac{\eta}{b}\sum_{i \in \Omega_1} \|\nabla f(\theta, x_i) - \nabla f(\theta_*, x_i)\| \\
&\leq (1 + K_1\eta)\|\theta - \theta_*\| \\
&\leq (1 + K_1\eta)\left(\frac{2K_0}{m}(1+\epsilon) - 1\right)^{1/2}.
\end{aligned}
\tag{E.62}
$$

Hence, we have

$$
\begin{aligned}
\frac{p_{\Omega_1}(\theta, \theta_1)}{p_{\Omega_1}(\theta_*, \theta_1)} &\geq \exp\left\{-\frac{(1 + K_1\eta)\left(\frac{2K_0}{m}(1+\epsilon) - 1\right)^{1/2}}{2\eta^2}\|\Sigma^{-1}\|\right. \\
&\left.\quad \cdot \left((1 + K_1\eta)\left(\frac{2K_0}{m}(1+\epsilon) - 1\right)^{1/2} + 2\left(M + \eta \sup_{x \in \mathcal{X}} \|\nabla f(\theta_*, x)\|\right)\right)\right\}.
\end{aligned}
\tag{E.63}
$$

Since it holds for every $\Omega_1$, we have

$$
\begin{aligned}
\frac{p(\theta, \theta_1)}{p(\theta_*, \theta_1)} &\geq \exp\left\{-\frac{(1 + K_1\eta)\left(\frac{2K_0}{m}(1+\epsilon) - 1\right)^{1/2}}{2\eta^2}\|\Sigma^{-1}\|\right. \\
&\left.\quad \cdot \left((1 + K_1\eta)\left(\frac{2K_0}{m}(1+\epsilon) - 1\right)^{1/2} + 2\left(M + \eta \sup_{x \in \mathcal{X}} \|\nabla f(\theta_*, x)\|\right)\right)\right\}.
\end{aligned}
\tag{E.64}
$$

Hence, we conclude that

$$
\begin{aligned}
\hat{\eta} \geq \Bigg(&\max_{M \geq \eta \sup_{x \in \mathcal{X}} \|\nabla f(\theta_*, x)\|} \Bigg\{\min\Bigg\{1 - \frac{\exp\left(-\frac{1}{2}\left(\frac{M}{\eta} - \sup_{x \in \mathcal{X}} \|\nabla f(\theta_*, x)\|\right)^2\right)}{\sqrt{\det(I_d - \Sigma)}}, \\
&\exp\Bigg\{-\frac{(1 + K_1\eta)\left(\frac{2K_0}{m}(1+\epsilon) - 1\right)^{1/2}}{2\eta^2}\|\Sigma^{-1}\| \\
&\quad \cdot \left((1 + K_1\eta)\left(\frac{2K_0}{m}(1+\epsilon) - 1\right)^{1/2} + 2\left(M + \eta \sup_{x \in \mathcal{X}} \|\nabla f(\theta_*, x)\|\right)\right)\Bigg\}\Bigg\}\Bigg\}\Bigg)^2.
\end{aligned}
\tag{E.65}
$$

This completes the proof. $\qquad\square$

# F   Proofs of Non-Convex Case without Additive Noise

## F.1   Proof of Theorem 4.1

*Proof.* Let us recall that $\theta_0 = \hat{\theta}_0 = \theta$ and for any $k \in \mathbb{N}$,

$$\theta_k = \theta_{k-1} - \frac{\eta}{b} \sum_{i \in \Omega_k} \nabla f(\theta_{k-1}, x_i), \tag{F.1}$$

$$\hat{\theta}_k = \hat{\theta}_{k-1} - \frac{\eta}{b} \sum_{i \in \Omega_k} \nabla f(\hat{\theta}_{k-1}, \hat{x}_i). \tag{F.2}$$

Thus it follows that

$$\theta_k - \hat{\theta}_k = \theta_{k-1} - \hat{\theta}_{k-1} - \frac{\eta}{b} \sum_{i \in \Omega_k} \left( \nabla f(\theta_{k-1}, x_i) - \nabla f\left(\hat{\theta}_{k-1}, x_i\right) \right) + \frac{\eta}{b} \mathcal{E}_k, \tag{F.3}$$

where

$$\mathcal{E}_k := \sum_{i \in \Omega_k} \left( \nabla f(\hat{\theta}_{k-1}, \hat{x}_i) - \nabla f\left(\hat{\theta}_{k-1}, x_i\right) \right). \tag{F.4}$$

This implies that

$$\left\| \theta_k - \hat{\theta}_k \right\|^2 = \left\| \theta_{k-1} - \hat{\theta}_{k-1} - \frac{\eta}{b} \sum_{i \in \Omega_k} \left( \nabla f(\theta_{k-1}, x_i) - \nabla f\left(\hat{\theta}_{k-1}, x_i\right) \right) \right\|^2 + \frac{\eta^2}{b^2} \left\| \mathcal{E}_k \right\|^2$$

$$+ 2 \left\langle \theta_{k-1} - \hat{\theta}_{k-1} - \frac{\eta}{b} \sum_{i \in \Omega_k} \left( \nabla f(\theta_{k-1}, x_i) - \nabla f\left(\hat{\theta}_{k-1}, x_i\right) \right), \frac{\eta}{b} \mathcal{E}_k \right\rangle. \tag{F.5}$$

By Assumption 3.1 and Assumption 3.3, we have

$$\left\| \theta_{k-1} - \hat{\theta}_{k-1} - \frac{\eta}{b} \sum_{i \in \Omega_k} \left( \nabla f(\theta_{k-1}, x_i) - \nabla f\left(\hat{\theta}_{k-1}, x_i\right) \right) \right\|^2$$

$$\leq (1 - 2\eta m) \left\| \theta_{k-1} - \hat{\theta}_{k-1} \right\|^2 + 2\eta K + \frac{\eta^2}{b^2} \left( b K_1 \left\| \theta_{k-1} - \hat{\theta}_{k-1} \right\| \right)^2$$

$$\leq (1 - \eta m) \left\| \theta_{k-1} - \hat{\theta}_{k-1} \right\|^2 + 2\eta K, \tag{F.6}$$

provided that $\eta \leq \frac{m}{K_1^2}$.

Since $X_n$ and $\hat{X}_n$ differ by at most one element and $\sup_{x \in \mathcal{X}} \|x\| \leq D$ for some $D < \infty$, we have $x_i = \hat{x}_i$ for any $i \in \Omega_k$ with probability $\frac{n-b}{n}$ and $x_i \neq \hat{x}_i$ for exactly one $i \in \Omega_k$ with probability $\frac{b}{n}$ and therefore

$$\mathbb{E} \left\| \mathcal{E}_k \right\|^2 \leq \frac{b}{n} \mathbb{E} \left[ \left( K_2 2D \left( 2\|\hat{\theta}_{k-1}\| + 1 \right) \right)^2 \right] \leq \frac{4 D^2 K_2^2 b}{n} \left( 8 \mathbb{E}\|\hat{\theta}_{k-1}\|^2 + 2 \right), \tag{F.7}$$

and moreover,

$$\mathbb{E} \left\langle \theta_{k-1} - \hat{\theta}_{k-1} - \frac{\eta}{b} \sum_{i \in \Omega_k} \left( \nabla f(\theta_{k-1}, x_i) - \nabla f\left(\hat{\theta}_{k-1}, x_i\right) \right), \frac{\eta}{b} \mathcal{E}_k \right\rangle$$

$$\leq \frac{\eta}{b} \frac{b}{n} \mathbb{E} \left[ (1 + K_1 \eta) \left\| \theta_{k-1} - \hat{\theta}_{k-1} \right\| K_2 2D \left( 2\|\hat{\theta}_{k-1}\| + 1 \right) \right]$$

$$\leq \frac{2 K_2 D \eta}{n} (1 + K_1 \eta) \mathbb{E} \left[ \left( \|\theta_{k-1}\| + \|\hat{\theta}_{k-1}\| \right) \left( 2\|\hat{\theta}_{k-1}\| + 1 \right) \right]$$

$$\leq \frac{2 K_2 D \eta}{n} (1 + K_1 \eta) \left( 1 + \frac{3}{2} \mathbb{E}\|\theta_{k-1}\|^2 + \frac{7}{2} \mathbb{E}\|\hat{\theta}_{k-1}\|^2 \right), \tag{F.8}$$

where we used the inequality that

$$(a + b)(2b + 1) = 2b^2 + 2ab + a + b \leq 1 + \frac{3}{2} a^2 + \frac{7}{2} b^2, \tag{F.9}$$

for any $a, b \in \mathbb{R}$. Therefore, we have

$$
\mathbb{E} \left\| \theta_k - \hat{\theta}_k \right\|^2 \leq (1 - \eta m) \mathbb{E} \left\| \theta_{k-1} - \hat{\theta}_{k-1} \right\|^2 + 2\eta K + \frac{4D^2 K_2^2 \eta^2}{bn} \left( 8\mathbb{E}\|\hat{\theta}_{k-1}\|^2 + 2 \right)
$$
$$
+ \frac{4K_2 D \eta}{n}(1 + K_1 \eta) \left( 1 + \frac{3}{2}\mathbb{E}\|\theta_{k-1}\|^2 + \frac{7}{2}\mathbb{E}\|\hat{\theta}_{k-1}\|^2 \right). \tag{F.10}
$$

In Lemma E.3, we showed that under the assumption $\eta < \min\left\{ \frac{1}{m}, \frac{m}{K_1^2 + 64D^2 K_2^2} \right\}$ and $\sup_{x \in \mathcal{X}} \|x\| \leq D$ for some $D < \infty$, we have that for every $k \in \mathbb{N}$,

$$
\mathbb{E}\hat{V}(\hat{\theta}_k) \leq (1 - \eta m)\mathbb{E}\hat{V}(\hat{\theta}_{k-1}) + 2\eta m - \eta^2 K_1^2 - 56\eta^2 D^2 K_2^2 + 64\eta^2 D^2 K_2^2 \|\hat{\theta}_*\|^2 + 2\eta K, \tag{F.11}
$$

where $\hat{V}(\theta) := 1 + \|\theta - \hat{\theta}_*\|^2$, where $\hat{\theta}_*$ is the minimizer of $\hat{F}(\theta, \hat{X}_n) := \frac{1}{n} \sum_{i=1}^n \nabla f(\theta, \hat{x}_i)$. This implies that

$$
\mathbb{E}\left[ \hat{V}\left( \hat{\theta}_k \right) \right]
$$
$$
\leq (1 - \eta m)^k \mathbb{E}\left[ \hat{V}\left( \hat{\theta}_0 \right) \right] + \frac{2\eta m - \eta^2 K_1^2 - 56\eta^2 D^2 K_2^2 + 64\eta^2 D^2 K_2^2 \|\hat{\theta}_*\|^2 + 2\eta K}{1 - (1 - \eta m)}
$$
$$
\leq 1 + \|\theta - \hat{\theta}_*\|^2 + 2 - \frac{\eta}{m} K_1^2 - \frac{56\eta}{m} D^2 K_2^2 + \frac{64\eta}{m} D^2 K_2^2 \|\hat{\theta}_*\|^2 + \frac{2K}{m}, \tag{F.12}
$$

so that

$$
\mathbb{E}\|\hat{\theta}_k\|^2 \leq 2\mathbb{E}\left\| \hat{\theta}_k - \hat{\theta}_* \right\|^2 + 2\|\hat{\theta}_*\|^2
$$
$$
\leq 2\left\| \theta - \hat{\theta}_* \right\|^2 + 4 - \frac{2\eta}{m} K_1^2 - \frac{112\eta}{m} D^2 K_2^2 + \frac{128\eta}{m} D^2 K_2^2 \|\hat{\theta}_*\|^2 + \frac{4K}{m} + 2\|\hat{\theta}_*\|^2
$$
$$
\leq 4\|\theta\|^2 + 4\left( \frac{E + \sqrt{E^2 + 4mK}}{2m} \right)^2 + 4 - \frac{2\eta}{m} K_1^2 - \frac{112\eta}{m} D^2 K_2^2
$$
$$
+ \frac{128\eta}{m} D^2 K_2^2 \left( \frac{E + \sqrt{E^2 + 4mK}}{2m} \right)^2 + \frac{4K}{m} + 2\left( \frac{E + \sqrt{E^2 + 4mK}}{2m} \right)^2 =: B, \tag{F.13}
$$

where we applied Lemma E.6. Similarly, we can show that

$$
\mathbb{E}\|\theta_k\|^2 \leq B. \tag{F.14}
$$

Since $\theta_0 = \hat{\theta}_0 = \theta$, it follows from (F.10), (F.13) and (F.14) that

$$
\mathcal{W}_2^2(\nu_k, \hat{\nu}_k)
$$
$$
\leq \mathbb{E}\left\| \theta_k - \hat{\theta}_k \right\|^2
$$
$$
\leq \left(1 - (1 - \eta m)^k\right) \left( \frac{4D^2 K_2^2 \eta}{bnm}(8B + 2) + \frac{4K_2 D}{nm}(1 + K_1 \eta)\left(1 + \frac{3}{2}B + \frac{7}{2}B\right) + \frac{2K}{m} \right), \tag{F.15}
$$

provided that $\eta < \min\left\{ \frac{m}{K_1^2}, \frac{1}{m}, \frac{m}{K_1^2 + 64D^2 K_2^2} \right\}$. This completes the proof. $\qquad \square$

# G   Proofs of Convex Case with Additional Geometric Structure

In the following technical lemma, we show that the $p$-th moment of $\theta_k$ and $\hat{\theta}_k$ can be bounded in the following sense.

**Lemma G.1.** *Let $\theta_0 = \hat{\theta}_0 = \theta$. Suppose Assumption 4.1 and Assumption 4.2 hold and $\eta \leq \frac{\mu}{K_1^2 + 2^{p+4} D^2 K_2^2}$. Then, we have*

$$\frac{1}{k} \sum_{i=1}^k \mathbb{E} \left\| \theta_{i-1} - \theta_* \right\|^p \leq \frac{\left\| \theta - \theta_* \right\|^2}{k \eta \mu} + \frac{8\eta}{\mu} D^2 K_2^2 \left( 2^{p+1} \|\theta_*\|^p + 5 \right), \tag{G.1}$$

$$\frac{1}{k} \sum_{i=1}^k \mathbb{E} \left\| \hat{\theta}_{i-1} - \hat{\theta}_* \right\|^p \leq \frac{\left\| \theta - \hat{\theta}_* \right\|^2}{k \eta \mu} + \frac{8\eta}{\mu} D^2 K_2^2 \left( 2^{p+1} \|\hat{\theta}_*\|^p + 5 \right), \tag{G.2}$$

*where $\hat{\theta}_*$ is the minimizer of $\frac{1}{n} \sum_{i=1}^n \nabla f(\theta, \hat{x}_i)$ and $\theta_*$ is the minimizer of $\frac{1}{n} \sum_{i=1}^n \nabla f(\theta, x_i)$.*

*Proof.* First, we recall that

$$\hat{\theta}_k = \hat{\theta}_{k-1} - \frac{\eta}{b} \sum_{i \in \Omega_k} \nabla f \left( \hat{\theta}_{k-1}, \hat{x}_i \right). \tag{G.3}$$

Therefore, we have

$$\hat{\theta}_k = \hat{\theta}_{k-1} - \frac{\eta}{n} \sum_{i=1}^n \nabla f \left( \hat{\theta}_{k-1}, \hat{x}_i \right) + \eta \left( \frac{1}{n} \sum_{i=1}^n \nabla f \left( \hat{\theta}_{k-1}, \hat{x}_i \right) - \frac{1}{b} \sum_{i \in \Omega_k} \nabla f \left( \hat{\theta}_{k-1}, \hat{x}_i \right) \right). \tag{G.4}$$

Moreover, we have

$$\mathbb{E} \left[ \frac{1}{b} \sum_{i \in \Omega_k} \nabla f \left( \hat{\theta}_{k-1}, \hat{x}_i \right) \middle| \hat{\theta}_{k-1} \right] = \frac{1}{n} \sum_{i=1}^n \nabla f \left( \hat{\theta}_{k-1}, \hat{x}_i \right). \tag{G.5}$$

This implies that

$$\mathbb{E} \left\| \hat{\theta}_k - \hat{\theta}_* \right\|^2 = \mathbb{E} \left\| \hat{\theta}_{k-1} - \hat{\theta}_* + \frac{\eta}{n} \sum_{i=1}^n \nabla f \left( \hat{\theta}_{k-1}, \hat{x}_i \right) \right\|^2$$

$$+ \eta^2 \mathbb{E} \left\| \frac{1}{n} \sum_{i=1}^n \nabla f \left( \hat{\theta}_{k-1}, \hat{x}_i \right) - \frac{1}{b} \sum_{i \in \Omega_k} \nabla f \left( \hat{\theta}_{k-1}, \hat{x}_i \right) \right\|^2. \tag{G.6}$$

We can compute that

$$\left\| \hat{\theta}_{k-1} - \hat{\theta}_* + \frac{\eta}{n} \sum_{i=1}^n \nabla f \left( \hat{\theta}_{k-1}, \hat{x}_i \right) \right\|^2$$

$$= \left\| \hat{\theta}_{k-1} - \hat{\theta}_* + \frac{\eta}{n} \sum_{i=1}^n \left( \nabla f \left( \hat{\theta}_{k-1}, \hat{x}_i \right) - \nabla f \left( \hat{\theta}_*, \hat{x}_i \right) \right) \right\|^2$$

$$\leq \left\| \hat{\theta}_{k-1} - \hat{\theta}_* \right\|^2 - 2\eta\mu \left\| \hat{\theta}_{k-1} - \hat{\theta}_* \right\|^p + \frac{\eta^2}{n^2} \left\| \sum_{i=1}^n \left( \nabla f \left( \hat{\theta}_{k-1}, \hat{x}_i \right) - \nabla f \left( \hat{\theta}_*, \hat{x}_i \right) \right) \right\|^2$$

$$\leq \left\| \hat{\theta}_{k-1} - \hat{\theta}_* \right\|^2 - 2\eta\mu \left\| \hat{\theta}_{k-1} - \hat{\theta}_* \right\|^p + \eta^2 K_1^2 \left\| \hat{\theta}_{k-1} - \hat{\theta}_* \right\|^p. \tag{G.7}$$

Moreover, we can compute that

$$\mathbb{E} \left\| \frac{1}{n} \sum_{i=1}^{n} \nabla f\left(\hat{\theta}_{k-1}, \hat{x}_i\right) - \frac{1}{b} \sum_{i \in \Omega_k} \nabla f\left(\hat{\theta}_{k-1}, \hat{x}_i\right) \right\|^2$$

$$= \mathbb{E} \left\| \frac{1}{b} \sum_{i \in \Omega_k} \left( \frac{1}{n} \sum_{j=1}^{n} \nabla f\left(\hat{\theta}_{k-1}, \hat{x}_j\right) - \nabla f\left(\hat{\theta}_{k-1}, \hat{x}_i\right) \right) \right\|^2$$

$$= \mathbb{E} \left( \frac{1}{b} \sum_{i \in \Omega_k} K_2 \frac{1}{n} \sum_{j=1}^{n} \|\hat{x}_i - \hat{x}_j\| (2\|\hat{\theta}_{k-1}\|^{p-1} + 1) \right)^2$$

$$\leq 4D^2 K_2^2 \mathbb{E} \left[ (2\|\hat{\theta}_{k-1}\|^{p-1} + 1)^2 \right]$$

$$\leq 8D^2 K_2^2 \left( 4\mathbb{E} \left[ \|\hat{\theta}_{k-1}\|^{2(p-1)} \right] + 1 \right)$$

$$\leq 8D^2 K_2^2 \left( 4\mathbb{E} \left[ \|\hat{\theta}_{k-1}\|^p \right] + 5 \right)$$

$$\leq 8D^2 K_2^2 \left( 2^{p+1} \mathbb{E} \|\hat{\theta}_{k-1} - \hat{\theta}_*\|^p + 2^{p+1} \|\hat{\theta}_*\|^p + 5 \right). \tag{G.8}$$

Hence, by applying (G.7) and (G.8) to (G.6), we conclude that

$$\mathbb{E} \left\| \hat{\theta}_k - \hat{\theta}_* \right\|^2 \leq \mathbb{E} \left\| \hat{\theta}_{k-1} - \hat{\theta}_* \right\|^2 - 2\eta\mu \mathbb{E} \left\| \hat{\theta}_{k-1} - \hat{\theta}_* \right\|^p + \eta^2 K_1^2 \mathbb{E} \left\| \hat{\theta}_{k-1} - \hat{\theta}_* \right\|^p$$

$$+ 8\eta^2 D^2 K_2^2 \left( 2^{p+1} \mathbb{E} \left\| \hat{\theta}_{k-1} - \hat{\theta}_* \right\|^2 + 2^{p+1} \|\hat{\theta}_*\|^2 + 5 \right)$$

$$\leq \mathbb{E} \left\| \hat{\theta}_{k-1} - \hat{\theta}_* \right\|^2 - \eta\mu \mathbb{E} \left\| \hat{\theta}_{k-1} - \hat{\theta}_* \right\|^p + 8\eta^2 D^2 K_2^2 \left( 2^{p+1} \|\hat{\theta}_*\|^p + 5 \right), \tag{G.9}$$

provided that $\eta \leq \frac{\mu}{K_1^2 + 2^{p+4} D^2 K_2^2}$. This implies that

$$\mathbb{E} \left\| \hat{\theta}_{k-1} - \hat{\theta}_* \right\|^p \leq \frac{\mathbb{E} \left\| \hat{\theta}_{k-1} - \hat{\theta}_* \right\|^2 - \mathbb{E} \left\| \hat{\theta}_k - \hat{\theta}_* \right\|^2}{\eta\mu} + \frac{8\eta}{\mu} D^2 K_2^2 \left( 2^{p+1} \|\hat{\theta}_*\|^p + 5 \right), \tag{G.10}$$

and hence

$$\frac{1}{k} \sum_{i=1}^{k} \mathbb{E} \left\| \hat{\theta}_{i-1} - \hat{\theta}_* \right\|^p \leq \frac{\left\| \hat{\theta}_0 - \hat{\theta}_* \right\|^2 - \mathbb{E} \left\| \hat{\theta}_k - \hat{\theta}_* \right\|^2}{k\eta\mu} + \frac{8\eta}{\mu} D^2 K_2^2 \left( 2^{p+1} \|\hat{\theta}_*\|^p + 5 \right)$$

$$\leq \frac{\left\| \theta - \hat{\theta}_* \right\|^2}{k\eta\mu} + \frac{8\eta}{\mu} D^2 K_2^2 \left( 2^{p+1} \|\hat{\theta}_*\|^p + 5 \right). \tag{G.11}$$

Similarly, we can show that

$$\frac{1}{k} \sum_{i=1}^{k} \mathbb{E} \|\theta_{i-1} - \theta_*\|^p \leq \frac{\|\theta - \theta_*\|^2}{k\eta\mu} + \frac{8\eta}{\mu} D^2 K_2^2 \left( 2^{p+1} \|\theta_*\|^p + 5 \right). \tag{G.12}$$

This completes the proof. $\qquad\square$

Next, let us provide a technical lemma that upper bounds the norm of $\theta_*$ and $\hat{\theta}_*$, which are the minimizers of $\hat{F}(\theta, X_n) := \frac{1}{n} \sum_{i=1}^{n} \nabla f(\theta, x_i)$ and $\hat{F}(\theta, \hat{X}_n) := \frac{1}{n} \sum_{i=1}^{n} \nabla f(\theta, \hat{x}_i)$ respectively.

**Lemma G.2.** *Under Assumption 4.1, we have*

$$\|\theta_*\| \leq \frac{1}{\mu^{\frac{1}{p-1}}} \sup_{x \in \mathcal{X}} \|\nabla f(0, x)\|^{\frac{1}{p-1}},$$

*and*

$$\|\hat{\theta}_*\| \leq \frac{1}{\mu^{\frac{1}{p-1}}} \sup_{x \in \mathcal{X}} \|\nabla f(0, x)\|^{\frac{1}{p-1}}.$$

*Proof.* Under Assumption 4.1, we have

$$\left\langle \nabla \hat{F}(0, X_n) - \nabla \hat{F}(\theta_*, X_n), 0 - \theta_* \right\rangle = -\frac{1}{n}\sum_{i=1}^{n}\langle \nabla f(0, x_i), \theta_* \rangle$$

$$= \frac{1}{n}\sum_{i=1}^{n}\langle \nabla f(0, x_i) - \nabla f(\theta_*, x_i), 0 - \theta_* \rangle \geq \mu\|\theta_*\|^p,$$

(G.13)

where $p \in (1, 2)$, which implies that

$$\mu\|\theta_*\|^p \leq \sup_{x \in \mathcal{X}}\|\nabla f(0, x)\| \cdot \|\theta_*\|,$$

(G.14)

which yields that

$$\|\theta_*\| \leq \frac{1}{\mu^{\frac{1}{p-1}}}\sup_{x \in \mathcal{X}}\|\nabla f(0, x)\|^{\frac{1}{p-1}}.$$

Similarly, one can show that

$$\|\hat{\theta}_*\| \leq \frac{1}{\mu^{\frac{1}{p-1}}}\sup_{x \in \mathcal{X}}\|\nabla f(0, x)\|^{\frac{1}{p-1}}.$$

This completes the proof. $\qquad\square$

Now, we are able to state the main result for the Wasserstein algorithmic stability.

**Theorem G.1.** *Let $\theta_0 = \hat{\theta}_0 = \theta$. Suppose Assumption 4.1 and Assumption 4.2 hold and $\eta \leq \frac{\mu}{K_1^2 + 2^{p+4}D^2K_2^2}$ and $\sup_{x \in \mathcal{X}}\|x\| \leq D$ for some $D < \infty$ and $\sup_{x \in \mathcal{X}}\|\nabla f(0, x)\| \leq E$ for some $E < \infty$. Let $\nu_k$ and $\hat{\nu}_k$ denote the distributions of $\theta_k$ and $\hat{\theta}_k$ respectively. Then, we have*

$$\frac{1}{k}\sum_{i=1}^{k}\mathcal{W}_p^p(\nu_{i-1}, \hat{\nu}_{i-1})$$

$$\leq \frac{4D^2K_2^2\eta}{bn\mu}\cdot 2^{p+2}\cdot\left(\frac{2\theta^2 + 2(E/\mu)^{\frac{2}{p-1}}}{k\eta\mu} + \frac{8\eta}{\mu}D^2K_2^2\left(2^{p+1}(E/\mu)^{\frac{p}{p-1}} + 5\right)\right)$$

$$+ \frac{4D^2K_2^2\eta}{bn\mu}\cdot 2^{p+2}\left(2^{p+2}(E/\mu)^{\frac{p}{p-1}} + 10\right)$$

$$+ \frac{4DK_2}{n\mu}(1 + K_1\eta)\cdot 3\cdot 2^{p-1}\left(\frac{2\theta^2 + 2(E/\mu)^{\frac{2}{p-1}}}{k\eta\mu} + \frac{8\eta}{\mu}D^2K_2^2\left(2^{p+1}(E/\mu)^{\frac{p}{p-1}} + 5\right)\right)$$

$$+ \frac{4DK_2}{n\mu}(1 + K_1\eta)\cdot 7\cdot 2^{p-1}\left(\frac{2\theta^2 + 2(E/\mu)^{\frac{2}{p-1}}}{k\eta\mu} + \frac{8\eta}{\mu}D^2K_2^2\left(2^{p+1}(E/\mu)^{\frac{p}{p-1}} + 5\right)\right)$$

$$+ \frac{4DK_2}{n\mu}(1 + K_1\eta)\left(10\cdot 2^{p-1}(E/\mu)^{\frac{p}{p-1}} + 5\right).$$

(G.15)

*Proof.* Let us recall that $\theta_0 = \hat{\theta}_0 = \theta$ and for any $k \in \mathbb{N}$,

$$\theta_k = \theta_{k-1} - \frac{\eta}{b}\sum_{i \in \Omega_k}\nabla f(\theta_{k-1}, x_i),$$

(G.16)

$$\hat{\theta}_k = \hat{\theta}_{k-1} - \frac{\eta}{b}\sum_{i \in \Omega_k}\nabla f\left(\hat{\theta}_{k-1}, \hat{x}_i\right).$$

(G.17)

Thus it follows that

$$\theta_k - \hat{\theta}_k = \theta_{k-1} - \hat{\theta}_{k-1} - \frac{\eta}{b}\sum_{i \in \Omega_k}\left(\nabla f(\theta_{k-1}, x_i) - \nabla f\left(\hat{\theta}_{k-1}, x_i\right)\right) + \frac{\eta}{b}\mathcal{E}_k,$$

(G.18)

where

$$\mathcal{E}_k := \sum_{i \in \Omega_k} \left( \nabla f\left(\hat{\theta}_{k-1}, \hat{x}_i\right) - \nabla f\left(\hat{\theta}_{k-1}, x_i\right) \right). \tag{G.19}$$

This implies that

$$\left\| \theta_k - \hat{\theta}_k \right\|^2 = \left\| \theta_{k-1} - \hat{\theta}_{k-1} - \frac{\eta}{b} \sum_{i \in \Omega_k} \left( \nabla f(\theta_{k-1}, x_i) - \nabla f\left(\hat{\theta}_{k-1}, x_i\right) \right) \right\|^2 + \frac{\eta^2}{b^2} \|\mathcal{E}_k\|^2$$

$$+ 2 \left\langle \theta_{k-1} - \hat{\theta}_{k-1} - \frac{\eta}{b} \sum_{i \in \Omega_k} \left( \nabla f(\theta_{k-1}, x_i) - \nabla f\left(\hat{\theta}_{k-1}, x_i\right) \right), \frac{\eta}{b} \mathcal{E}_k \right\rangle. \tag{G.20}$$

By Assumption 4.2 and Assumption 4.1, we have

$$\left\| \theta_{k-1} - \hat{\theta}_{k-1} - \frac{\eta}{b} \sum_{i \in \Omega_k} \left( \nabla f(\theta_{k-1}, x_i) - \nabla f\left(\hat{\theta}_{k-1}, x_i\right) \right) \right\|^2$$

$$\leq \left\| \theta_{k-1} - \hat{\theta}_{k-1} \right\|^2 - 2\eta\mu \left\| \theta_{k-1} - \hat{\theta}_{k-1} \right\|^p + \frac{\eta^2}{b^2} \left( bK_1 \left\| \theta_{k-1} - \hat{\theta}_{k-1} \right\|^{\frac{p}{2}} \right)^2$$

$$= \left\| \theta_{k-1} - \hat{\theta}_{k-1} \right\|^2 - 2\eta\mu \left\| \theta_{k-1} - \hat{\theta}_{k-1} \right\|^p + \eta^2 K_1^2 \left\| \theta_{k-1} - \hat{\theta}_{k-1} \right\|^p$$

$$\leq \left\| \theta_{k-1} - \hat{\theta}_{k-1} \right\|^2 - \eta\mu \left\| \theta_{k-1} - \hat{\theta}_{k-1} \right\|^p, \tag{G.21}$$

provided that $\eta \leq \frac{\mu}{K_1^2}$.

Since $X_n$ and $\hat{X}_n$ differ by at most one element and $\sup_{x \in \mathcal{X}} \|x\| \leq D$ for some $D < \infty$, we have $x_i = \hat{x}_i$ for any $i \in \Omega_k$ with probability $\frac{n-b}{n}$ and $x_i \neq \hat{x}_i$ for exactly one $i \in \Omega_k$ with probability $\frac{b}{n}$ and therefore

$$\mathbb{E} \|\mathcal{E}_k\|^2 \leq \frac{b}{n} \mathbb{E} \left[ \left( K_2 2D(2\|\hat{\theta}_{k-1}\|^{p-1} + 1) \right)^2 \right]$$

$$\leq \frac{4D^2 K_2^2 b}{n} \left( 8\mathbb{E} \left[ \|\hat{\theta}_{k-1}\|^{2(p-1)} \right] + 2 \right)$$

$$\leq \frac{4D^2 K_2^2 b}{n} \left( 8\mathbb{E} \left[ \|\hat{\theta}_{k-1}\|^{p} \right] + 10 \right)$$

$$\leq \frac{4D^2 K_2^2 b}{n} \left( 2^{p+2} \mathbb{E} \|\hat{\theta}_{k-1} - \hat{\theta}_*\|^p + 2^{p+2} \|\hat{\theta}_*\|^p + 10 \right), \tag{G.22}$$

and moreover,

$$\mathbb{E} \left\langle \theta_{k-1} - \hat{\theta}_{k-1} - \frac{\eta}{b} \sum_{i \in \Omega_k} \left( \nabla f(\theta_{k-1}, x_i) - \nabla f\left(\hat{\theta}_{k-1}, x_i\right) \right), \frac{\eta}{b} \mathcal{E}_k \right\rangle$$

$$\leq \frac{\eta}{b} \frac{b}{n} \mathbb{E} \left[ \left( \left\| \theta_{k-1} - \hat{\theta}_{k-1} \right\| + K_1 \eta \left\| \theta_{k-1} - \hat{\theta}_{k-1} \right\|^{\frac{p}{2}} \right) K_2 2D \left( 2\|\hat{\theta}_{k-1}\|^{p-1} + 1 \right) \right]$$

$$\leq \frac{2DK_2\eta}{n} \mathbb{E} \left[ \left( (1 + K_1\eta) \left\| \theta_{k-1} - \hat{\theta}_{k-1} \right\| + K_1\eta \right) \left( 2\|\hat{\theta}_{k-1}\|^{p-1} + 1 \right) \right]$$

$$\leq \frac{2DK_2\eta}{n} (1 + K_1\eta) \mathbb{E} \left[ \left( \|\theta_{k-1}\| + \|\hat{\theta}_{k-1}\| + 1 \right) \left( 2\|\hat{\theta}_{k-1}\|^{p-1} + 1 \right) \right]. \tag{G.23}$$

Notice that for any $x, y \geq 0$ and $p \in (1, 2)$, we have $xy^{p-1} \leq x^p + y^p$, $y \leq y^p + 1$, $x \leq x^p + 1$ and $y^{p-1} \leq y^p + 1$, which implies that

$$(x + y + 1)(2y^{p-1} + 1) = 2xy^{p-1} + 2y^p + 2y^{p-1} + x + y + 1 \leq 3x^p + 7y^p + 5. \tag{G.24}$$

Therefore, by applying (G.24) to (G.23), we have

$$
\mathbb{E} \left\langle \theta_{k-1} - \hat{\theta}_{k-1} - \frac{\eta}{b} \sum_{i \in \Omega_k} \left( \nabla f(\theta_{k-1}, x_i) - \nabla f\left(\hat{\theta}_{k-1}, x_i\right) \right), \frac{\eta}{b}\mathcal{E}_k \right\rangle
$$

$$
\leq \frac{2DK_2\eta}{n}(1 + K_1\eta)\left( 3\mathbb{E}\|\theta_{k-1}\|^p + 7\mathbb{E}\|\hat{\theta}_{k-1}\|^p + 5 \right)
$$

$$
\leq \frac{2DK_2\eta}{n}(1 + K_1\eta)\Big( 3 \cdot 2^{p-1}\mathbb{E}\|\theta_{k-1} - \theta_*\|^p + 3 \cdot 2^{p-1}\|\theta_*\|^p
$$

$$
+ 7 \cdot 2^{p-1}\mathbb{E}\|\hat{\theta}_{k-1} - \hat{\theta}_*\|^p + 7 \cdot 2^{p-1}\|\hat{\theta}_*\|^p + 5 \Big). \qquad \text{(G.25)}
$$

Hence, by applying (G.21), (G.22), (G.25) into (G.20), we conclude that

$$
\mathbb{E}\left\|\theta_k - \hat{\theta}_k\right\|^2 \leq \mathbb{E}\left\|\theta_{k-1} - \hat{\theta}_{k-1}\right\|^2 - \eta\mu\mathbb{E}\left\|\theta_{k-1} - \hat{\theta}_{k-1}\right\|^p
$$

$$
+ \frac{4D^2K_2^2\eta^2}{bn}\left( 2^{p+2}\mathbb{E}\left\|\hat{\theta}_{k-1} - \hat{\theta}_*\right\|^p + 2^{p+2}\|\hat{\theta}_*\|^p + 10 \right)
$$

$$
+ \frac{4DK_2\eta}{n}(1 + K_1\eta)\Big( 3 \cdot 2^{p-1}\mathbb{E}\|\theta_{k-1} - \theta_*\|^p + 3 \cdot 2^{p-1}\|\theta_*\|^p
$$

$$
+ 7 \cdot 2^{p-1}\mathbb{E}\|\hat{\theta}_{k-1} - \hat{\theta}_*\|^p + 7 \cdot 2^{p-1}\|\hat{\theta}_*\|^p + 5 \Big),
$$

$$
\text{(G.26)}
$$

provided that $\eta \leq \frac{\mu}{K_1^2}$. This, together with $\theta_0 = \hat{\theta}_0 = \theta$, implies that

$$
\frac{1}{k}\sum_{i=1}^{k}\mathbb{E}\left\|\theta_{i-1} - \hat{\theta}_{i-1}\right\|^p
$$

$$
\leq \frac{4D^2K_2^2\eta}{bn\mu}\left( 2^{p+2}\frac{1}{k}\sum_{i=1}^{k}\mathbb{E}\left\|\hat{\theta}_{i-1} - \hat{\theta}_*\right\|^p + 2^{p+2}\|\hat{\theta}_*\|^p + 10 \right)
$$

$$
+ \frac{4DK_2}{n\mu}(1 + K_1\eta)\Big( 3 \cdot 2^{p-1}\frac{1}{k}\sum_{i=1}^{k}\mathbb{E}\|\theta_{i-1} - \theta_*\|^p + 3 \cdot 2^{p-1}\|\theta_*\|^p
$$

$$
+ 7 \cdot 2^{p-1}\frac{1}{k}\sum_{i=1}^{k}\mathbb{E}\|\hat{\theta}_{i-1} - \hat{\theta}_*\|^p + 7 \cdot 2^{p-1}\|\hat{\theta}_*\|^p + 5 \Big). \quad \text{(G.27)}
$$

In Lemma G.1, we showed that under the assumption $\eta \leq \frac{\mu}{K_1^2 + 2^{p+4}D^2K_2^2}$, we have

$$
\frac{1}{k}\sum_{i=1}^{k}\mathbb{E}\|\theta_{i-1} - \theta_*\|^p \leq \frac{\|\theta - \theta_*\|^2}{k\eta\mu} + \frac{8\eta}{\mu}D^2K_2^2\left( 2^{p+1}\|\theta_*\|^p + 5 \right), \qquad \text{(G.28)}
$$

$$
\frac{1}{k}\sum_{i=1}^{k}\mathbb{E}\left\|\hat{\theta}_{i-1} - \hat{\theta}_*\right\|^p \leq \frac{\left\|\theta - \hat{\theta}_*\right\|^2}{k\eta\mu} + \frac{8\eta}{\mu}D^2K_2^2\left( 2^{p+1}\|\hat{\theta}_*\|^p + 5 \right). \qquad \text{(G.29)}
$$

Hence, by plugging (G.28) and (G.29) into (G.27), we conclude that

$$
\frac{1}{k}\sum_{i=1}^{k}\mathbb{E}\left\|\theta_{i-1}-\hat{\theta}_{i-1}\right\|^{p}
$$

$$
\leq \frac{4D^{2}K_{2}^{2}\eta}{bn\mu}\left(2^{p+2}\left(\frac{\left\|\theta-\hat{\theta}_{*}\right\|^{2}}{k\eta\mu}+\frac{8\eta}{\mu}D^{2}K_{2}^{2}\left(2^{p+1}\|\hat{\theta}_{*}\|^{p}+5\right)\right)+2^{p+2}\|\hat{\theta}_{*}\|^{p}+10\right)
$$

$$
+\frac{4DK_{2}}{n\mu}(1+K_{1}\eta)\cdot3\cdot2^{p-1}\left(\frac{\|\theta-\theta_{*}\|^{2}}{k\eta\mu}+\frac{8\eta}{\mu}D^{2}K_{2}^{2}\left(2^{p+1}\|\theta_{*}\|^{p}+5\right)\right)
$$

$$
+\frac{4DK_{2}}{n\mu}(1+K_{1}\eta)\cdot7\cdot2^{p-1}\left(\frac{\left\|\theta-\hat{\theta}_{*}\right\|^{2}}{k\eta\mu}+\frac{8\eta}{\mu}D^{2}K_{2}^{2}\left(2^{p+1}\|\hat{\theta}_{*}\|^{p}+5\right)\right)
$$

$$
+\frac{4DK_{2}}{n\mu}(1+K_{1}\eta)\left(3\cdot2^{p-1}\|\theta_{*}\|^{p}+7\cdot2^{p-1}\|\hat{\theta}_{*}\|^{p}+5\right). \tag{G.30}
$$

Moreover, $\|\theta-\hat{\theta}_{*}\|^{2}\leq2\|\theta\|^{2}+2\|\hat{\theta}_{*}\|^{2}$, $\|\theta-\theta_{*}\|^{2}\leq2\|\theta\|^{2}+2\|\theta_{*}\|^{2}$ and by applying Lemma G.2, we obtain

$$
\frac{1}{k}\sum_{i=1}^{k}\mathbb{E}\left\|\theta_{i-1}-\hat{\theta}_{i-1}\right\|^{p}
$$

$$
\leq \frac{4D^{2}K_{2}^{2}\eta}{bn\mu}\left(2^{p+2}\left(\frac{2\theta2+2(E/\mu)^{\frac{2}{p-1}}}{k\eta\mu}+\frac{8\eta}{\mu}D^{2}K_{2}^{2}\left(2^{p+1}(E/\mu)^{\frac{p}{p-1}}+5\right)\right)\right.
$$

$$
\left.+2^{p+2}(E/\mu)^{\frac{p}{p-1}}+10\right)
$$

$$
+\frac{4DK_{2}}{n\mu}(1+K_{1}\eta)\cdot3\cdot2^{p-1}\left(\frac{2\theta2+2(E/\mu)^{\frac{2}{p-1}}}{k\eta\mu}+\frac{8\eta}{\mu}D^{2}K_{2}^{2}\left(2^{p+1}(E/\mu)^{\frac{p}{p-1}}+5\right)\right)
$$

$$
+\frac{4DK_{2}}{n\mu}(1+K_{1}\eta)\cdot7\cdot2^{p-1}\left(\frac{2\theta2+2(E/\mu)^{\frac{2}{p-1}}}{k\eta\mu}+\frac{8\eta}{\mu}D^{2}K_{2}^{2}\left(2^{p+1}(E/\mu)^{\frac{p}{p-1}}+5\right)\right)
$$

$$
+\frac{4DK_{2}}{n\mu}(1+K_{1}\eta)\left(10\cdot2^{p-1}(E/\mu)^{\frac{p}{p-1}}+5\right). \tag{G.31}
$$

Finally, by the definition of $p$-Wasserstein distance, we have

$$
\frac{1}{k}\sum_{i=1}^{k}\mathcal{W}_{p}^{p}(\nu_{i-1},\hat{\nu}_{i-1})\leq\frac{1}{k}\sum_{i=1}^{k}\mathbb{E}\left\|\theta_{i-1}-\theta_{*}\right\|^{p}. \tag{G.32}
$$

This completes the proof. $\qquad\square$

### G.1   Proof of Theorem 4.2

*Proof.* First, let us show that the sequences $\theta_{k}$ and $\hat{\theta}_{k}$ are ergodic.

First, let us show that the limit $\theta_{\infty}$, if exists, is unique. Consider two sequences $\theta_{k}^{(1)}$ and $\theta_{k}^{(2)}$ starting at $\theta_{0}^{(1)}$ and $\theta_{0}^{(2)}$ respectively with two limits $\theta_{\infty}^{(1)}$ and $\theta_{\infty}^{(2)}$. For any $k\in\mathbb{N}$,

$$
\theta_{k}^{(1)}=\theta_{k-1}^{(1)}-\frac{\eta}{b}\sum_{i\in\Omega_{k}}\nabla f\left(\theta_{k-1}^{(1)},x_{i}\right), \tag{G.33}
$$

$$
\theta_{k}^{(2)}=\theta_{k-1}^{(2)}-\frac{\eta}{b}\sum_{i\in\Omega_{k}}\nabla f\left(\theta_{k-1}^{(2)},x_{i}\right). \tag{G.34}
$$

By Assumption 4.2 and Assumption 4.1, we have

$$\left\| \theta_k^{(2)} - \theta_k^{(1)} \right\|^2 = \left\| \theta_{k-1}^{(2)} - \theta_{k-1}^{(1)} - \frac{\eta}{b} \sum_{i \in \Omega_k} \left( \nabla f \left( \theta_{k-1}^{(2)}, x_i \right) - \nabla f \left( \theta_{k-1}^{(1)}, x_i \right) \right) \right\|^2$$

$$\leq \left\| \theta_{k-1}^{(2)} - \theta_{k-1}^{(1)} \right\|^2 - 2\eta\mu \left\| \theta_{k-1}^{(2)} - \theta_{k-1}^{(1)} \right\|^p + \frac{\eta^2}{b^2} \left( bK_1 \left\| \theta_{k-1}^{(2)} - \theta_{k-1}^{(1)} \right\|^{\frac{p}{2}} \right)^2$$

$$= \left\| \theta_{k-1}^{(2)} - \theta_{k-1}^{(1)} \right\|^2 - 2\eta\mu \left\| \theta_{k-1}^{(2)} - \theta_{k-1}^{(1)} \right\|^p + \eta^2 K_1^2 \left\| \theta_{k-1}^{(2)} - \theta_{k-1}^{(1)} \right\|^p$$

$$\leq \left\| \theta_{k-1}^{(2)} - \theta_{k-1}^{(1)} \right\|^2 - \eta\mu \left\| \theta_{k-1}^{(2)} - \theta_{k-1}^{(1)} \right\|^p, \tag{G.35}$$

provided that $\eta \leq \frac{\mu}{K_1^2}$. This implies that

$$\left\| \theta_k^{(2)} - \theta_k^{(1)} \right\|^2 \leq \left\| \theta_{k-1}^{(2)} - \theta_{k-1}^{(1)} \right\|^2 - \eta\mu \left\| \theta_{k-1}^{(2)} - \theta_{k-1}^{(1)} \right\|^p, \tag{G.36}$$

provided that $\eta \leq \frac{\mu}{K_1^2}$. Thus, $\left\| \theta_k^{(2)} - \theta_k^{(1)} \right\|^2 \leq \left\| \theta_{k-1}^{(2)} - \theta_{k-1}^{(1)} \right\|^2$ for any $k \in \mathbb{N}$. Suppose $\left\| \theta_{j-1}^{(2)} - \theta_{j-1}^{(1)} \right\|^2 \geq 1$ for every $j = 1, 2, \ldots, k$, then we have

$$\left\| \theta_k^{(2)} - \theta_k^{(1)} \right\|^2 \leq \left\| \theta_0^{(2)} - \theta_0^{(1)} \right\|^2 - k\eta\mu, \tag{G.37}$$

such that

$$\left\| \theta_k^{(2)} - \theta_k^{(1)} \right\|^2 \leq 1, \qquad \text{for any } k \geq k_0 := \frac{\left\| \theta_0^{(2)} - \theta_0^{(1)} \right\|^2 - 1}{\eta\mu}. \tag{G.38}$$

Since $p \in (1, 2)$,

$$\left\| \theta_k^{(2)} - \theta_k^{(1)} \right\|^2 \leq (1 - \eta\mu) \left\| \theta_{k-1}^{(2)} - \theta_{k-1}^{(1)} \right\|^2, \tag{G.39}$$

for any $k \geq k_0 + 1$, which implies that $\theta_k^{(2)} - \theta_k^{(1)} \to 0$ as $k \to \infty$ so that $\theta_\infty^{(2)} = \theta_\infty^{(1)}$.

Next, let us show that for any sequence $\theta_k$, it converges to a limit. It follows from (G.39) that

$$\left\| \theta_k^{(2)} - \theta_k^{(1)} \right\|^2 \leq (1 - \eta\mu)^{k - \lceil k_0 \rceil} \left\| \theta_{\lceil k_0 \rceil}^{(2)} - \theta_{\lceil k_0 \rceil}^{(1)} \right\|^2, \tag{G.40}$$

for any $k \geq k_0 + 1$. Let $\theta_0^{(1)}$ be a fixed initial value in $\mathbb{R}^d$, and let $\theta_0^{(2)} = \theta_1^{(1)}$ which is random yet takes only finitely many values given $\theta_0^{(1)}$ so that $k_0$ is bounded. Therefore, it follows from (G.40) that

$$\mathcal{W}_2^2(\nu_{k+1}, \nu_k) \leq (1 - \eta\mu)^k \mathbb{E} \left[ (1 - \eta\mu)^{-\lceil k_0 \rceil} \left\| \theta_{\lceil k_0 \rceil}^{(2)} - \theta_{\lceil k_0 \rceil}^{(1)} \right\|^2 \right], \tag{G.41}$$

where $\nu_k$ denotes the distribution of $\theta_k$, which implies that

$$\sum_{k=1}^{\infty} \mathcal{W}_2^2(\nu_{k+1}, \nu_k) < \infty. \tag{G.42}$$

Thus, $(\nu_k)$ is a Cauchy sequence in $\mathcal{P}_2(\mathbb{R}^d)$ equipped with metric $\mathcal{W}_2$ and hence there exists some $\nu_\infty$ such that $\mathcal{W}_2(\nu_k, \nu_\infty) \to 0$ as $k \to \infty$.

Hence, we showed that the sequence $\theta_k$ is ergodic. Similarly, we can show that the sequence $\hat{\theta}_k$ is ergodic.

Finally, by ergodic theorem and Fatou's lemma, we have

$$\mathbb{E} \left\| \theta_\infty - \hat{\theta}_\infty \right\|^p = \mathbb{E} \left[ \lim_{k \to \infty} \frac{1}{k} \sum_{i=1}^k \left\| \theta_{i-1} - \hat{\theta}_{i-1} \right\|^p \right] \leq \limsup_{k \to \infty} \frac{1}{k} \sum_{i=1}^k \mathbb{E} \left\| \theta_{i-1} - \hat{\theta}_{i-1} \right\|^p. \tag{G.43}$$

We can then apply (G.31) from the proof of Theorem G.1 to obtain:

$$\mathcal{W}_p^p(\nu_\infty, \hat{\nu}_\infty) \leq \frac{C_2}{bn\mu} + \frac{C_3}{n},$$ 
(G.44)

where

$$C_2 := \frac{4D^2 K_2^2 \eta}{\mu} \left( 2^{p+2} \left( \frac{8\eta}{\mu} D^2 K_2^2 \left( 2^{p+1} (E/\mu)^{\frac{p}{p-1}} + 5 \right) \right) + 2^{p+2} (E/\mu)^{\frac{p}{p-1}} + 10 \right),$$ 
(G.45)

$$C_3 := \frac{32 D^3 K_2^3 \eta}{\mu^2} (1 + K_1 \eta) \cdot 10 \cdot 2^{p-1} \left( 2^{p+1} (E/\mu)^{\frac{p}{p-1}} + 5 \right)$$

$$+ \frac{4DK_2}{\mu} (1 + K_1 \eta) \left( 10 \cdot 2^{p-1} (E/\mu)^{\frac{p}{p-1}} + 5 \right).$$ 
(G.46)

This completes the proof. $\qquad\square$

