# OpenReview forum: "Uniform-in-Time Wasserstein Stability Bounds for (Noisy) Stochastic Gradient Descent"
_NeurIPS.cc/2023/Conference — NeurIPS 2023 poster_

### Official Review · Reviewer_9dAF · 2023-07-05

**Soundness:** 3 good
**Presentation:** 4 excellent
**Contribution:** 3 good
**Rating:** 7
**Confidence:** 3

**Summary:**

This paper provides new stability bounds for SGD: they rely on perturbation theory for Markov Chains and on the ergodicity of SGD.

**Strengths:**

- Well written, well presented paper.
- New connection between the theory for Markov Chain and generalization of learning algorithms.
- First (to my knowledge) uniform in time stability bounds.
- Extensions and limits are properly discussed.

**Weaknesses:**

N/A

**Questions:**

- On the fact that most of the theory requires $\ell$ to be Lispchitz-continuous: would it be because this work is only considering the 1-Waserstein distance? With the 2-Wasserstein, maybe this technique would be able to handle $L$-smooth loss, what do you think? In particular, could you leverage Thm. 4.2 in that case?

---

> ### Author Rebuttal · Authors · 2023-08-09
>
> Thanks for a careful reading of our paper and finding that our paper is well-written with some new results on the connections between the theory for Markov Chain and generalization of learning algorithms.
>
> Thanks for the excellent question. In our paper, most of the theory is derived using the tool of the Wasserstein perturbation method developed in [RS18], and using this particular tool, 1-Wasserstein algorithmic stability bound is obtained. In order to obtain the generalization error bound from the 1-Wasserstein algorithmic stability bound, one has to assume that the surrogate loss is Lipschitz [RRT+16]. However, some of the results in this paper do not rely on the Wasserstein perturbation method [RS18], such as Theorem 4.2. For these results, we are able to obtain algorithmic stability in $p$-Wasserstein for some $p>1$. In particular, with the 2-Wasserstein, we are indeed able to handle smooth loss functions if we also establish uniform L^2 bounds following the approach in [RRT17]; see also [FR21]. We will add these discussions in the revised version of the paper.

---

> > ### Comment · Reviewer_9dAF · 2023-08-14
> > **Response to authors**
> >
> > Thank you for your answer !
> > For the sake of the discussion with the other reviewers, let me further detail my opinion on the paper:
> > - I agree with the other reviewers that the strongly convex case is not novel, and that the worse dependency on $\mu$ is a bit problematic.
> > - However, the proof technique is new, as well as the other results, and I believe it can be extended by future works.

---

> > > ### Author Response · Authors · 2023-08-18
> > >
> > > Thank you very much for going over our rebuttal.

---

### Official Review · Reviewer_Wt2h · 2023-07-06

**Soundness:** 3 good
**Presentation:** 3 good
**Contribution:** 2 fair
**Rating:** 6
**Confidence:** 2

**Summary:**

This paper studies the algorithmic stability of SGD in order to bound its expected generalization error when using Lipschitz losses. To do so, they consider Wasserstein stability instead of the standard uniform stability, which is defined using the dual representation of the Wasserstein distance and the representation of the generalization error from [HRS16].

To bound the Wasserstein stability the paper employs Markov chain perturbation theory.  The main idea to do so is to follow three steps: (i) showing that the optimizer is geometrically ergodic, which essentially boils down to showing that the Wasserstein distance of the transition kernels can be bounded from above by the distance of the inputs of said kernels; (ii) obtaining a Lyapunov function for the optimizer and the loss, which gives a sense of stability to the system (at least intuitively, as it is a necessary and sufficient condition for stability for certain classes of ODEs); and (iii) bounding the Wasserstein distance between the transition kernels of the optimizer with the same initialization but using two neighbouring datasets.

Using this three-step process they start giving a bound for the quadratic case with rate $O(\eta / n)$. Similarly, under strong convexity and some pseudo-Lipschitzness assumption on the gradients they show a bound with rate $O(1/n)$ with an appropriate choice of the learning rate. To deal with non-convex cases, they show that if the loss dissipates with Assumption 3.3 (which seems like a relaxed version of strong convexity), then adding a zero-mean noise with bounded second moment is sufficient to obtain a bound with rate $O(b/n)$, where $b$ is the batch size. This bound of course includes SGLD, but is not restricted to this case.

The addition of the noise in the non-convex case was to make sure that the ergodicity condition in step (i) was provable. The paper also studies what happens when noise is not included. To deal with this, still assuming that the loss dissipates with Assumption 3.3 they can bound the squared Wasserstein distance of order 2. This way, they can obtain bounds with rate $O(1/\sqrt{bn} + c)$, where $c$ is some constant. Therefore, the exclusion of noise in the analysis generates an extra bias term.

All the bounds up to here are uniform in time, that is, they hold for every step of the optimization process.

Fortunately, assuming that the loss $f$ is $\mu$ strongly-ish convex, in the sense that $\langle \nabla f(\theta, x) - \nabla f(\vartheta, x) , \theta - \vartheta \rangle \geq \mu \lVert \theta - \vartheta \rVert^p$ for some $p \in (1,2)$; and that the gradients Lipschitz-ish in the sense that $\lVert \nabla f (\theta, x) - \nabla (\vartheta, y) \rVert \leq K\_1 \lVert \theta - \vartheta \rVert^{p / 2} + K\_2 \lVert x - y \rVert ( \lVert \theta \rVert^{p-1} + \lVert \vartheta \rVert^{p-1} + 1)$; then they can show asymptotic generalization bounds with rate $O(1/n^{1/p})$.


**Strengths:**

The paper is very well written and "easy" to follow despite its technical difficulty, especially the proofs. I have to congratulate the authors for making difficult proofs look simple with their explanations.

I believe that studying the stability of algorithms in terms of the stability of the final transition kernels in terms of the Wasserstein distance is novel (although similar bounds on the generalization error are given in [RGBTS21, Theorem 2] in a different context). This is certainly interesting as not only gives some new results but also new strategies to tackle the problem. These new strategies can be useful to build upon for other researchers in the future.

The bounds for the strongly-convex functions and those for non-convex losses with additive noise pass the "eye test", as they achieve the expected rates with $n$, although some of the other parameters are a bit unclear (see weaknesses).

The bounds for non-convex losses with additive noise with bounded variance are interesting, as they hold for noises other than Gaussian. This can be helpful either (i) to construct SGLD-like algorithms that work better in practice or (ii) to study more general situations in which one could show that the gradient noise in GD actually behaves like an additive noise with bounded variance (see e.g. [WHXHBZ20]).

**Additional references**

Borja Rodríguez-Gálvez, Germán Bassi, Ragnar Thobaben, and Mikael Skoglund. "Tighter expected generalization error bounds via Wasserstein distance". NeurIPS 2021.

Jingfeng Wu, Wenqing Hu, Haoyi Xiong, Jun Huan, Vladimir Braverman, and Zhanxing Zhu. "On the Noisy Gradient Descent that Generalizes as SGD". ICML 2020.

**Weaknesses:**

* A criticism of the paper could be that the results from Section 3 recover the rate of previous results with respect to $n$, while having sometimes worse rate with other parameters. This is actually not very important in my opinion, as I value more the new techniques and the fact that this new analysis can be employed in different settings to achieve the desired rate.

* Something more concerning to me are some of the assumptions and parameters in the results. These are sometimes hard for me to interpret and contextualize. For example:
  * Assumption 3.1. The authors explain that some problems such as GLMs satisfy this condition [Bac14]. Which other problems satisfy the condition? For example, the Lipschitz constant of gradients in neural networks can be very large (or even infinite in some situations), yielding bounds that depend on it essentially vacuous. Is this similar to this pseudo-Lipschitzness condition? How does it behave in familiar scenarios? This can be accompanied by a simplification of Theorem 3.2 in some familiar settings.
* Assumption 3.3. This assumption seems to me as essentially a strong-convexity assumption with some slack. While this makes the problem more general than a strongly convex problem, it is unclear to me how much more general it is. I would have liked more intuition and explanation around this assumption other than just saying that is common in stochastic analysis [RRT17, GGZZ22] and that has been considered in neural network settings [AS22]. Could you give some examples of familiar settings where this assumption holds, and give an intuitive explanation of how much more general than strong convexity this is?
* Theorem 3.3. Some of the elements such as the parameter $M$ and $\bar{\eta}$ are hard to into context and in general the bound is hard to interpret. Even though the authors make an effort to clarify that in Remark 3.4 and Corollary 3.5, it is still difficult to understand well the result in terms of these parameters.
* Theorem 4.1. How big is the bias term in certain familiar scenarios? Could it be that his term is small enough for us not to care in some settings?
* Assumptions 4.1 and 4.2. Could you please give some examples of losses and settings where these assumptions are satisfied? This could help contextualize the results.

While I already like the paper and its ideas as it is, without a better understanding of these issues, it is hard for me to assess the actual contribution of the presented results. I am happy to increase my score if they are addressed.







**Questions:**

My questions mainly concern the assumptions employed throughout the text and some of the parameters appearing in the results. These questions are mainly outlined in the weaknesses, but to summarize:
* Could you give more intuitions on the assumptions that are employed (e.g. how much more general Assumption 3.3. is with respect to strong convexity)?
* Could you give examples of losses and situations where these assumptions are satisfied so we can better contextualize the results?
* Similarly, could you give some examples of the parameters (or some order of magnitude empirically maybe) that appear in the results?



**Limitations:**

Yes, the authors discuss some of the limitations of the paper through the text.

Some of the limitations regarding other assumptions could be clarified in the text.

---

> ### Author Rebuttal · Authors · 2023-08-09
>
> Thanks for your time invested in our paper. Below are our responses to your questions:
>
> **Weaknesses**:
>
> 1. Weaker result with respect to other parameters: We get an optimal rate with respect to the number of samples. But dependence on other parameters might not be optimal  in some results, for instance dependency to the strong convexity constant. However, our result is more general with respect to existing SGD stability results. Our bound on expected generalization error applies to an entire class of Lipschitz loss functions, not limited to a single one. However, the existing stability results for SGD that we are aware of hold typically for a single loss function at a time and do not apply to a class of functions. Also, as the referee mentioned, the value of our analysis technique is that it can be employed in different settings.
>
> 2. Assumption 3.1 : Assumption 3.1 is more general than the smoothness (Lipschitzness of the gradients of $f$) assumption. In fact, for $K_2 =0$, the assumption becomes a smoothness assumption on $f$ in parameter $\theta$. Substituting $K_2=0$ in our results provides us the bound for smooth $f$. For example, regularized logistic regression with bounded data will fall into the $K_2 = 0$ case. We will add a remark about this during the revision.  We conclude that, allowing $K_2 >0$ in Assumption 3.1. allows us to cover a larger class. Therefore, we interpret Assumption 3.1 as a more general version of smoothness assumption. As a concrete example, consider the loss $f(x,\theta) = | \theta^T x - b|^p$ with $p\in (1,2)$ which arises in robust regression. The gradient of $f$ with respect to $\theta$ is Holder with Holder constant $p-1 \in (0,1)$, and is not Lipschitz but it will be pseudo-Lipschitz.
>
> We agree with the referee that Lipschitz constants in neural networks can be very large (assuming the norms of the weights are bounded), in this case our constants $K_1$ and $K_2$ in Assumption 3.1 will also be unfortunately large. However, if weight regularization is used; i.e. if a quadratic penalty term is added to the training loss then by a Lagrangian reformulation, this will be equivalent to training with norm constraints on the weights, and this will allow us to control the Lipschitz constants.
>
> 3. Assumption 3.3: The class of dissipative functions we consider are the ones that admit some gradient growth in radial directions outside a compact set. Inside the compact set though, they can have quite general non-convexity patterns. As concrete examples, they include certain one-hidden-layer neural networks (arXiv:2205.14818), they arise in non-convex formulations of classification problems; for instance in logistic regression with a sigmoid/non-convex link function. They can also arise in robust regression problems, see e.g. [X Gao, M Gürbüzbalaban, L Zhu - Operations Research, 2022]. Also, any function $f$ that is strongly convex outside of a ball of radius $R$ for some $R >0 $ will satisfy this assumption. Consequently, regularized regression problems where the loss is a strongly convex quadratic plus a smooth penalty that grows slower than a quadratic will belong to this class; a concrete example would be smoothed Lasso regression. Many other examples are also given in arxiv:2007.11612v3). Dissipative functions also arise frequently in the sampling and Bayesian learning literature, for instance in the study stochastic gradient Langevin dynamics (SGLD), this assumption is used to show that a  stationary distribution exists (see e.g. arxiv:1702.03849)
>
> 4. Theorem 3.3 : Thanks for the comment. In the revised version, we will add further explanations to discuss these parameters and how they affect the main results of the paper. For example, the parameter $\bar{\eta}$ appears in the upper bound in equation (3.9) that controls the 1-Wasserstein algorithmic stability of the SGD. It is easy to see from equation (3.9) that the smaller $\bar{\eta}$, the smaller the 1-Wasserstein bound. By the definition of $\bar{\eta}$, the larger $\hat{\eta}$, the smaller the 1-Wasserstein bound. As a result, we would like to choose $\hat{\eta}$ to be as large as possible, and the equation (3.10) provides an explicit value that $\hat{\eta}$ can take (which is already as large as possible from our theory). The parameter $M$ can be interpreted in a similar way. We will add more discussions and intuitive explanations in the revised version of the paper.
>
> 5. Theorem 4.1 :   If the loss is close to being strongly convex, then the bias term $K/m$ will be small. For example, consider a loss which is $\mu$-strongly convex outside a ball of radius $R$, and inside this ball of radius $R$, it can be non-convex with a bounded Hessian. As the size of the ball radius $R$ gets smaller, we have $K\to 0$ and $\mu\to m$ so that $K/m$ will go to zero as $R$ to zero. A simple toy example to visualize this would be the double-well example (see Example 1 in [X Gao, M Gürbüzbalaban, L Zhu - Operations Research, 2022]). That being said, we admit that such losses are not common in the statistical learning literature and by simply adding Gaussian noise to the iterations, we can analyze more general non-convex losses as we detail in Section 3.3.
>
> 6. Assumptions 4.1, 4.2 : Even though Assumptions 4.1 and 4.2 have been used in prior theoretical studies as we cited in the paper, after your question, we now realized that unfortunately they did not provide any concrete examples that satisfy these assumptions. Nevertheless, as an example, we suspect that the loss f(\theta, x) = |a^T \theta - b|^p satisfies the assumption where $p \in (1,2)$ and $x=(a,b)$ is the input-output pair. We could not formally prove this in the limited rebuttal period; however, we will try our best to come up with concrete examples for these assumptions. We agree with you that such examples would play an important role in terms of appreciation of the results, and we would like to thank you for bringing this to our attention.

---

> > ### Comment · Reviewer_Wt2h · 2023-08-14
> > **Answer to rebuttal**
> >
> > Thank you for your rebuttal.
> >
> > I liked that you went over all my questions and comments, especially regarding interpretation and examples. Please, if possible, make sure to incorporate all of them into the main text of the paper. I think it improves it.
> >
> > Also, if you end up proving that $| a^T \theta - b|^p$ satisfies assumptions 4.1, 4.2 for $p \in (1,2)$, please respond to this answer as I am curious to see that.

---

> > > ### Comment · Reviewer_Wt2h · 2023-08-18
> > > **Score update**
> > >
> > > Dear authors,
> > >
> > > We can now update the scores. I increased it.

---

> > > > ### Author Response · Authors · 2023-08-18
> > > >
> > > > Thank you very much for going over our rebuttal and increasing your score.
> > > >
> > > > We will detail here whenever we have progress finding examples satisfying Assumptions 4.1 and 4.2.

---

### Official Review · Reviewer_RR88 · 2023-07-10

**Soundness:** 3 good
**Presentation:** 3 good
**Contribution:** 3 good
**Rating:** 7
**Confidence:** 3

**Summary:**

In this paper, the authors studied an approach to obtain generalization bounds on SGD algorithm under different class of objective functions such as "quadratic strongly convex", "smooth strongly convex" and one subclass of non-necessarily convex functions described in Assumption 3.3.

Interestingly they convey the message that, as in the strongly convex settings, they can provide generalization bounds in the non convex setting as soon as they add some additive noise to the SGD update.

Their approach is based on stability analysis, together with convergence properties of Markov chains.



**Strengths:**

- Mixing stability and Markov chain perturbation theory is a very interesting approach that I had never seen before. The technique might be reused in other settings.
- The trick of adding noise to create generalization and make it work thanks to Markov chain theory is quite beautiful.

**Weaknesses:**

- Authors claim "non convex" everywhere without more precision. I first thought of conditions like hypoconvexity. Some people study smooth optimization without any other assumption. Here the setting is very specific and authors should be more precise about it in introduction. One cannot claim solving problems in "non-convex" setting just as soon as its class contains one non-convex function.
- It is actually quite natural to believe that adding noise reducing generalization error as the dependency in data is reduced (same argument for differential privacy). But we can do it with an algorithm that simply returns a constant independently of the training data. The interesting thing to do is to reduce generalization error while preserving good optimization properties. In this paper, the authors are degrading optimization performance as a constant step-size eta leads to convergence to neighborhood of the solution, and adding noise leads to a larger size of this neighborhood. This brings two questions:
    - Can we generalize this analysis to decreasing step size as the ergocity would not be geometric anymore?
    - In case of constant step-size, can authors discuss the values of « optimization error + generalization error » terms depending on the additive noise variance?

- l.291: « This indicates that it is necessary to add additional noise » -> No! While this section can serve as a good intuition of why additive noise what essential, this is not a clear proof of this necessity. Authors showed they found some weaker upper bound without additive noise. They did not show one cannot find a tighter one.

- Notation suggestions:
    - The word « algorithm » is never defined before being used in Definition 2.5. Since in definition 2.1, $X$ and $\hat{X}$ are not stochastic (instead we optimize on their values), it is even more important to stress here that an algorithm is indeed stochastic and that the expectation here is taken over this source of randomness conditioned by the values of $X$, $\hat{X}$ and $z$. However, the notation $\mathcal{A}: \bigcup_{n=1}^{\infty}\mathcal{X}^n \rightarrow \mathbb{R}^d$ means that $\mathcal{A}$ is deterministic. So either $\mathcal{A}$ returns a random variable, or it takes an additional random variable as input.
    - In 2.5, I find weird the fact to associate $\nu$ and $\hat{\nu}$ to $X$ and $\hat{X}$ as if those two datasets were particular, while we optimize over all the possible datasets. So $\nu$ and $\hat{\nu}$ vary also. I would rather defined a systematic mapping $\nu: X \rightarrow \nu_X$ so that this distribution is defined for all $X$. Or the variables over which the supremum is computed should be $\nu$ and $\hat{\nu}$ that are neighbors to each other. In the latter case, this should come with the definition that two distributions are neighbors iif there exist two neighbor datasets they are generated from.

Minor / typos:

l.133: « $R(\omega)$ » -> $R(\theta)$

l.229: I guess $\xi_k$ is also independent with $\theta_{k-1}$ and $\Omega_k$? Maybe specify it.

l252: « our result in » -> our result is

l237: « Thereom » -> Theorem


**Questions:**

- Just to be sure I understood the proof correctly, can authors confirm the only place the neighborhood assumption between datasets is used is in the expression of $\gamma$ in equation 2.9.  If this is the case, I think this little intuition should be explicit in the paper.
- I am surprised to see in lemma 3.3 that the bound on $\gamma$ does not depend on the batch size. Indeed, there is a certain probability that the batch does not contain the data that differs from one dataset to the other, in which case the step $P$ and $\hat{P}$ are identical. Then, the error is 0. The average error should be the error in the case that the different data is in the batch, times this probability, the latter being proportional to the batch size.
- The dependency on $\sigma^2$ is unclear. In Th D1, the bound seems increasing in $\sigma^2$, then removing the additive noise seems hurtless. However, looking closer, one notices that $\psi$ depends on $K_0$ that itself depends on $\sigma^2$. Why cannot we still consider $\sigma^2=0$? The only thing we have to verify is that $\psi$ will not explode, i.e. $K_0>0$. But $K_0$'s expression is a complicated mix between many other variables. Hence it is hard to understand what happens when $\sigma^2=0$. Can you elucidate this question?


**Limitations:**

Not much except maybe it applies to algorithms that contracts, i.e. SGD with constant step-size that cannot converge to the optimum.  It would be nice to handle decreasing step-sizes.

---

> ### Author Rebuttal · Authors · 2023-08-09
>
> We thank the reviewer for their careful reading of our paper. Below are responses to your questions:
>
> **Weaknesses**:
>
> * As the referee pointed out, our analysis applies to non-convex losses that satisfy some conditions (a dissipativity condition and a pseudo-Lipschitz gradient growth condition). As suggested, we will clarify this in our introduction further, emphasizing the assumptions we make.
>
>
>
> * Our approach mainly relies on the Wasserstein perturbation method developed in [RS18], which requires that the Markov chain is time-homogeneous. When the step-size is decreasing, the associated Markov chain is no longer time-homogeneous and thus [RS18] is no longer applicable. Therefore, the regime with decreasing step-size is out of the scope of the current paper. However, we do believe that it is possible to extend the analysis of [RS18] to deal with time-inhomogeneous Markov chains. But that will require us to write a paper that first extends the analysis of [RS18] and then apply it to the context of decreasing step-size. That will be left as a future research direction which will be interesting to pursue.
>
> * In [RRT17], the authors provided analysis for the optimization error depending on the additive noise variance, later on these results were extended to different settings. We can directly combine our results with their results to obtain an optimization error + generalization error bound. We will mention this in the next version of the paper.
>
> * In line 291, what we meant to say was that for our proof technique to work well, adding noise was beneficial. We absolutely agree with the referee that this does not mean that additive noise is a must for an algorithm to perform/generalize better.
>
> * Thanks for the suggestions on the notations. The notation $\mathcal{A}$ does not mean that it is deterministic, even though it is a function of the dataset, which is deterministic. The extra randomness comes from the randomness in the SGD, not the dataset. We will make this more transparent in the revised version of the paper.
> Indeed, the notation $\nu_{X}$, $\nu_{\hat{X}}$ might be better than the notation $\nu$ and $\hat{\nu}$ that we currently use. However, since the upper bound we eventually obtain does not rely on the particular dependence on $X$ and $\hat{X}$, but only on the fact that they differ by only one element and data points are bounded with radius $D$, so that there are factors $1/n$ and $D$ that appear in the bound, independent of $X$ and $\hat{X}$, our notations as they currently are will not cause a misunderstanding. But we do agree that more formally, it is better to use the notation $\nu_{X}$, $\nu_{\hat{X}}$.
>
> We will fix the issues mentioned in the minor comments, indeed these are typographical errors.
>
>
> **Questions:**
>
> * Thanks for this excellent observation. Indeed, the only place the neighborhood assumption between datasets is used is in the expression of $\gamma$ in equation (2.9). Lemma 2.1 ([RS18], Theorem 3.1) relies on three conditions, the Wasserstein construction in (2.7), the drift condition for the Lyapunov function in (2.8) and finally the estimate on $\gamma$ in equation (2.9) which is about the one-step 1-Wasserstein distance between two semi-groups that in our context are associated with two datasets that differ by at most one element. As a result, the only place the neighborhood assumption between datasets is used is in the expression of $\gamma$ in equation (2.9). We will add some discussions to make this little intuition more explicit in the revised version of the paper.
>
> * Thanks for the comment. Indeed, the probability you mentioned is proportional to the batch-size $b$. However, do not forget that there is the term $\eta/b$ in (B.9) and (B.10) in the proof of Lemma 3.3., and here $b$ comes from the definition of the SGD, where the factor $1/b$ helps the stochastic gradient become an unbiased estimator of the full gradient. Combining these two effects, you will obtain a bound on $\gamma$ which is independent of the batch-size $b$.
>
> * Indeed, it is very natural to ask if one can let $\sigma^{2}=0$ so that one can remove the additional noise added to the SGD in the case of non-convex loss functions. Unfortunately, the answer is no. Note that for the sake of notational convenience, we let $\sigma^2=\mathbb{E}\Vert\xi_1\Vert^{2}$ because the term $\mathbb{E}\Vert\xi_1\Vert^{2}$ appears many times in the bounds that we derived and it is convenient to use the notation $\sigma^{2}$. On the other hand, the probability density function $p(x)$ of $\xi_{1}$ implicitly depends on $\sigma^{2}$. If you let $\sigma^{2}\rightarrow 0$, $p(x)$ will converge to the Dirac delta distribution concentrated at $x=0$. When $\sigma^{2}\rightarrow 0$, there will be no mixing since $p(x)$ will converge to the Dirac delta distribution concentrated at $x=0$ such that in equation (3.8), $\hat{\eta}$ will have to go to zero. If $\hat{\eta}$ goes to zero, by the definition of $\bar{\eta}$, we will have $\bar{\eta}$ goes to $1$, which will make the upper bound in equation (3.9) explode. Therefore, to conclude, we cannot let $\sigma^{2}=0$ because otherwise, the upper bound in equation (3.9) will diverge and become a trivial upper bound.

---

> > ### Comment · Reviewer_RR88 · 2023-08-18
> > **Answer to rebuttal**
> >
> > First of all, I thank the authors for their detailed response.
> >
> > My main concern remains the combination of optimization and generalization bounds. SGD with constant step-size does not converge, and the final error depends on the noise. Adding an artificial noise increases this error. And authors decreases the generalization error. But it is quite natural to understand that a more random algorithm generalizes better. We need to compare the sum of those 2 bounds to conclude on the result.
> > Authors say "We can directly combine our results with their results to obtain an optimization error + generalization error bound. We will mention this in the next version of the paper.". I believe the authors that we can do it easily, and I am glad authors consider doing it in the revised version.
> > Can they quickly do it here by answering this message? I think this is an important point!
> >
> > Except this, I mostly like this paper due this novel approach, whereas it indeed is not a good advertisement for the technique to have some worse results than the known ones in "simple" cases, as pointed by several reviewers.
> > A couple of minor remarks:
> >
> > - "In line 291, what we meant to say was that for our proof technique to work well, adding noise was beneficial. We absolutely agree with the referee that this does not mean that additive noise is a must for an algorithm to perform/generalize better.": I understood you point. Mine is just to say that it should be phrased differently.
> >
> > - "The notation A does not mean that it is deterministic, even though it is a function of the dataset, which is deterministic. The extra randomness comes from the randomness in the SGD, not the dataset.". Yes it does! A function is by definition deterministic. It must return the same output each time it receives the same input. The notion of randomness necessitate the definition of measured spaces and a function between them called random variable. When we talk about a "random function", we generally refer to as a random variable which outputs a function, i.e. its arrival space is a measured space of functions. So we can either write "there are lots of functions $A_X: dataset \rightarrow R^d$, and we draw X at random, or equivalently, we can define a single function and place $X$ as second input, i.e. $A(X, dataset) \in R^d$, or say that $A$ takes a dataset in input and returns a random variable that itself returns a point in $R^d$. It only depends which signature you prefer for A, and I guess the most natural one is $A: (dataset, random sampling) \rightarrow R^d$. But this random sampling must be explicit in the signature of A, otherwise A is meant to be deterministic.
> >
> >
> > In conclusion, it would be great if authors could
> > - acknowledge they will rephrase l.291 (it was not clear in their rebuttal)
> > - acknowledge they will modify the notation for the signature of A and the $\nu$ that were discussed in the same point of their rebuttal.
> > - provide here a complete optimization + generalization bound in the non convex case to compare with previously known ones.
> >
> > Best regards.

---

> > > ### Author Response · Authors · 2023-08-19
> > >
> > > Thank you very much going over our rebuttal.
> > >
> > > We do acknowledge that we will rephrase l.291 and modify the notations and definitions for $A$ and $\nu$.
> > >
> > > Regarding the last point, we can give the following example. Recently, there have been multiple results on optimizing non-convex function using stochastic gradient Langevin dynamics. We will state the optimization error result from the paper “Global Convergence of Langevin Dynamics Based Algorithms for Nonconvex Optimization, Xu et al.”
> > >
> > > In theorem 3.3 of the above mentioned work, it was shown that gradient Langevin dynamics has the following optimization error after K iterations,
> > >
> > > $$ \text{Optimization error}(K) \leq \Theta e^{-\lambda K \eta} + \frac{C_{\psi}\eta}{\beta} + R_M$$
> > >
> > > where $\lambda, \Theta, C_{\psi} \beta$ and $R_M$ are problem dependent parameter described in the paper. Similar results are given for stochastic gradient Langevin dynamics in Theorem 3.6. A simplified version of the results for GLD and SGLD are presented in Corollary 3.4 and 3.7 respectively. According to Corollary 3.4, the optimization error for gradient Langevin is of $O(\varepsilon)$ after $K = O(d \varepsilon ^{-1} \lambda^{-1} \log 1/\varepsilon)$ where $\lambda$ is the uniform spectral gap for continuous-time Markov process generated by Langevin dynamics. According to Corollary 3.7, the optimization error is of $O(\varepsilon + d^{3/2}B^{-1/4} \lambda^{-1} \log 1/\varepsilon )$ after $K = O(d \varepsilon ^{-1} \lambda^{-1} \log 1/\varepsilon)$ where B is the mini-batch size of Algorithm. Combining these two results will directly give the optimization+generalization performance of SGLD on a non-convex function with dissipative loss.
> > >
> > > We will add this example in a more formal way to the appendix in the next version. We hope that this clarifies the concern.

---

> > > > ### Comment · Reviewer_RR88 · 2023-08-21
> > > > **Thank you for your answer**
> > > >
> > > > Dear authors,
> > > >
> > > > I thank you for your answer.
> > > > I think this analysis would indeed complete your results.
> > > > Note that in the paper you provided, it is hard however to make asymptotic analysis, like wondering what happens when $\beta\rightarrow\infty$ since $\rho_\beta$ is not explicit.
> > > >
> > > > Best regards.

---

### Official Review · Reviewer_yPZ3 · 2023-07-12

**Soundness:** 3 good
**Presentation:** 3 good
**Contribution:** 3 good
**Rating:** 6
**Confidence:** 4

**Summary:**

This paper studies the generalization bounds of (noisy) SGD via Wasserstein stability. The paper presents a unified guideline to derive the Wasserstein stability for stochastic optimization with a constant step size, which allows to derive stability bounds with a three-step proof technique: showing the optimizer is geometrically ergodic, obtaining a Lyapunov function for the optimizer and the loss, and bounding the discrepancy between the Markov transition kernels associated with the chains. With this guideline, the paper derives stability bounds for SGD with strongly convex problems, nonconvex problems and a class between convex and strongly convex functions. The paper also develops stability bounds for noisy SGD in a nonconvex case.

**Strengths:**

The paper presents a new perspective to derive stability bounds by techniques in applied probability. This connection between machine learning theory and applied probability is interesting and can have potential applications to various optimization algorithms.

The paper develops time-uniform stability bounds, meaning that the stability bounds would not increase to infinity as the number of iterations goes to infinity. The analysis also applies to general additive noise, which extends the existing analysis developed for Gaussian noise.

The paper also develops stability bounds for the standard SGD in the nonconvex case, although a dissipativity assumption is required.

**Weaknesses:**

The analysis for SGD in Section 3.3 is a bit complicated. For example, the choice of $\hat{\eta}$ in Eq (3.10) is too complex. The result is not quite intuitive and it is not clear how we can use this bound to explain the behavior of SGD.

The bound in Theorem 3.2 is of the order larger than $O(1/(n\mu^5))$. This bound has a crude dependency on $\mu$. Since $\mu$ is often very small in practice, the bound is worse than the existing bound of the order $O(1/(n\mu))$ (HRS16).

The bound in Theorem 4.1 is vacuous due to the term $K/m$. From this stability bound, we cannot get meaningful generalization bounds.


**Questions:**

Lemma 2.1 requires to build a Lyapunov function. The paper choose the function $V(\theta)=1+|\theta-\theta_*|^2$. Can you explain why this is a good choice of Lyapunov function? What is the intuition behind it? What is the basic principle to build a Lyapunov function?

The paper considers noisy SGD for nonconvex case. Can you explain intuitively why noise is needed in this case? What is the benefit of adding noise in the nonconvex case?

**Minor comments**:

Eq (2.6): should $\mathbb{P}(\theta_{n-1},A)$ be $P(\theta_{n-1},A)$?

Eq (2.8): the meaning of $\hat{P}\hat{V}$ is not given. The meaning is given in the appendix.

Below Assumption A2: "that that"

Eq (D.21): $\theta - \hat{\theta}_*+...$

 should be $ \theta - \hat{\theta}_*-...$.

The same change should be also made in Eq (D.22)

**Limitations:**

I do not see negative societal impact.

---

> ### Author Rebuttal · Authors · 2023-08-09
>
> We thank the reviewer for their careful reading of our paper. Below are responses to your questions:
>
> **Weaknesses:**
>
> * We will add an intuitive explanation behind formula (3.10) and its implications for SGD. The parameter $\hat{\eta}$ appears in the definition of $\bar{\eta}$ that appears in the upper bound in equation (3.9) that controls the 1-Wasserstein algorithmic stability of the SGD. It is easy to see from equation (3.9) that the smaller $\bar{\eta}$, the smaller the 1-Wasserstein bound. By the definition of $\bar{\eta}$, the larger $\hat{\eta}$, the smaller the 1-Wasserstein bound. As a result, we would like to choose $\hat{\eta}$ to be as large as possible, and the equation (3.10) provides an explicit value that $\hat{\eta}$ can take (which is already as large as possible from our theory). We will add more discussions and intuitive explanations in the revised version of the paper.
>
> * Our bound in Theorem 3.2 for the strongly convex case $O(\frac{1}{n\mu^5})$ is indeed worse than the $O(\frac{1}{n\mu})$ bound given in [HRS16] in terms of its dependency to the strong convexity constant $\mu$. That being said, as we mentioned in the paper, the main novelty of the results for the strongly convex case is *not* the bound itself we derived, but *how* we obtained it. Here, our novelty is the introduction of the proof technique, which we believe is very novel and flexible: can be applied strongly convex / non-convex (dissipative) losses, and it can be rather easily extended to other optimization algorithms like SGD-momentum – will hopefully form a fertile ground for further articles.
>
> * The bound in Theorem 4.1 is indeed not tight due to the persistent $K/m$ term that does not vanish to zero as the number of samples $n$ increases. However, the bound is **not vacuous** unless $n$ is very large: depending on the value of $K$ (the measure of nonconvexity), the bound can be still informative for a nonasymptotic $n$. On the other hand, the existing algorithmic stability bounds (cited in the paper) for similar non-convex problems typically increase with the number of iterations and become infinite and actually vacuous for any range of $n$ as the number of iterations increases. Compared to those bounds, our bound does not increase with the number of iterations, which we believe is a significant improvement.
>
>
> **Questions:**
>
> * Often, the Lyapunov function is required to be bounded from below by a positive constant, and some mathematics literature for simplicity requires $V\geq 1$. In order to create a positive function that is greater than $1$, the simplest function one can come up with is a quadratic plus $1$. The quadratic choice for the Lyapunov function is quite standard for SGD and related algorithms and is usually the first one to try. On the other hand, in order to use the strong-convexity and dissipativity conditions, it is easier to work with $|\theta-\theta_{\ast}|^{2}$ instead of $|\theta|^{2}$. The basic principle of building a Lyapunov function is to show a drift condition can be satisfied so that the expected value of the Lyapunov function can be uniformly bounded in time so that it will not grow as time increases. That being said, there is no principled guideline to design a Lyapunov function for an arbitrary optimization algorithm and objective function; however, for popular algorithms and problems classes several Lyapunov functions have already been developed as we mentioned in Line 100. Note that in optimization, to show convergence, we usually need the Lyapunov function to go to zero as iteration number increases; here because we do not need to show convergence to a point, it suffices that Lyapunov function over the iterations stays as a constant.
>
> * The paper considers noisy SGD for the nonconvex case. The noise is useful in this case because in order to apply Wasserstein perturbation theory [RS18]: we need contraction in Wasserstein distance (geometric ergodicity). Informally, in the nonconvex case, the Markov chain associated with SGD might not be geometrically ergodic if the only source of randomness comes from minibatching. Adding noise in this case enables the Markov chain to access all the regions of the state-space with non-zero probability and makes the chain geometrically ergodic.
>
> Formally, in the strongly-convex case, one can easily use the synchronous coupling method to obtain Wasserstein contraction such that no additional noise is needed. However, in the non-convex case, under a more general dissipativity condition, Wasserstein contraction can not be obtained by using the synchronous coupling method. Instead, in order to obtain Wasserstein contraction (geometric ergodicity), we apply the theory developed in [HM11] which says that as long as the Markov chain satisfies a drift condition (Assumption A.1) that relies on the construction of an appropriate Lyapunov function and a minorization condition (Assumption A.2), the Markov chain would converge in some weighted total variation distance that can imply convergence in 1-Wasserstein. The dissipativity condition helps us to show that we can indeed construct a Lyapunov function that satisfies Assumption A.1. However, a minorization condition (Assumption A.2.) requires a certain mixing property that requires some continuity-type assumptions on the noise structure, and this is where we need the additional noise (since the noise from the mini-batch has discrete nature and is not sufficient to obtain mixing property, i.e the minorization property.)
>
>
> ** Minor comments **
>
> We will fix these typographical errors, and revise along your suggestions. Thanks for the feedback.

---

> > ### Comment · Reviewer_yPZ3 · 2023-08-19
> >
> > Thank you for the point-to-point response. While the results for the strongly convex cases are not tight, I appreciate the novelty of the analysis. One thing I am still not quite clear is about the dependency on $K/M$. The authors mention that the bounds involving this term are not vacuous. These terms are involved in Assumption 3.3. It seems that this assumption is not quite related to the sample size, and therefore it is not clear to me how this term is not vacuous.

---

> > > ### Author Response · Authors · 2023-08-19
> > >
> > > Thank you very much for going over our rebuttal.
> > >
> > > Regarding the $K/m$ term, we thought the concern was about this term does not go to zero as the sample size increases: hence the bound becomes vacuous when $n \to \infty$ (please let us know if we misunderstood your concern).
> > >
> > > What we tried to convey in our response was, we agree with you that the bound is not tight for large $n$ for the reason we mentioned above -- it indeed becomes vacuous **when $n$ goes to infinity**. However, when $n$ is not large, we believe that the bound can still be informative. More precisely, as the bound is essentially
> > >
> > > $$C/n + K/m $$
> > >
> > > for **not large** $n$, the first term $C/n$ can dominate the $K/m$ term. Hence in such case, we stil obtain a meaningful bound, which **does not** explode as the number of iterations goes to infinity (as opposed to all existing bounds to our knowledge).
> > >
> > > We will mention this explicitly in the new version.
> > >
> > > We hope that this clarifies the concern. We would be happy to respond if there is any other questions.

---

### Official Review · Reviewer_tckZ · 2023-07-24

**Soundness:** 2 fair
**Presentation:** 2 fair
**Contribution:** 2 fair
**Rating:** 5
**Confidence:** 5

**Summary:**

The paper derives Wasserstein stability bounds for a variety of cases for a "surrogate loss" under convex/non-convex settings.

**Strengths:**

The problem is interesting. The bounds in the non-convex case can be useful. The paper is well-written.

**Weaknesses:**

I am not sure how novel the convex part is, also noted by authors. This part seems to remove the projection step, however, also done for a surrogate loss, so I am not sure if these are comparable. Also for the non-convex case, it is hard to see the relationship w.r.t. existing results due to the notion of surrogate loss.

**Questions:**

I have the following questions.

One thing I did not understand is the notion of the surrogate loss (as noted). The relationship between the original cost function $f$ and $\ell$ is never clarified (I checked the appendix and no definition of this). It seems that the whole paper is written for $\ell$, this is quite confusing. Why would authors not assume $f$ is $L$-Lipschitz and derive everything for $f$ under this restrictive assumption? This should be made clear as there is no way for a reader to understand what is going on in this paper, fundamentally, if the notion of the surrogate and its relationship to $f$ is not made clear at the beginning of this work. Note also that in Definition 2.1, it reads like the paper cited (HRS16) defines the stability for a surrogate loss, whereas there is no mention of surrogate loss in HRS16.

Upon checking a few related work which uses surrogate loss notion, it seems that the convergence in function value may not happen here. This of course makes the comparison impossible with relevant results in the literature.

Does any surrogate loss work for these results? Do the results hold for a particular selection of losses, e.g., sometimes depends on $p$ as in earlier works? Please clarify.

**Limitations:**

Theory work.

---

> ### Author Rebuttal · Authors · 2023-08-09
>
> We thank the reviewer for their time invested in our paper.
>
> As far as we can understand, the reviewer has two main concerns: 1) novelty of the strongly convex part and 2) the use of the surrogate loss function. Below, we clarify both of these points and we hope that the reviewer could reconsider their score based on our explanations.
>
> * Novelty of the strongly convex part:
>
> As we mentioned in the paper (and also as the reviewer acknowledged as well), the main novelty of the results in the strongly convex part is **not** the bound itself we derived, but **how** we obtained it. Here, our novelty is the introduction of the proof technique, which we believe is novel and very flexible: can be applied strongly convex / non-convex (dissipative) losses, and it can be rather easily extended to other optimization algorithms like SGD-momentum – will hopefully form a fertile ground for further articles. We hope that the reviewer could reposition our contributions from this perspective.
>
> * The use of the surrogate losses
>
> We agree that the requirement of surrogate losses is a drawback of our framework. However, based on the reviewer’s concerns, we suspect that the use of surrogate losses might seem more daunting than it actually is. We will now clarify why we need surrogate losses, and we will provide an example, which we believe is a very natural case and has been used in prior studies.
>
> *The need for surrogate losses:* As our bounds are based on 1-Wasserstein distance, we need the surrogate loss $\ell$ to be a Lipschitz continuous function. On the other hand, for the original loss $f$ we need some sort of convexity (e.g., strongly convex, convex, or dissipative) and we need the gradient of $f$ to be Lipschitz continuous. Unfortunately, under these assumptions, we cannot further impose $f$ **itself** to be Lipschitz because there is no function that satisfies these assumptions. Hence, the requirement for the surrogate losses.
>
> *Example for surrogate losses:*
>
> Example 1: We can choose the surrogate loss as the *truncated loss*, such that:
>
> $$\ell(\theta,x) = \min(f(\theta,x) , C)  $$
>
> where $C>0$ is a chosen constant. This can be seen as a “robust” version of the original loss, which has been widely used in robust optimization and (only) conceptually linked to adding a projection step to the optimizer.
>
> Example 2: Another natural setup for our framework is the $l_2$ regularized Lipschitz loss that was also used in [FR2021] (see Section 3 in their paper). As opposed to the previous case, for the sake of this example let us consider $\ell$ as the true loss and $f$ as the surrogate loss.
>
> Then, we can choose the pair $f$ and $\ell$ as follows:
>
> $$ f(\theta,x) = \ell(\theta,x) + \lambda ||\theta||_2^2 $$
>
> where $\lambda >0$. Intuitively, this setting means that, we have a true loss $\ell$ which can be Lipschitz, but in the optimization framework we consider a regularized version of the loss – which has also been considered in various studies, the closest to us being [FR2021].
>
> We sincerely appreciate that the reviewer has gone over the literature for surrogate losses. We agree this part is a little bit overlooked by assuming the reader might be familiar with the concepts. In the next version, we will provide further explanation and examples about surrogate losses, as we noted above. We will also rephrase the part where we introduced the concept of algorithmic stability, as the reviewer suggested.
>
> If there are further questions, we remain at your disposal.
>
> –
>
> References:
>
> [FR2021] Tyler Farghly and Patrick Rebeschini. Time-independent generalization bounds for 364 SGLD in non-convex settings. In Advances in Neural Information Processing Systems, 365 volume 34, pages 19836–19846, 2021.

---

> > ### Comment · Reviewer_tckZ · 2023-08-18
> >
> > Thanks for your response. I think however the main difficulty of obtaining such results in the literature is precisely lies on the technical challenge of using the original cost function to measure the performance, whereas, again as all results are stated with a vague notion of surrogate loss here, I do not think there is any comparison to the rest of the literature (I'm surprised, also, that other reviewers commented on the results as if they apply to the standard function value setting).
> >
> > Essentially, while you are running the algorithm using the gradients of $f(\theta, x)$, you are _measuring_ the performance w.r.t. to an arbitrary loss (since it's not specified here) $\ell$. Naturally $\min_{\theta} \ell(\theta)$ could arbitrarily differ from the true minimum. For me, this makes it impossible to assess the results here.
> >
> > Is there a way to assume $f$ and $\ell$ are "close" in some way, which can then be used in the error bounds as an extra term? I think this really needs clarification for the reader -- if the results are interpreted without this observation, then it may not even serve the community positively.

---

> > > ### Author Response · Authors · 2023-08-18
> > >
> > > Thank you for going over our rebuttal.
> > >
> > > We shall reiterate that there are various generalization bounds that have to rely on surrogate losses for different technical reasons. For recent examples, we kindly request you to check the settings in [FR2021, RZGS23, RBG+23]. Using the original loss in the analysis is usually not the only challenge -- or not the main challenge in general (yet we agree it is indeed a challenge); there might be several other important challenges that need to be solved, and solutions to such challenges can still be valuable to the community.
> > >
> > > On the other hand, sometimes it is even more desirable to use surrogate losses. For example, consider the classification task where the loss function we are interested in is a 0-1 loss function which is non-smooth and non-convex in nature. Hence, one runs gradient descent on (squared) hinge loss or logistic loss to better optimize the model. But eventually, the final risk bound is obtained in terms of 0-1 loss function risk and we are not interested in getting the risk bound on logistic loss or(squared) hinge loss. For the details, please refer to Bartlett et al. (2006).
> > >
> > > That being said, we agree with you that the setting might seem arbitrary for a reader who is not accustomed to surrogate losses. As you mentioned, in all the examples we have provided, the two losses are "close" to each other in certain senses. For instance in Example 1 we gave in the first response, as $C \to \infty$ the losses match. For Example 2, the two losses match as $\lambda \to 0$.
> > >
> > > We can definitely include an additional error term in the bounds coming from the use of surrogate losses, as you suggested. Essentially this will include a term in the following form:
> > >
> > > $$ | \mathbb{E}_{\theta, x}[f(\theta, x) - \ell(\theta,x) ] |  $$
> > >
> > > which measures how much the surrogate loss deviates from the original loss **on average**, and this can be a small value depending on the distribution of the algorithm output.
> > >
> > > We agree that this would increase the clarity of the presentation, we will add this in the next version. Thank you for the suggestion.
> > >
> > > We hope that this would address the concern and we remain at your disposal if there is any more questions.
> > >
> > > (1). Bartlett, Peter L., Michael I. Jordan, and Jon D. McAuliffe. ``Convexity, classification, and risk bounds.'' Journal of the American Statistical Association.

---

> > > > ### Comment · Reviewer_tckZ · 2023-08-18
> > > >
> > > > Thanks for your response.
> > > >
> > > > In your previous comment above, I think in both cases, different challenges may arise and it is not as simple as considering the limiting situations. For example, if you include the expectation in your last comment and if this difference is $\lambda \|\theta\|^2$ as in your Example 2, then do you need second moment bounds to be proven? (I may be mistaken here too).
> > > >
> > > > I am happy to reconsider my rating, if you could include very simple surrogate functions here (that are motivated by practice) and could develop the result in its full generality (that expectation term you wrote being also bounded, not just left in the bound and without introducing any compactness assumptions in parameter space which was one of the strengths of your paper). The paper is currently written in a way that the reader has to be an expert in the latest developments in this field to guess what you mean by surrogate function (which is only mentioned in a few papers) and what kind of problems you solve with these bounds. As seen from your comments, this is not trivial and the results are not straightforward to adapt.
> > > >
> > > > This will remain my main concern for this otherwise nicely written paper.

---

> > > > > ### Author Response · Authors · 2023-08-19
> > > > >
> > > > > Let us derive the additional error term in more detail for Example 2. Specifically, let us consider the following setting:
> > > > >
> > > > > $\ell$ is convex and Lipschitz in the first parameter and let us define $f$ as we did previously:
> > > > >
> > > > > $$f(\theta,x) = \ell(\theta,x) + \frac{\mu}{2} ||\theta||^2 $$
> > > > >
> > > > > Then $f$ is $\mu$-strongly convex. Further consider that we initialize SGD from 0 for simplicity, i.e., $\theta_0 = 0$ and set the batch size to 1 for notational clarity. Denote $\theta = \theta_k$ as the k-th iterate of SGD when applied on $\hat{F}(\theta,X_n)$ (Eqn 1.1).
> > > > >
> > > > > Further define the minimum:
> > > > >
> > > > > $$\theta^\star_{X_n} = \arg\min_{\theta} \hat{F}(\theta,X_n). $$
> > > > >
> > > > > We can now analyze the error induced by the surrogate loss as follows: (openreview did not render the latex code correctly, hence we provide it as an image):
> > > > >
> > > > > https://ibb.co/99Mqg34
> > > > >
> > > > > The fourth line follows from standard convergence analysis for SGD, e.g., Theorem 5.7 of https://gowerrobert.github.io/pdf/M2_statistique_optimisation/grad_conv.pdf
> > > > >
> > > > > Here, we define $\sigma_{X_n}$ as the stochastic gradient noise variance:
> > > > >
> > > > > $$\sigma_{X_n} := \mathrm{Var} [ \nabla f(\theta^\star_{X_n}, x_i) ] $$
> > > > >
> > > > > where for a random vector $V$ we define $ \mathrm{Var} [V] := \mathbb{E} ||V - \mathbb{E}[V]||^2$.
> > > > >
> > > > > Hence, we can see that the error induced by the surrogate loss depends on the following factors:
> > > > >
> > > > > 1. The regularization parameter $\mu$
> > > > > 2. The expected norm of the minimizers
> > > > > 3. The step-size $\eta$
> > > > > 4. The expected stochastic gradient noise variance
> > > > >
> > > > > We can see that these terms can be controlled by adjusting $\mu$ and $\eta$.
> > > > >
> > > > > We will add this derivation in the next version.
> > > > >
> > > > > On the other hand, we would like to insist on the fact that, even though there is this additional term due to the use of the surrogate loss, *for this example* it does not have any practical significance:
> > > > >
> > > > > > In this setting we have a loss $\ell$ but we run the algorithm on the regularized loss $f$ to have better convergence properties. But finally we would like to understand if the algorithm generalizes on $\ell$ or not, we are typically not interested if the algorithm generalizes well on the regularized loss $f$.
> > > > >
> > > > > We hope that this would address your concerns.

---

> > > > > > ### Comment · Reviewer_tckZ · 2023-08-20
> > > > > >
> > > > > > Thanks for your response. I agree that this is a good first step and a good example. But, imho, the paper should have this approach from the beginning. I can see now that there are examples that make sense for this sort of result.
> > > > > >
> > > > > > I bumped my score from 4 to 5. I suggest authors to add a detailed discussion about the relationship between $f$ and $\ell$ with practical examples in the next version of their paper, to avoid confusing the reader.

---

> > > > > > > ### Author Response · Authors · 2023-08-20
> > > > > > >
> > > > > > > Thank you very much for going over the derivation and increasing your score.
> > > > > > >
> > > > > > > We will add a dedicated section in the next version as you suggested.

---

### Decision · Program_Chairs · 2023-09-21

**Decision:**

Accept (poster)

**Comment:**

In this paper, the authors have introduced Wasserstein stability bounds for stochastic optimization algorithms, utilizing tools derived from the Wasserstein perturbation bound for Markov chains. The outcomes not only recover some existing uniform stability bounds for SGD in the literature but also extend to encompass other nonconvex scenarios under certain assumptions. Reviewers  acknowledge the intriguing link forged between learning theory and applied probability and the technical innovation. This approach can find potential applications in other optimization algorithms.

While concerns were initially raised regarding the discrepancy between the surrogate loss function $f$ and the original loss function $\ell$, extensive discussions led to a consensus that there exist avenues (e.g.by adding L2 regularization terms) to mitigate this discrepancy.

Taking into account the collective recommendations from the reviewers, I recommend accepting the paper. I would like to encourage the authors to thoroughly address the raised comments, particularly by incorporating discussions about the relationship between $f$ and $\ell$ and bolstering this with practical examples from machine learning.